


# Predicting tidal heights for extreme environments: From 25 h observations to accurate predictions at Jang Bogo Antarctic Research Station, Ross Sea, Antarctica

Do-Seong Byun[1], Deirdre E. Hart[2]

[1]Ocean Research Division, Korea Hydrographic and Oceanographic Agency, Busan 49111, Republic of Korea
[2]School of Earth and Environment, University of Canterbury, Christchurch 8140, Aotearoa New Zealand

*Correspondence to*: Do-Seong Byun (dsbyun@korea.kr)

**Abstract.** Accurate tidal height data for the seas around Antarctica are much needed, given the crucial role of tidal processes as represented in regional and global climate, ocean and marine cryosphere models. Though obtaining long term sea level records for traditional tidal predictions is extremely difficult around ice affected coasts. This study evaluates the ability of a relatively new, tidal species based approach, the Complete Tidal Species Modulation with Tidal Constant Corrections (CTSM+TCC) method, to accurately predict tides for a temporary tidal station in the Ross Sea, Antarctica using records from a nearby reference station characterized by a different regime. Predictions for the 'mixed, mainly diurnal' regimes of Jang Bogo Antarctic Research Station (JBARS) were made and evaluated based on summertime (2017; and 2018 to 2019) short-term (25 h) observations at this temporary station, along with tidal prediction data derived from yearlong observations (2013) from the nearby, 'diurnal' regime of Cape Roberts (ROBT). Results reveal the CTSM+TCC method can produce accurate (to within ~5 cm Root Mean Square Errors) tidal predictions for JBARS when using short-term (25 h) tidal data from periods with higher than average tidal ranges (i.e. tropic-spring periods). Predictions were successful due to the similar relationships between the main tidal constituents' ($K_1$ and $O_1$ tides) phase-lag differences at the prediction and reference stations, and despite these tidal stations being characterized by different tidal regimes according to their form factors (i.e. mixed, mainly diurnal versus diurnal). We demonstrate how to determine optimal short-term data collection periods based on the Moon's declination. The importance of using long period tides to improve tidal prediction accuracy is also considered, along with the characteristics of the different decadal scale tidal variations around Antarctica, from the four major FES2014 tidal harmonic constants.

## 1 Introduction

Conventionally, yearlong sea level records are used to generate accurate tidal height predictions via harmonic methods (e.g. Codiga, 2011; Foreman, 1977; Pawlowicz et al. 2002). Obtaining long term records for such tidal analyses is extremely difficult for sea ice affected coasts, like that surrounding Antarctica (Rignot et al. 2000). However, Byun and Hart (2015) developed a new approach to successfully predict tidal heights based on as little as ≥25 h of sea level records, combined with nearby reference site records, using their Complete Tidal Species Modulation with Tidal Constant Corrections (CTSM+TCC) method, on the coasts of Korea and New Zealand. Demonstrating the usefulness of this method for generating accurate tidal predictions for new sites on sea ice affected coasts is the motivation for this study. We focus on the Ross Sea, Antarctica, as our case study area.

Long-term, quality sea level records in the Ross Sea are few and far between, and include observations from gauges operated by New Zealand at Cape Roberts (ROBT); by the United States in McMurdo Sound; and by Italy at Mario Zucchelli Station, all in eastern Terra Nova Bay. Permanent sea level gauge installations in this extreme environment must accommodate or somehow avoid surface vents freezing over with sea ice, as well as damage to subsurface instruments from icebergs. At ROBT,





these issues have been avoided by sheltering the sea level sensor towards the bottom of a 10 m long hole, drilled through a
large shore boulder, from its surface ~2 m above the sea and sea ice level, to ~6 m below sea level below the base of the sea
ice (Glen Rowe, Technical Leader Sea Level Data, New Zealand Hydrographic Authority, *pers. comm.* 13 Dec. 2019). In the
absence of a suitable permanent gauge site, such as the current situation at the Korean Jang Bogo Antarctic Research Station
(JBARS), hydrographic surveys are best conducted during the summertime predominantly sea ice free window around mid-
January to mid-February. Even then, mobile ice (Figure 1) and severe weather events frequently hinder such surveys via
instrument damage or loss, not to mention the logistical difficulties of instrument deployment and recovery (Rignot et al.
2000). Accurate tidal records from the Ross Sea and other areas around Antarctica are thus scarce compared to those available
from other regions, though these data are much needed given the crucial role of tidal processes around this continent (Han and
Lee, 2018; Han et al., 2005; Jourdain et al., 2018; Padman et al., 2002; 2003; 2008; 2018).
Floating ice shelves occupy around 75% of Antarctica's perimeter (Padman et al., 2018). Tidal oscillations at the ice-ocean
interface influence the location and extent of grounding zones (Padman et al., 2002; Rosier and Gudmundsson, 2018), and
control heat transfers and ocean mixing in cavities beneath the marine cryosphere (Padman et al., 2018; Wild et al., 2019) and
the calving and subsequent drift of icebergs (Rignot et al. 2000). Tides also affect variability in polynyas; patterns of seasonal
sea ice; and thus the functioning of marine ecosystems (Han and Lee, 2018). In addition, tides affect the dynamics of landfast
sea ice, which provides aircraft landing zones for Antarctic science operations (Han and Lee, 2018).
Accurate Antarctic region tidal input data are needed for models examining changes in global climate and ocean circulation,
including for the generation of Antarctic bottom water (Han and Lee, 2018; Wild et al., 2019). Data on coastal tides are also
essential for studies of ice mass balances and motions (Han and Lee, 2018; Padman et al., 2008; 2018; Rignot et al. 2000;
Rosier and Gudmundsson, 2018; Wild et al., 2019). Ice thickness is typically measured via the subtraction of tidal height
oscillations from highly accurate, but relatively low frequency, satellite imagery based observations of ice surface elevation
and/or from in situ Global Positioning System (GPS) instrument observations (Padman et al., 2008). For floating ice, this
procedure is relatively straightforward but where ice shelves and glacier tongues occur, the mechanics of grounding zones and
ice flexure render the determination of ice thickness and motion very challenging (Padman et al. 2018; Rosier and
Gudmundsson, 2018), making the accuracy of the tidal height inputs crucial for effective ice modelling (Wild et al. 2019).
In this study, we tested applicability of Byun and Hart's (2015) CTSM+TCC method in an extreme observation environment
using 25 h short-term records from JBARS, our temporary tidal observation station, and yearlong data from ROBT, the nearby
reference station. Sect. 2 of this paper details the JBARS and ROBT observation data sets used to generate harmonic tidal
analysis results and CTSM+TCC tidal predictions. Sect. 3 explains the CTSM+TCC method and settings, while Section 4
demonstrates the CTSM+TCC tidal prediction capability. Sect. 5 discusses the generation of double tidal peaks, particularly
during low tides, and tidal characteristics around Antarctica.

## 69    2 Antarctica's major tides: Observations and background

### 70    2.1 Study sites and data records

The Korea Hydrographic and Oceanographic Agency (KHOA) survey team went to JBARS in Northern Victoria Land's Terra
Nova Bay, Ross Sea, Antarctica, in the summertime of 2017 (Figure 2) for a preliminary fieldtrip to conduct hydrographic
surveys and produce a nautical chart. This mission collected the first 19 day sea level records for JBARS: 10 min interval
observation data were recorded from 29 January to 16 February 2017 using a bottom-mounted pressure sensor (WTG-256S
AAT, Korea). High frequency signals were removed from the observation record using a fifth-order low-pass Butterworth
filter, with a cut-off frequency of 3 h. We use these data in our study's tidal prediction experiments as the temporary tidal
station's primary observation record.





For the purposes of a full-scale survey, 3 additional, discontinuous sea level observation records were measured by KHOA at
JBARS between 29 December 2018 and 11 March 2019, all at 10 min intervals using the same type of instrument. Of these,
the 20.54 day record produced between 29 December 2018 and 18 January 2019 comprised relatively high quality data with
small residuals. We used this additional dataset to verify the CTSM+TCC method tidal predictions generated from input
parameters derived from daily (25 h) slices of the 2017 sea level records. Due to the short duration of the KHOA survey team's
2017 and 2018 to 2019 forays into the Ross Sea, and the absence of in situ instruments, it was not possible to collect the
yearlong sea level records that are commonly employed to obtain reliable tidal constituents.
Approximately 269 km south of JBARS, there is a permanent tidal observation station named after its location on Cape Roberts
(ROBT), operated by Land Information New Zealand (LINZ) and recording at intervals since November 1990 (Figure 2). Five
minute interval sea level data have been collected at ROBT since November 2011 using Standard Piezometers (Model 4500,
GEOKON). Part of the 2017 record from this site was unavailable at the time of starting this research, so instead we chose as
our reference records the 2013 ROBT sea level data, a quality yearlong dataset with few missing points.

**2.2 Tidal characteristic analyses and descriptions**

Using the T_TIDE toolbox (Pawlowicz et al., 2002), we obtained the tidal harmonic constants of the 8 and 6 major tidal
constituents for ROBT and JBARS, respectively. In order to separate out the two major diurnal ($K_1$ versus $P_1$) and semi-diurnal
($S_2$ versus $K_2$) tide constituents from the short term records at JBARS, we used the inference method. That is, we used inference
parameters (i.e., amplitude ratios and phase-lag differences) for each tidal constituent pair ($K_1$ versus $P_1$; and $S_2$ versus $K_2$)
derived from harmonic analysis of records from the nearby ROBT reference station. Analysis revealed that the dominant tides
in this area are diurnal ($O_1$ and $K_1$), with the second most important tides being semi-diurnal ($M_2$ and $S_2$) (Table 1). These four
tides were characterized by similar amplitudes between ROBT and JBARS: 21.1 and 19.6 cm for $O_1$; 20.5 and 16.3 cm for $K_1$;
5.3 and 6.7 cm for $M_2$; and 4.9 and 6.4 cm for $S_2$. Note that the diurnal amplitudes were slightly larger at ROBT than at JBARS,
whereas the semi-diurnal amplitudes were slightly smaller at ROBT than at JBARS. Despite the relatively close distance (269
km) between ROBT and JBARS in tidal terms, the phase-lags showed slightly different values. The amplitude differences
result in slightly different tidal patterns as indicated by the two sites' tidal form factors ($F$). At ROBT $F$ is 4.1 while at JBARS
$F$ is 2.7: that is, ROBT has 'diurnal' type tides whereas JBARS has 'mixed, mainly diurnal' type tides.
Next we explored the characteristics of the four main tidal constituents around the entire Antarctic continent, using the
FES2014 database (Carrère et al., 2016). The horizontal distributions of the co-amplitudes and co-tides for $K_1$, $O_1$, $M_2$ and $S_2$
show that the diurnal tides rotate in an anticlockwise direction around Antarctica, with increasing co-amplitudes towards the
south, in particular, towards the Ronne and Ross ice shelves and hinterlands (Figure A1). In contrast, the $M_2$ and the $S_2$ tides
exhibit more complex patterns, with 5 and 7 amphidromic points respectively around Antarctica. Most of the semi-diurnal
tides rotate clockwise around their amphidromic points, except at one amphidromic point occurring ~150° W, where the $S_2$
tide rotates in an anticlockwise direction (Figure A2). The semi-diurnal co-amplitudes increase landwards in the Weddell Sea
but reduce across the entire Ross Sea quadrant, with relatively low semi-diurnal tide co-amplitudes (>7 cm) in this area. These
FES2014 results reveal that the tides of the Ross Sea are very different to regimes elsewhere in Antarctica.

**3 Using the CTSM+TCC tidal prediction methodology in the Ross Sea**

In this study, we used the -CTSM+TCC method (Byun and Hart, 2015) to predict tidal heights for JBARS. This prediction
approach is based on the idea of being able to use comparisons between the tidal harmonic constants at a *temporary observation*
*station* (JBARS in our study) and at a nearby *reference tidal observation station* (ROBT in this case) that is situated in an area
with similar tidal characteristics to that of the temporary observation station. It requires three data sets: long-term (ideally ≥183
days duration, from anytime) sea level records ($LH_R$) from the reference station, plus concurrent 25 h sea level records from



both the temporary observation station (SH$_O$) and the reference station (SH$_R$). Note that Byun and Hart (2015) recommend
using short-term data from periods with larger than average tidal ranges (e.g., in their situation these were spring tide periods
due to their study site having a semi-diurnal tidal regime) to produce accurate CTSM+TCC predictions, with periods of below
average tidal ranges (e.g., neap records) producing less accurate predictions. They also recommended the use of temporary
records gathered during periods of calm weather, to minimize errors due to atmospheric influences.
A complicating factor in this study was that, for the 2017 summertime period when SH$_O$ were recorded at JBARS, the ROBT
records were poor quality, including multiple missing data up until 12 February 2017. As such we did not start with two quality,
concurrent short-term observation records from our 2 stations. This issue was solved simply, using T_TIDE (Pawlowicz et al.,
2002) to produce accurate 10 min interval 2017 yearlong tidal height predictions for ROBT, based on LH$_R$ - that station's 2013
yearlong and high quality record. In short, the LH$_R$ dataset was harmonically analyzed to obtain harmonic constants for the
tidal constituents. In turn, these harmonic constants were used to produce the modulated amplitudes ($A_{r\eta}^{(s)}(\tau)$) and phase-lags
($\varphi_{r\eta}^{(s)}(\tau)$) over the 2017 tidal prediction period. Note that the period of the modulated species amplitudes and phase-lags
determine the tidal prediction period. Seventeen days of daily (25 h) data slices from the resulting 2017 tidal prediction data,
overlapping temporally with the SH$_O$ dataset, was then used as our SH$_R$ dataset. Figure 3 shows the modulated amplitudes and
phase-lags for the diurnal and semi-diurnal species, calculated from this SH$_R$ summertime 2017 tidal prediction data.
Using the CTSM+TCC approach, tidal predictions for the temporary station ($\eta_o(\tau)$) were initially derived from reference tidal
station predictions ($\eta_r(\tau)$) on the assumption that the tidal peculiarities between the two stations remain similar through time.
This step is expressed in Byun and Hart (2015) as:
$$\eta_r(\tau) = \sum_{s=1}^{k} A_{r\eta}^{(s)}(\tau) \cos\left(\omega_R^{(s)} t - \varphi_{r\eta}^{(s)}(\tau)\right) \tag{1}$$

with
$$A_{r\eta}^{(s)}(\tau) = \sqrt{\sum_{i=1}^{m}[f(\tau)_i^{(s)} a_i^{(s)}]^2 + 2\sum_{i<j}^{m}\left[f(\tau)_i^{(s)} a_i^{(s)}\right]\left[f(\tau)_j^{(s)} a_j^{(s)}\right]\cos\left\{\left(\omega_i^{(s)} - \omega_j^{(s)}\right)t + \left[V(t_0)_i^{(s)} + u(\tau)_i^{(s)} - g_i^{(s)}\right] - [V(t_0)_j^{(s)} + u(\tau)_j^{(s)} - g_j^{(s)}]\right\}} \tag{2}$$

and
$$\varphi_{r\eta}^{(s)}(\tau) = \tan^{-1}\left(\frac{\sum_{i=1}^{m} a_i^{(s)} \sin[(\omega_i^{(s)} - \omega_R^{(s)})t + V(t_0)_i^{(s)} + u(\tau)_i^{(s)} - g_i^{(s)}]}{\sum_{i=1}^{m} a_i^{(s)} \cos[(\omega_i^{(s)} - \omega_R^{(s)})t + V(t_0)_i^{(s)} + u(\tau)_i^{(s)} - g_i^{(s)}]}\right) \tag{3}$$

where superscript $s$ denotes the type of tidal species (e.g., 1 for diurnal species and 2 for semi-diurnal species), $\tau$ is time, $m$ is
the number of tidal constituents and $\omega_i^{(s)}$ and $\omega_R^{(s)}$ are the angular frequencies of each tidal constituent (subscripts $i$ and $j$) and
of representative tidal constituents (subscript $R$) for each species (e.g., K$_1$ and M$_2$ used as representative diurnal and semi-
diurnal species, respectively). For each tidal constituent, $a_i^{(s)}$ and $g_i^{(s)}$ are the tidal harmonic amplitudes and phase-lags,
$f(\tau)_i^{(s)}$ is the nodal factor, $u(\tau)_i^{(s)}$ is the nodal angle and $V(t_0)_i^{(s)}$ are the astronomical arguments. T_TIDE was used for tidal
harmonic analysis as well as for calculation of the nodal factors, nodal angles and astronomical arguments for the representative
species.
The amplitude ratio $\left(\frac{a_{o\eta}^{(s)}}{a_{r\eta}^{(s)}}\right)$ and phase-lag difference ($g_{o\eta}^{(s)} - g_{r\eta}^{(s)}$) of each representative tidal species between the temporary
tidal observation station (subscript $o\eta$) and the reference station (subscript $r\eta$) were then calculated from tidal harmonic
analysis of concurrent 25 h tidal records from both stations. In order to explore the best 25 h data window to use during this
step, we sliced the 17 day SH$_O$ and SH$_R$ records (from 29 January to 14 February 2017) into individual 'daily' data slices, each
starting at 00:00 and 25 h in duration. The 17 daily data slices from each station were harmonically analyzed (Figure 4) to
calculate daily amplitude ratios $\left(\frac{a_{ou}^{(s)}}{a_{r\eta}^{(s)}}\right)$ and phase-lag differences ($g_{ou}^{(s)} - g_{r\eta}^{(s)}$) for the diurnal and semi-diurnal representative
species (i.e., K$_1$ and M$_2$), as illustrated in Figure 5. The initial tidal predictions were then adjusted to represent those for the
temporary station ($\eta_o(\tau)$) by substituting the above amplitude ratios and phase-lag differences between the temporary and
reference stations into Eq. (1) as follows (Byun and Hart, 2015):




$\eta_o(\tau) = \sum_{s=1}^{k} A_{o\eta}^{(s)}(\tau) \cos\left(\omega_R^{(s)} t - \varphi_{o\eta}^{(s)}(\tau)\right)$        (4)
with $A_{o\eta}^{(s)}(\tau) = A_{r\eta}^{(s)}(\tau) \left(\dfrac{a_{o\eta}^{(s)}}{a_{r\eta}^{(s)}}\right)$ and        (5)
$\varphi_{o\eta}^{(s)}(\tau) = \varphi_{r\eta}^{(s)}(\tau) + g_{o\eta}^{(s)} - g_{r\eta}^{(s)}$        (6)
Substituting Eqs. (5) and (6) into Eq. (4), $\eta_o(\tau)$ can be expressed as:
$\eta_o(\tau) = \sum_{s=1}^{k} A_{r\eta}^{(s)}(\tau) \left(\dfrac{a_{o\eta}^{(s)}}{a_{r\eta}^{(s)}}\right) \cos[\omega_R^{(s)} t - (\varphi_{r\eta}^{(s)}(\tau) + g_{o\eta}^{(s)} - g_{r\eta}^{(s)})]$        (7)
where $t_0$ is the reference time, $t$ is the time ($t$) elapsed since $t_0$ and $\tau = t_0 + t$.
In addition to the 2017 tidal height prediction experiments, we examined the capacity of the CTSM+TCC method to generate
tidal predictions for the period 29 December 2018 to 18 January 2019 (hereafter referred to in shorthand as '2019
summertime'), using the same 2017 input data (i.e. using data from Figure 3 and Figure 5 in Eq. (7)). This 2019 summertime
prediction period corresponds to the second tidal observation mission made to JBARS by KHOA surveyors.
**4 Results**
**4.1 Tidal predictions**
The CTSM+TCC experiments produced seventeen datasets, each comprising 17 day long, 10 min interval tidal height
predictions for JBARS, together with data on the 'daily' (25 h) amplitude ratios and phase-lag differences between our two
observation stations (JBARS and ROBT). In order to evaluate these CTSM+TCC results, each predicted tidal height dataset
was compared with the concurrent JBARS field observations via Root Mean Square Error (RMSE) and coefficient of
determination ($R^2$) statistics.
As illustrated in Figure 6, the RMSE and $R^2$ results varied in relation to the JBARS tidal ranges, with greater accuracy evident
in predictions made using data derived from 25 h periods where the tidal range was higher than average. In the JBARS 'mixed,
mainly diurnal' type tide area of the Ross Sea, during our 2017 observation period, greater than average tidal ranges
corresponded to the spring tide period when the moon was near its maximum (tropic) declination. RMSEs between
observations and predictions ranged from 4.26 cm to 20.56 cm while $R^2$ varied from 0 to 0.94 across the 17 'daily' experiments.
Eleven of the experiments produced accurate results (i.e. excluding those based on 25 h input data derived from 31 January;
and 1 to 4 and 14 February records), with their RMSEs <5 cm and $R^2$ values >0.92. In contrast to the majority of successful
experiments, the experiment based on data derived from the '2 February' 25 h data slice produced predictions with very high
RMSE (20.56 cm) and very low $R^2$ (0.00) values. Notably, the 2 February tides were characterized by the smallest tidal range
(11.95 cm) of the temporary JBARS record, during an equatorial tide period. In contrast, daily datasets from periods with
relatively high tidal ranges (>83.5 cm) produced predictions with RMSEs <5 cm and $R^2$ values >0.92. The maximum spring
tidal range occurred on 9 February: the data slice from this occasion produced predictions with a low (but not the lowest)
RMSE (4.81 cm). The predictions with the lowest RMSE (4.259 cm) and highest $R^2$ value (0.941) were produced using inputs
derived from 25 h data recorded one day earlier, on 8 February 2017.
As with the 2017 predictions, RMSEs between the 2019 summertime predictions and observations were lower when generated
using input data derived from 25 h data slices from the 2017 tropic – spring (as opposed to equatorial and/or neap) tide periods
(Figure 7). As in the earlier experiments, the 2019 summertime predictions made using input data derived from the 8 February
2017 (25 h) data slice produced the lowest RMSE (5.3 cm) and highest $R^2$ (0.913) values of the 2019 summertime experiments
(Figure 8).
These results demonstrate that the CTSM+TCC method can be successfully employed to predict tidal heights for JBARS for
any particular period, using 25 h observation records gathered from tropic or tropic-spring tide periods with relatively calm





weather, together with yearlong sea level observation or prediction records from the nearby reference station ROBT, despite
the two stations having slightly different types of tidal regime.

**4.2 Determining the ideal short-term sea level observation period when using CTSM+TCC**

The previous section verified that the CTSM+TCC method can be used to generate accurate tidal predictions based on 25 h
sea level records, from periods with higher than average tidal ranges, for a temporary station in a 'mixed, mainly diurnal'
regime and a reference station in a 'diurnal' regime. The question arises as to how to determine the ideal day from which to
source the 25 h observation records in order to produce the most accurate tidal predictions.
For semi-diurnal or mixed, mainly semi-diurnal tidal regimes, we can estimate preferred temporary observation days based on
the moon's phase, without reference to tide tables. That is, spring tides commonly occur just a day or two after the full and
new moon, which reoccurs at a period of 14.7653 days. The time lag between the full or new moon and the spring tide is called
the age of the tide (*AT*).
Similarly, in a 'diurnal' tide regime or a 'mixed, mainly diurnal' tide regime (Figure 6), the preferred temporary observation
days can be estimated based on the lunar declination, which varies at a period of 13.6608 days. That is, maximum range tropic
tide days can be estimated for JBARS based on the day of the Moon's maximum and minimum declinations. The time between
the Moon's semi-monthly maximum (and minimum) declinations and its maximum effect on tidal range, called the age of
diurnal inequality (*ADI*), is commonly 1 to 2 days.
As shown in Figure 9, the maximum and minimum Moon's declinations during our 2 temporary summertime observation
periods occurred on 8 February 2017 (max) and on 6 January 2019 (min) respectively. The diurnal maximum tide tends to
occur ~1 day after the maximum declination, during one half of the tropic month and about 2 days after the minimum
declination during the other half of the tropic month.

**4.3 Comparison of ROBT and JBARS tidal species characteristics**

The CTSM+TCC tidal prediction method is based on the assumption that the tidal harmonic characteristics of each tidal species
are very similar between the temporary observation and reference stations. This is because the reference station tidal species'
CTSMs, derived from yearlong reference station sea level records or tidal harmonic analysis results, form the basis of the tidal
predictions for the temporary observation station. To test the validity of this assumption, we examined the phase-lag differences
of the 2 major diurnal ($K_1$ and $O_1$) and 2 major semi-diurnal tidal constituents ($M_2$ and $S_2$) using the age of diurnal inequality
(*ADI*) and the age of the tide (*AT*), calculated as:
$$ADI \ (day) = \left( \frac{g_{K_1} - g_{O_1}}{\omega_{K_1} - \omega_{O_1}} \right) / 24 \ , \ \text{and} \tag{7}$$
$$AT \ (day) = \left( \frac{g_{S_2} - g_{M_2}}{\omega_{S_S} - \omega_{M_2}} \right) / 24 \ , \tag{8}$$
where $\omega_{K_1}$ (= 15.0410686° hr$^{-1}$), $\omega_{O_1}$ (= 13.9430356° hr$^{-1}$), $\omega_{M_2}$ (= 28.9841042° hr$^{-1}$) and $\omega_{S_2}$ (= 30.0000000° hr$^{-1}$) are the
angular        speeds        of        the        $K_1$,        $O_1$,        $M_2$        and        $S_2$        tides,        respectively.
Results revealed that the *ADI* are very similar, and there is <1 day *AT* difference, between the 2 stations: the *ADI* values were
0.57 versus 0.23 day, while the and *AT* values were -2.30 versus -1.44 day, for ROBT versus JBARS (Table 1). These values
indicate that the tidal characteristics of the representative tidal constituents for each species between ROBT and JBARS are
very similar, in particular the dominant diurnal species. Hence the applicability and success of the CTSM+TCC method for
generating JBARS tidal predictions, using concurrent 25 h records from both stations and reference records from ROBT.





## 5 Discussion

### 5.1 Fortnightly tide effects around Antarctica

We have so far demonstrated that the CTSM+TCC approach can produce reasonably accurate tidal predictions (RMSE <5 cm, $R^2$ >0.92) for a new site in the Ross Sea, Antarctica, based on 25 h temporary observation records from periods with higher than average tidal ranges, plus nearby reference station records.

Our results compare favourably with those of Han et al. (2013), who reviewed the tidal height prediction accuracy of 4 models for Terra Nova Bay, Ross Sea: that is, TPXO7.1 developed by Egbert and Erofeeva (2002), FES2004 from Lyard et al (2006); the Circum-Antarctic Tidal Solution (CATS2008a) from by Padman et al. (2008), and the Ross Sea Height-Based Tidal Inverse Model (Ross_Inv_2002) from Padman et al. (2003). Han et al. (2013) compared the model datasets to 11 days of February 2011 in situ sea level observations, corrected for inverse barometer effects, and considered model usefulness for investigating tidal signals in satellite data from the Campbell Glacier tongue. The 4 models generated similar quality results to those generated by the CTSM+TCC method in this study, with $R^2$ values varying between 0.876 and 0.907, and RMSEs ranging from 3.6 and 4.1 cm.

However, as shown in Figure 8, our results appear to contain a changing bias in estimates occurring at fortnightly timescales, with predictions slightly overestimating tides during the period from the equatorial to tropic tides (the ETT), and slightly underestimating tides from the period between the -tropic to equatorial tides (the TET). This error pattern likely resulted from our application of CTSM+TCC only considering 2 major tidal species, those representing diurnal and semi-diurnal constituents, whilst ignoring long period tides.

In their GPS field measurement and modelling study of the Ronne Ice Shelf in the Weddell Sea, Rosier and Gudmundsson (2018) found that ice shelf and ice stream horizontal flows are strongly modulated at a variety of tidal frequencies, with a significant $M_{sf}$ tide correlated signal occurring across their field site. Modelling without vertical tidal oscillations produced horizontal ice flow rates almost 30% lower than observed. In an earlier Synthetic Aperture Radar (SAR) interferometry and tide model comparison study of the Ronne and Filchner Ice Shelves, Rignot et al. (2000) found that eight 'major' tidal constituents ($M_2$, $S_2$, $K_2$, $N_2$, $O_1$, $K_1$, $Q_1$, and $2N_2$,) plus an additional 18 'minor' constituents measurably influenced patterns of ice flexure and motion. The authors of both of these papers recommend the inclusion of both major and minor tidal constituents, including long period tides, for successful ice flow and ice-ocean front modelling.

Long period tides cycle across timeframes including 18.61 y, seasons, months and fortnights (Woodworth, 2012). Table 2 summarizes the characteristics of 6 long-period tides ($S_a$, $S_{sa}$, $M_{sm}$, $M_m$, $M_f$, $M_{sf}$) at the ROBT station, derived from tidal harmonic analysis of yearlong (2013) in situ records. Comparisons between Tables 1 and 2 reveal that the $S_a$ amplitude (5.8 cm) was similar to that of the $M_2$ (5.3 cm), the amplitudes of the $M_m$ and $M_f$ tides were >50 % of the $M_2$ (≥2.7 cm), while the $S_{sa}$ and $M_{sm}$ amplitudes were all minor (≤0.4 cm). While the 2013 $S_a$ amplitude was equivalent to that of the $M_2$, inter-annual variation in the $S_a$ harmonic constant is large (1.2 cm to 9.1 cm for amplitude; 75° to 131° for phase-lag, Table 3). This is because the $S_a$ constituent comprises both astronomical and seasonal components (Pugh, 1987). Hence our focus here on the error bias between the ETT and TET periods.

In order to verify the main cause of the apparent fortnightly prediction biases, in particular that found in the 2019 summertime results (Figure 8b), we examined the effects of two fortnightly period tidal constituents ($M_f$ and $M_{sf}$) at ROBT. Three 2019 summertime tidal prediction experiments were conducted: 1) *Srun* excluding all long-period tides (i.e. those in Table 2); 2) *Run1* incorporating the $M_f$ alone; 3) *Run2* incorporating the $M_f$ and the $M_{sf}$ alone.

Results revealed that exclusion of the $M_f$ tide (2.7 cm amplitude) alone can produce ETT and TET prediction biases (Figure 10a), with exclusion of the $M_{sf}$ tide (1.2 cm amplitude) intensifying the biases (Figure 10b). Thus, consideration of additional, fortnightly timescale tidal constituents in predictions is our recommended next step for improving tidal prediction accuracies for JBARS.





**5.2 Decadal scale tidal variations around Antarctica**
Tidal regime characteristics in the seas surrounding Antarctica mostly fall into three of the four daily form factor types, as
revealed by FES2014 model data. Figure 11 shows there are areas of 'diurnal' (DD); 'mixed, mainly diurnal' (MD); and
'mixed, mainly semi-diurnal' (MS) forms. Only in a small area half-way along the Weddell Sea coast of Antarctic Peninsula
do tides exhibit a 'semi-diurnal' form. Strong 'diurnal' tides ($F$>3) predominate in the Ross Sea area of West Antarctica,
around to the Amundsen Sea. In addition, a small area near Prydz Bay in East Antarctica exhibits diurnal and mixed mainly
diurnal tides. The rest of the seas surrounding Antarctica, including the Weddell Sea, are predominantly characterized by
'mixed, mainly semi-diurnal' tides.
Tides around the Weddell Sea coast are significantly amplified due to shoreline shape and bathymetric shoaling effects, with
the increase in semi-diurnal amplitudes ($a_{M_2} + a_{S_2}$) being more pronounced than those of the diurnal tides ($a_{K_1} + a_{O_1}$). Tidal
ranges >2 m are largely confined to the Weddell Sea region, with the exception of the area surrounding the $M_2$ tide
amphidromic point at the head of the Weddell Sea embayment, where relatively large 'mixed, mainly semi-diurnal' tides occur
thanks to the pronounced $M_2$ and $S_2$ amplitudes there.
The contrasting tidal environments of the Weddell and Ross Seas feature different tidal dynamics and, thus, different tidal
influences on their environments, across the full 18.61 y tidal cycle. Accurate (cm scale) quantification of the tidal cycle
patterns resulting from these different regimes are essential for calculating ice-sheet motion near Antarctica's different ocean
margins, based on the subtraction of ice flexure and tidal elevation changes from land ice elevation measurements (Wild et al.,
2019). Such studies contribute to our understanding of global climate models, providing estimates of ice sheet and glacier
flows to the sea.
A question arises as to how much the Weddell and Ross Sea tidal form differences can be explained by tidal height changes.
To answer this, we explored variation in nodal modulation correction factors (nodal factors and nodal angles) over an 18.61 y
cycle. Daily nodal modulation correction factor values for the 3 major lunar tide constituents ($K_1$, $O_1$ and $M_2$) were estimated
for JBARS over the 20 y period 2011 to 2030, as illustrated in Figure 12. Interestingly the diurnal, $O_1$ tide variations in nodal
modulation correction factors were the largest (nodal factor range = 0.3833; nodal angle range = 22.59°), with those of the $K_1$
tide being second largest (nodal factor range = 0.2320; nodal angle range = 17.86°). In comparison, those of the semi-diurnal,
$M_2$ tide were relatively small (nodal factor range = 0.0754; nodal angle range = 4.39°).
These nodal modulation correction factor variations have different implications for the Weddell and Ross Seas due to their
differing tidal regimes. The resulting variations in tidal height are less pronounced in the semi-diurnal Weddell Sea, while the
diurnal regime of the Ross Sea experiences large tidal range variations across 18.61 y cycles due to the influence of diurnal
nodal factor variation. Of note, variations in the nodal factors of the $O_1$ and $K_1$ tides are out of phase with that of the $M_2$ tide
(Figure 12a). Variations in the nodal angle of the $K_1$ tide is in phase with that of the $M_2$ tide but out of phase with that of the
$O_1$ tide (Figure 12b).
**6 Conclusions**
This paper has demonstrated the usefulness of the CTSM+TCC method for tidal prediction in extreme environments, where
long-term tidal station installations are difficult, using the Ross Sea in Antarctica for our case study. Here CTSM+TCC
methods can be employed for accurate tidal height predictions for a temporary tidal observation station using short-term (≥25
h) sea level records from this site, plus long-term (1 y) tidal records from a nearby reference tidal station. Essentially the
temporary and reference station sites must share similarities in their main tidal constituent and tidal species characteristics for
CTSM+TCC to produce accurate results.
Using this approach, an initial tidal prediction time series is generated for the temporary station using CTSM and the reference
station long-term records. The temporary station predicted time series can then be adjusted via TCC of each tidal species,





based on comparisons between the short-term temporary station observation record and its corresponding modelled
predictions, leading to improved accuracy in the tidal predictions.
This paper has further demonstrated that the CTSM+TCC approach can be employed successfully in the absence of concurrent
short-term (25 h) records from the reference station, since a tidal harmonic prediction program can be used to produce a
synthetic short-term record for the reference station based on a quality long-term record from that site.
The proper consideration of long-period tides in the CTSM+TCC approach remains a challenge, as outlined in this study, with
the solutions to this issue likely to improve the accuracy of CTSM+TCC tidal predictions even further. However, this study
demonstrates that the method can already produce tidal predictions of sufficient accuracy to aid scientists studying important
issues such as the rate and role of ice loss along polar coastlines.
**Code Availability**
The T_TIDE based CTSM code is available from https://au.mathworks.com/matlabcentral/fileexchange/73764-ctsm_t_tide.
**Data Availability**
The sea level data used in this paper are available from LINZ (2019) for selected ROBT records, with the remaining ROBT
records available by email application (customersupport@linz.govt.nz); and the JBARS records used are available on request
from KHOA (infokhoa@korea.kr). Details of the FES2014 tide model database are found in Carrère et al. (2016) and via
https://www.aviso.altimetry.fr/en/data/products/auxiliary-products/global-tide-fes.html.



**Appendix 1**

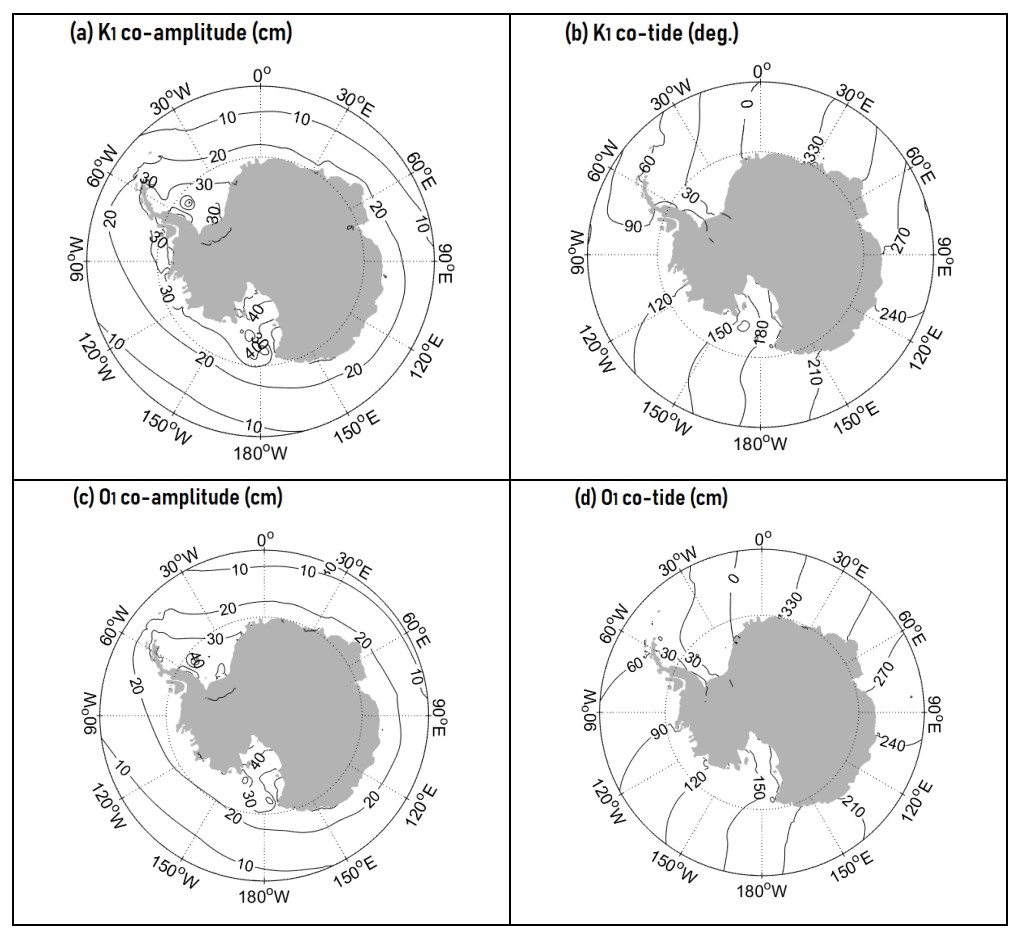

**Figure A1. Horizontal distributions of the $K_1$ and $O_1$ constituents' co-amplitudes (a, c) and co-tides (b, d) around Antarctica.**





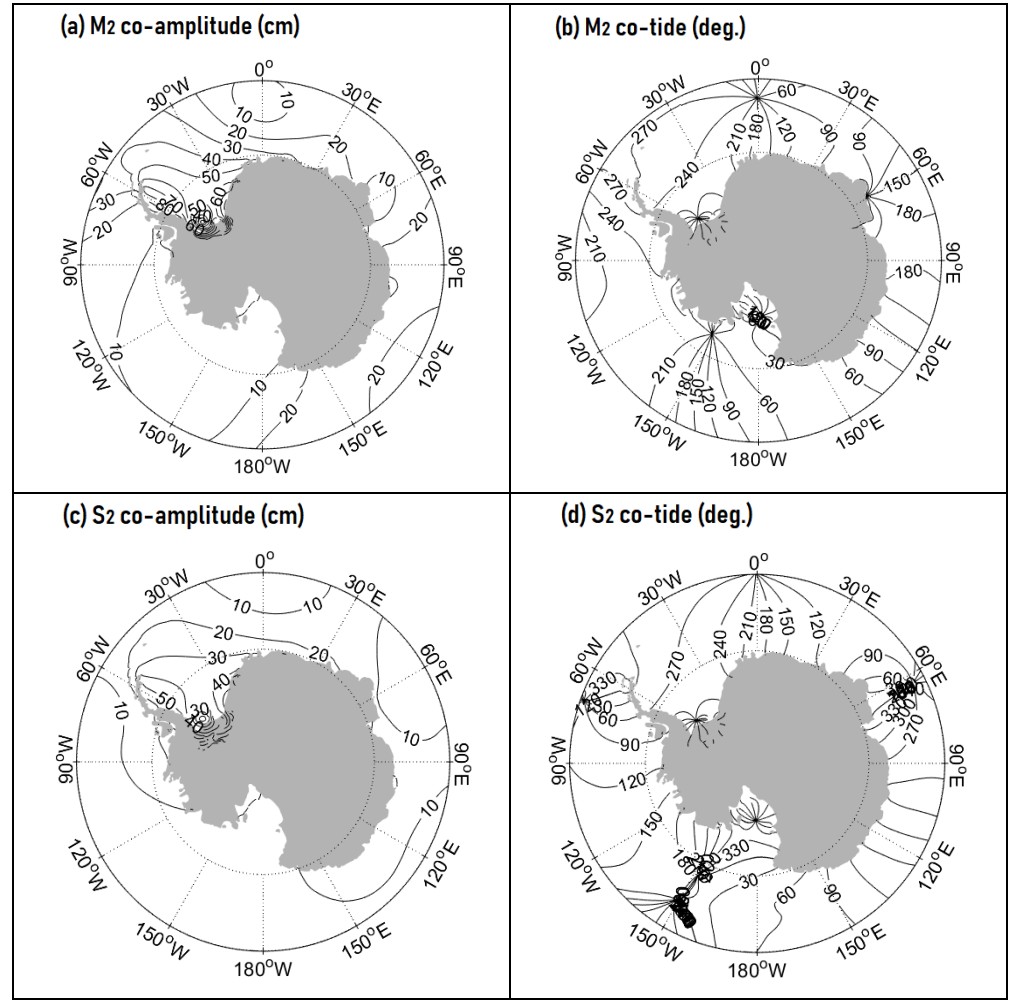

Figure A2. Horizontal distributions of the M$_2$ and S$_2$ constituents' co-amplitudes (a, c) and co-tides (b, d) around Antarctica.



**Author contribution**

D-SB conceived of the tidal prediction idea behind this paper, and wrote the results sections. Both authors worked on initial and final versions of the full manuscript.

**Competing interests**

The authors declare that the research was conducted in the absence of any commercial or financial relationships that could be construed as a potential conflict of interest.

**Acknowledgements**

We are grateful to Land Information New Zealand (LINZ) and the Korea Hydrographic and Oceanographic Agency (KHOA) for supplying the tidal data used in this research. A special thank you to Glen Rowe from LINZ for sharing his extensive knowledge of the Cape Roberts sea level gauge site and its records. Further, we gratefully thank Ms. Hyowon Kim at KHOA for her kind assistance with drafting figures.

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





**Table 1. Major tidal harmonic results for diurnal and semi-diurnal constituents from harmonic analyses of yearlong (2013) sea level**
**observations recorded at Cape Roberts (ROBT), and from 17 day sea level observations (29 January to 15 February 2017) and 20.54**
**day sea level observations (29 December 2018 to 18 January 2019) recorded at the Jang Bogo Antarctic Research Station (JBARS),**
**in Antarctica. For the JBARS tidal harmonic analyses, the inference method was applied to separate out the $K_1$ ($S_2$) and $P_1$ ($K_2$) tidal**
**constituents, using inference parameters estimated from the ROBT 2013 harmonic analysis. Phase-lags are referenced to 0°,**
**Greenwich.**

| Tidal constituents | | ROBT (2013) 369 days | | JBARS (2017) 17 days | | JBARS (2019) 21 days | | Note |
|---|---|---|---|---|---|---|---|---|
| | | $a_{r\eta}$ (cm) | $g_{r\eta}$ (°) | $a_{o\eta}$ (cm) | $g_{o\eta}$ (°) | $a_{o\eta}$ (cm) | $g_{o\eta}$ (°) | |
| Diurnal | $O_1$ | 21.1 | 202 | 19.6 | 208 | 16.0 | 208 | ROBT: Diurnal tides ($F$=4.1) |
| | $K_1$ | 20.5 | 217 | 16.3 | 214 | 14.9 | 216 | $ADI$=0.57 day |
| | $P_1$ | 6.6 | 215 | 5.2 | 213 | 4.8 | 214 | $AT$=-2.30 days |
| | $Q_1$ | 4.4 | 190 | - | - | - | - | |
| Semi-diurnal | $M_2$ | 5.3 | 5 | 6.7 | 4 | 6.3 | 34 | JBARS: Mixed, mainly diurnal tides ($F$=2.7) |
| | $S_2$ | 4.9 | 309 | 6.4 | 329 | 6.6 | 324 | $ADI$=0.23 day |
| | $N_2$ | 3.8 | 255 | - | - | - | - | $AT$=-1.44 days |
| | $K_2$ | 1.8 | 315 | 2.4 | 333 | 2.4 | 328 | |

**Note that $ADI$ and $AT$ denote the age of diurnal inequality and the age of the tide.**





**Table 2. Harmonic constants for 6 long-period tidal constituents, derived from harmonic analyses of yearlong observations (2013)**
**measured at the Cape Roberts sea level gauge (ROBT)**

| Constituent | | Period (day) | Angular speed (° hr$^{-1}$) | Amplitude (cm) | Phase-lag (°) |
|---|---|---|---|---|---|
| Solar annual | $S_a$ | 365.24 | 0.0410686 | 5.8 | 75 |
| Solar semi-annual | $S_{sa}$ | 182.62 | 0.0821373 | 0.1 | 352 |
| Lunar monthly | $M_{sm}$ | 31.81 | 0.4715280 | 0.4 | 57 |
| | $M_m$ | 27.55 | 0.5443747 | 2.9 | 139 |
| Lunar fortnightly | $M_{sf}$ | 14.77 | 1.0158958 | 1.2 | 281 |
| | $M_f$ | 13.66 | 1.0980331 | 2.7 | 153 |

**Phase-lags are referenced to 0°, Greenwich.**


**Table 3. Harmonic constants for the $S_a$ constituent derived from harmonic analyses of 4 separate yearlong observation records**
**(2008; 2011; 2012; 2013) measured at the Cape Roberts sea level gauge (ROBT)**

| Year | Amplitude (cm) | Phase-lag (°) |
|------|----------------|---------------|
| 2008 | 9.1 | 131 |
| 2011 | 1.2 | 90 |
| 2012 | 3.4 | 108 |
| 2013 | 5.8 | 75 |

**Phase-lags are referenced to 0°, Greenwich.**



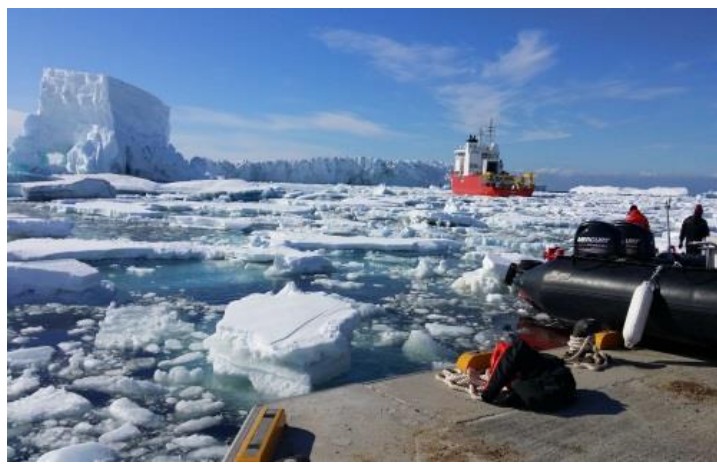


**Figure 1. Drifting ice, including icebergs and mobile sea ice, around the Jang Bogo Antarctic Research Station (JBARS).**




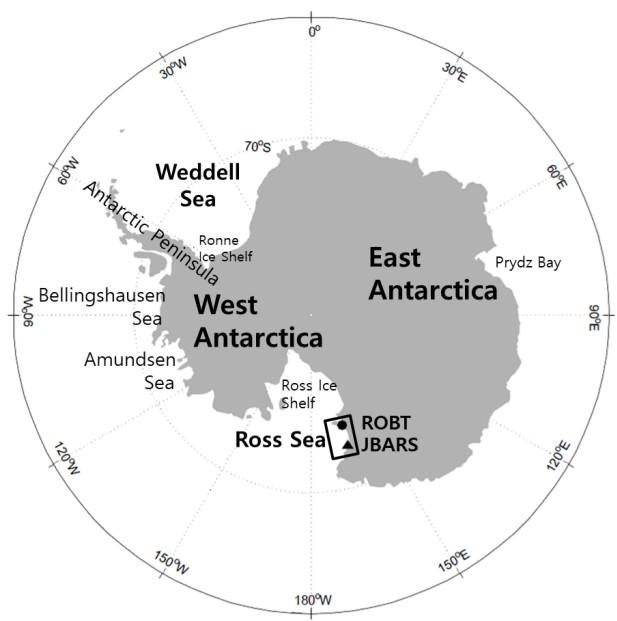


**Figure 2.** Map showing locations of two tidal observation stations in the Ross Sea of Antarctica: Jang Bogo Antarctic Research
Station (JBARS, ▲) and Cape Roberts (ROBT, ●).



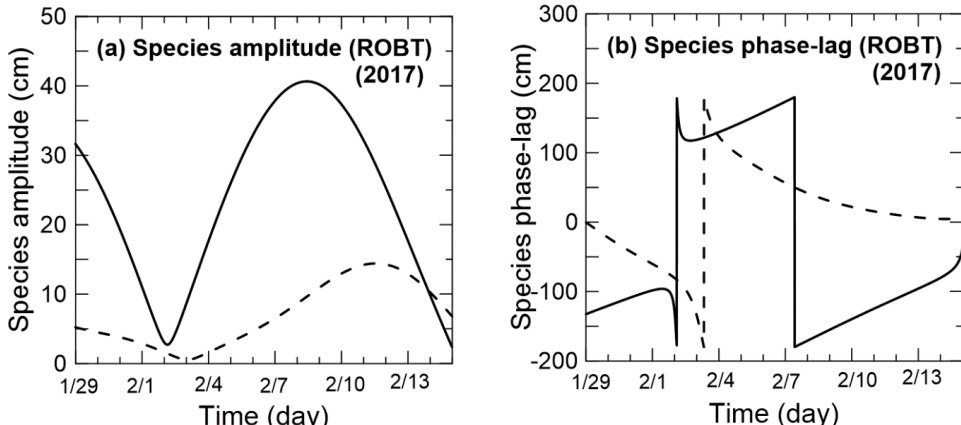

**Figure 3. Seventeen day time series (29 January to 14 February 2017) of the modulated tidal (a) species amplitudes and (b) phase-lags for the diurnal (solid lines) and semi-diurnal tides (dashed lines), estimated from the 2017 Cape Roberts (ROBT) tidal prediction data.**



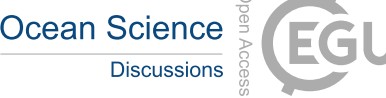


**Figure 4. Daily amplitudes and phase-lags of the K₁ tide (diurnal representative species) and M₂ tide (semi-diurnal representative**
**species) at ROBT and JBARS, estimated from 25 h daily data slices of the 17 day ROBT tidal predictions and JBARS sea level**
**observations, 29 January to 14 February 2017. Thick solid (K₁) and thin gray (M₂) lines in each panel indicate the amplitude and**
**phase-lag derived from results of the 369 day 2013 ROBT and 17 day 2017 summertime JBARS sea level record harmonic analyses,**
**respectively.**


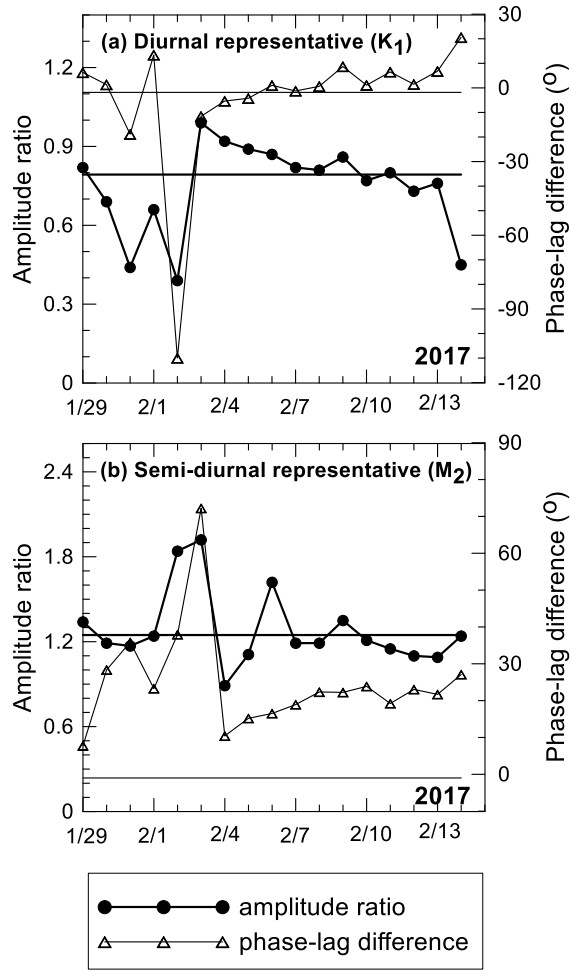

**Figure 5. The daily amplitude ratios and phase-lag differences of the (a) diurnal (K₁) and (b) semi-diurnal (M₂) species representative**
**tidal constituents, calculated harmonically from JBARS sea level data and ROBT predicted tidal height data, using harmonic inputs**
**derived from analysis of 'daily' (25 h) data slices. Thick solid and dashed lines in each panel indicate the amplitude ratio and phase-**
**lag differences, respectively for each tide, derived from harmonic analysis of the 17 day 2017 JBARS sea level data.**

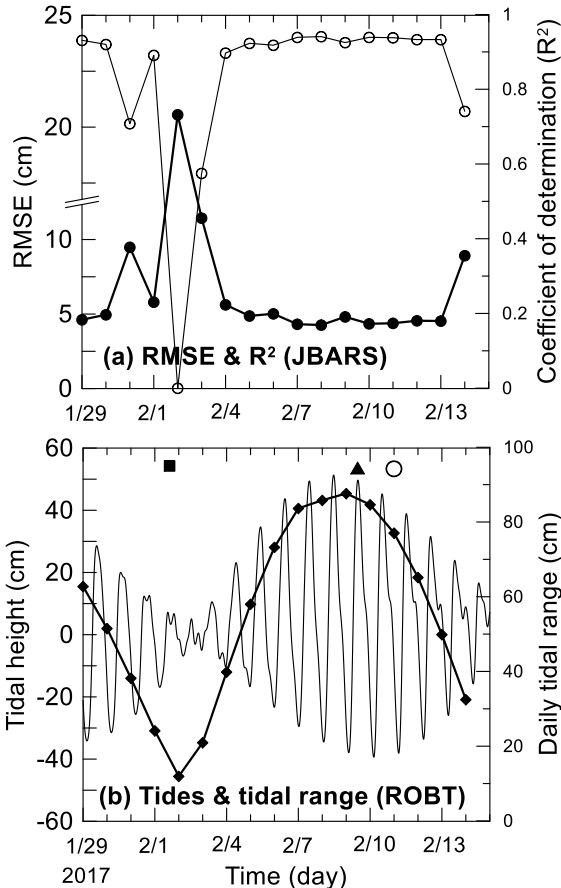

Key: ○: full moon; ▲: Moon's maximum

declination; ■: Moon's declination is zero.

**Figure 6. (a) Time series of Root Mean Square Errors (RMSE, thick line with ●) and coefficients of determination (R², thin line with ○) between JBARS 10 min interval sea level observations (29 January to 15 February 2017) and the CTSM+TCC prediction datasets generated for this site using harmonic analysis results from the daily (25 h) sea level data slices from JBARS plus concurrent daily (25 h) tidal prediction slices and harmonic analysis results from ROBT station's yearlong (2017) tidal predictions. (b) Time series of predicted 2017 tidal heights (thin line) and daily tidal ranges (thick line with ○) for ROBT, based on harmonic analysis of this station's 2013, 5 min interval sea level records, plus an indication of the moon's phase and declination.**



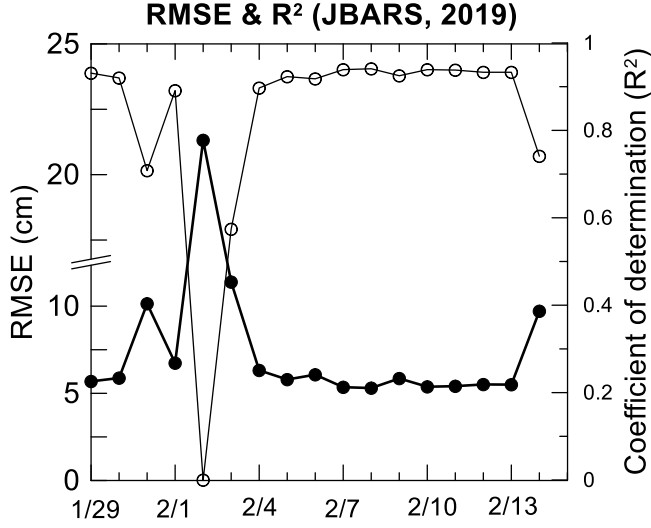

431

**Figure 7. Time series of Root Mean Square Errors (RMSE, thick line with ●) and coefficients of determination (R², thin line with ○)**
**between JBARS 10 min interval sea level observations (29 December 2018 to 18 January 2019) and the CTSM+TCC prediction**
**datasets generated for this site using harmonic analysis results from daily (25 h) summertime 2017 sea level data slices from JBARS**
**plus concurrent daily (25 h) tidal prediction slices and harmonic analysis results from ROBT station's yearlong (2017) tidal**
**predictions.**

437




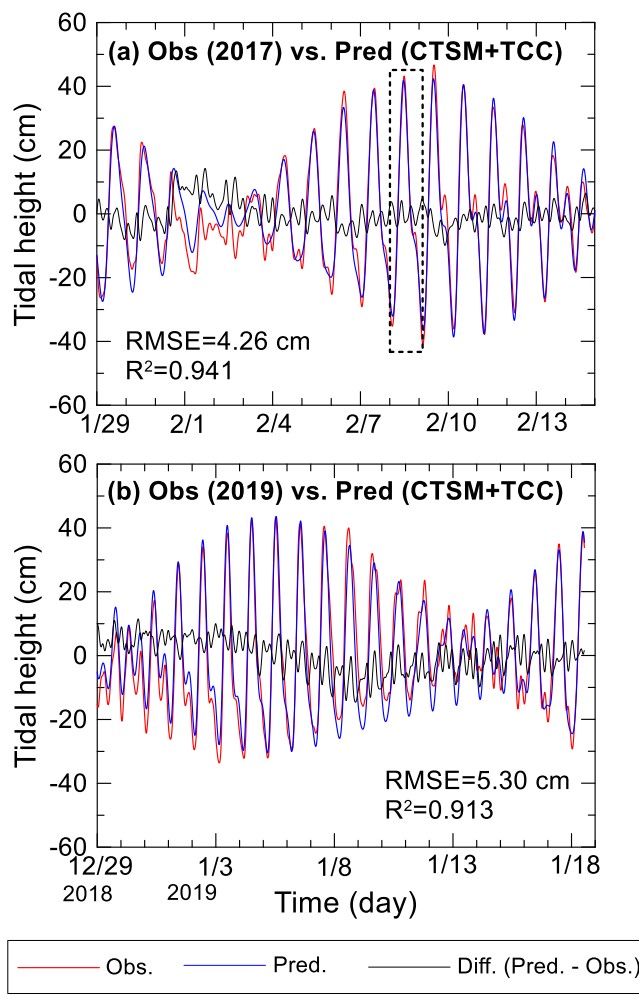

438

**Figure 8. Time series of JBARS sea level observations, predicted tidal heights, and sea level residuals (i.e. predictions minus observations) from (a) 29 January to 15 February 2017 and (b) 29 December 2018 to 18 January 2019. The JBARS predictions were generated using CSTM+TCC, with a daily (25 h) slice of local sea level observations from 8 February 2017 (dashed box in (a)), plus concurrent predictions and yearlong (2017), 5 min interval ROBT tidal predictions.**



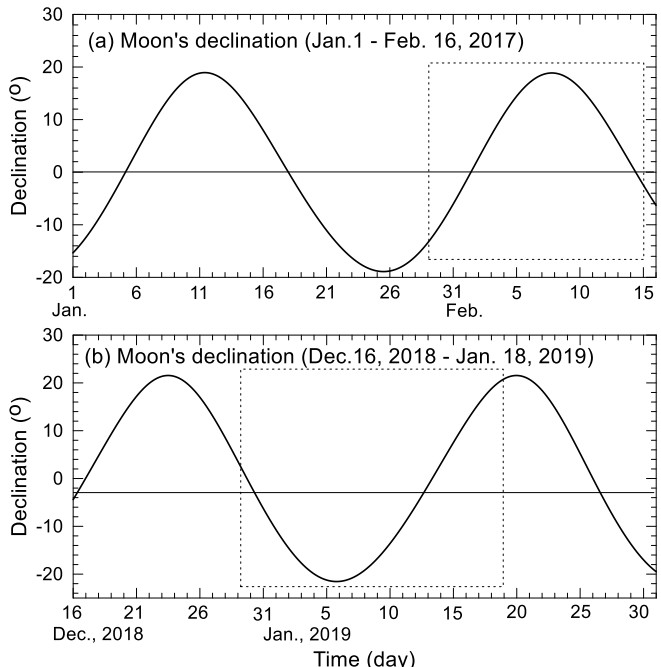


**Figure 9. Time series of the Moon's declination, estimated at daily intervals for two observation periods: (a) 1 January to 15 February**
**2017; and (b) 16 December 2018 to 30 January 2019. Dashed boxes indicate the sea level observation windows examined in this**
**study.**





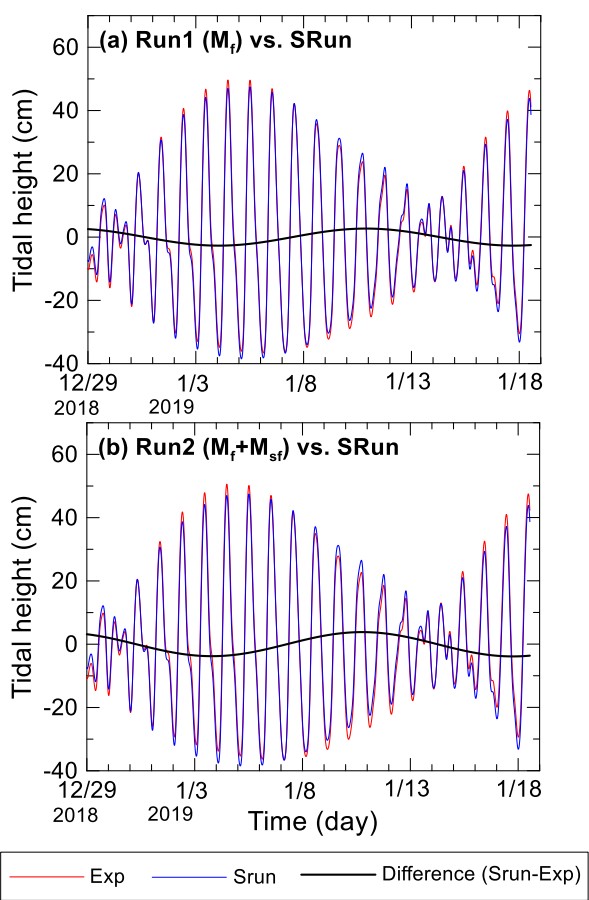


**Figure 10. Time series of ROBT tidal predictions (a) made without long-period constituents ('SRun', i.e. excluding the constituents**
**listed in Table 2) versus with the $M_f$ tide ('Exp1'); and (b) time series of ROBT tidal predictions made ('SRun') without the long-**
**period constituents versus ('Exp2') with the $M_{sf}$ tide. All predictions were generated based on tidal harmonic analysis results from**
**the yearlong (2013) ROBT sea level records.**


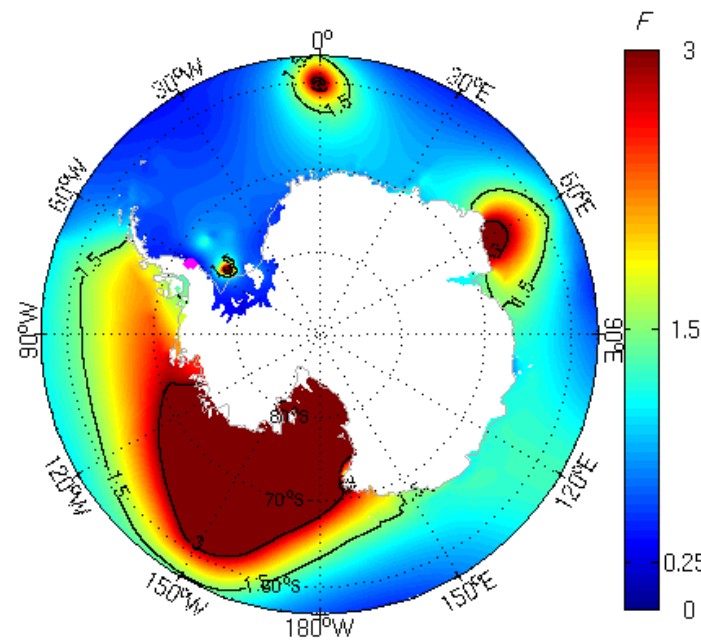


**Figure 11. Horizontal distribution of tidal form factor ($F$) values around Antarctica. Note the magenta area on the Antarctic Peninsula's Weddell Sea coast denotes the only area of semi-diurnal tides ($F<0.25$) in the Antarctic region.**





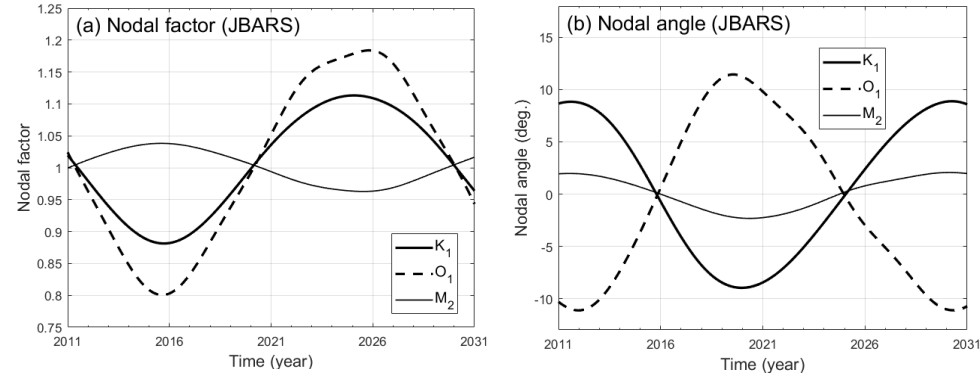

Figure 12. Variation in nodal factors and nodal angles for the three main lunar tidal constituents (K1, O1 and M2) over a 20 year period from 2011 to 2030 at Jang Bogo Antarctic Research Station (JBARS), estimated at daily intervals from the t_vuf.m program of T_Tide (Pawlowicz et al., 2002).