# Peer review of "Predicting tidal heights for extreme environments: From 25 hr"

_Ocean Science, 2019_

## Referee Comment (RC1) · Anonymous Referee #1 · 17 Jan 2020

This paper is a case-study demonstrating the use of a method termed "CTSM+TCC" for deriving tidal predictions from only 25h of observations and a good nearby tidal record, at a site in Antarctica. The method itself is similar to the Response Method (Munk & Cartwright, 1966) applied to neighbouring "standard stations", as described in Pugh & Woodworth 2014. (Chapter 4.3). It is not therefore particularly novel in principle, but the paper has merit as a very clear description of both method and results. It is also a useful reminder that Antarctic tides are important and short of data. I have a number of minor comments, but am happy to recommend publication in Ocean Science.

[Figure]

Minor comments:

p1, line35: Could you add these neighbouring sites to the map? And it would be good to find out what data is publicly available, and use them for further validation if possible.

p4, line22: thanks for mentioning atmospheric conditions, too often ignored.

p4, line148: you could mention somewhere here that bundling all the constituents in a species together is valid due to the "credo of smoothness" assumption.

p6, line206: In figure 6, it looks like the ADI is negative as the peak is before the max declination?

p7, line 251: (And elsewhere, please check all), Msf should be MSf [Moon-Sun-fortnight]. Similarly Msm should be MSm [Moon-Sun-month].

p7, line 270: Given MSf is important, I wonder if it might be worth including MS4? It might mop up the high frequency residual in figure 8. Worth checking the amplitude in the long record.

p8, line 302: So the tides in the Ross Sea will be almost 1.5 times larger in 2025 than in 2016? I wonder how aware the ice modelling community are of this?

fig 6: Is the split y axis really necessary here?

Language:

I am particularly impressed by how clearly written this paper is - I thank the authors for making the reviewing task easy. I wish I wrote as well!

p1,line9: "Though" should be "However"

p7 line 246: -tropic ?

p8 line 275: The abreviations DD etc aren't used again, delete.

References:
P&W 2014: Pugh, D.T. and Woodworth, P.L. 2014. Sea-level science : understanding tides, surges tsunamis and mean sea-level changes. Cambridge University Press https://doi.org/10.1080/00107514.2015.1005682 M&C 1966: Tidal spectroscopy and prediction, Walter Heinrich Munk and David Edgar Cartwright https://doi.org/10.1098/rsta.1966.0024

Oh, and you need to add doi to some of your other references!

---

## Author Comment (AC1) · 3 Feb 2020

Format: We are very grateful for this review as it has been useful in helping to improve our paper. Below we have copied each individual reviewer comment, and written below it a response.

p1, line35: Could you add these neighbouring sites to the map? And it would be good to find out what data is publicly available, and use them for further validation if possible. Response: According to reviewer's comment, these sites have been added. Thank

[Figure]

you for the suggestion regards validation and other publically available records. Unfortunately it appears relatively difficult to find recent online records but we found mention of a 1 year record from McMurdo Station in a Padman et al. (2003) paper and of a tide gauge being set up at Mario Zucchelli Station (formerly named Terra Nova Station) from 1996 (see https://www.geoscience.scar.org/geodesy/perm_ob/tide/terranova.htm). We will indeed attempt to track down these and any other available Ross Sea records for a further paper on the tides of this very interesting area. We have added these references to our paper so that out authors can see the data sources behind our comment. Padman, L., Erofeeva, S. and Joughin, I.: Tides of the Ross Sea and Ross Ice Shelf cavity. Antarctic Science 15(1), 31-40, 2003.

p4, line22: thanks for mentioning atmospheric conditions, too often ignored. Response: Yes, agreed.

p4, line148: you could mention somewhere here that bundling all the constituents in a species together is valid due to the "credo of smoothness" assumption. Response: According to reviewer's comment, this has been added.

p6, line206: In figure 6, it looks like the ADI is negative as the peak is before the max declination? Response: Thank you for this query – upon checking, we found that location of symbols for Moon's maximum (âŰš) and zero declination (âŰă) was not correct. The Moon's maximum declination is 1900 7/2/2017 (18.867°) and the zero declination is around 0930 1/2/2017. We have now fixed these in Figure 6.

p7, line 251: (And elsewhere, please check all), Msf should be MSf [Moon-Sun-fortnight]. Similarly Msm should be MSm [Moon-Sun-month]. Response: Yes, these are now fixed throughout.

p7, line 270: Given MSf is important, I wonder if it might be worth including MS4? It might mop up the high frequency residual in figure 8. Worth checking the amplitude in the long record. Response: Thank you - we have now checked the MS4 amplitude from the one year (2013) harmonic analysis results of ROBT. The amplitude was 0.69

cm, indicating that the MS4 tide is not a major tidal constituent here.

p8, line 302: So the tides in the Ross Sea will be almost 1.5 times larger in 2025 than in 2016? I wonder how aware the ice modelling community are of this? Response: We have added some additional text to draw attention to the diurnal tide variation and this phenomenon as follows: "The resulting variations in tidal height are less pronounced in the semi-diurnal Weddell Sea, while the diurnal regime of the Ross Sea experiences large tidal range variations across 18.61 y cycles due to the influence of diurnal nodal factor variation, which is greater than that of the semi-diurnal M2 tide (e.g. compare nodal factors between 2016 and 2025 in Figure 12a). Of note, variations in the nodal factors of the O1 and K1 tides are out of phase with that of the M2 tide. Variations in the nodal angle of the K1 tide is in phase with that of the M2 tide but out of phase with that of the O1 tide (Figure 12b). Our results clearly indicate that such spatial and temporal tidal variation processes should be represented in studies of tide-coupled ice-ocean models for Antarctic waters.".

fig 6: Is the split y axis really necessary here? Response: We originally thought to employ a split y-axis scale in order to show as clearly as possible (magnify) the difference in RMSE results between Fig. 6(a) and Fig. 7. However, the effect of the split was a minor one, so we have changed these axes in line with your comment as it was not fully necessary.

Language: I am particularly impressed by how clearly written this paper is - I thank the authors for making the reviewing task easy. I wish I wrote as well!

p1,line9: "Though" should be "However" Response: This has been changed according to this comment.

p7 line 246: -tropic ? Response: The misplaced hyphen before 'tropic' has been removed.

p8 line 275: The abreviations DD etc aren't used again, delete. Response: Yes, these

have been deleted.

References: P&W 2014: Pugh, D.T. and Woodworth, P.L. 2014. Sea-level science : understanding tides, surges tsunamis and mean sea-level changes. Cambridge University Press https://doi.org/10.1080/00107514.2015.1005682 M&C 1966: Tidal spectroscopy and prediction, Walter Heinrich Munk and David Edgar Cartwright https://doi.org/10.1098/rsta.1966.0024 Oh, and you need to add doi to some of your other references! Response: We have added the Pugh and Woodworth reference, and added the doi numbers where they were missing from existing references.

List of figures submitted with this comment:

Figure 2. Maps showing locations of (a) the two tidal observation stations in the Ross Sea of Antarctica employed in this study: Jang Bogo Antarctic Research Station (JBARS, âŰš) and Cape Roberts (ROBT, âŮŔ), and. (b) these study sites relative to McMurdo Station (âŰă), and Mario Zucchelli Station (âŮŔ).

Figure 6. (a) Time series of Root Mean Square Errors (RMSE, thick line with âŮŔ) and coefficients of determination (R2, thin line with âŮŃ) between JBARS 10 min interval sea level observations (29 January to 15 February 2017) and the CTSM+TCC prediction datasets generated for this site using harmonic analysis results from the daily (25 h) sea level data slices from JBARS plus concurrent daily (25 h) tidal prediction slices and harmonic analysis results from ROBT station's yearlong (2017) tidal predictions. (b) Time series of predicted 2017 tidal heights (thin line) and daily tidal ranges (thick line with âŮŃ) for ROBT, based on harmonic analysis of this station's 2013, 5 min interval sea level records, plus an indication of the moon's phase and declination.

Figure 7. Time series of Root Mean Square Errors (RMSE, thick line with âŮŔ) and coefficients of determination (R2, thin line with âŮŃ) between JBARS 10 min interval sea level observations (29 December 2018 to 18 January 2019) and the CTSM+TCC prediction datasets generated for this site using harmonic analysis results from daily (25 h) summertime 2017 sea level data slices from JBARS plus concurrent daily (25
h) tidal prediction slices and harmonic analysis results from ROBT station's yearlong (2017) tidal predictions.

Figure 8. Time series of JBARS sea level observations, predicted tidal heights, and sea level residuals (i.e. predictions minus observations) from (a) 29 January to 15 February 2017 and (b) 29 December 2018 to 18 January 2019. The JBARS predictions were generated using CSTM+TCC, with a daily (25 h) slice of local sea level observations from 8 February 2017 (dashed box in (a)), plus concurrent predictions and yearlong (2017), 5 min interval ROBT tidal predictions.

Figure 12. Variation in nodal factors and nodal angles for the three main lunar tidal constituents (K1, O1 and M2) over a 20 year period from 2011 to 2030 at Jang Bogo Antarctic Research Station (JBARS), estimated at daily intervals from the t_vuf.m program of T_Tide (Pawlowicz et al., 2002).

[Figure]

**Fig. 1.** Figure 2

[Figure]

**Fig. 2.** Figure 6

[Figure]

Figure: RMSE & R² (JBARS, 2019). RMSE (cm) on left axis, Coefficient of determination (R²) on right axis versus Time (day) of 2017 daily (25 h) input prediction dataset.

**Fig. 3.** Figure 7

[Figure]

**Fig. 4.** Figure 8

[Figure]

[Figure]

**(a) Nodal factor (JBARS)**

Nodal factor vs Time (year), 2011–2031, with curves $K_1$, $O_1$, $M_2$.

**Fig. 5.** Figure 12a

(b) Nodal angle (JBARS)

Nodal angle (deg.)

Time (year)

Legend: $K_1$, $O_1$, $M_2$

**Fig. 6.** Figure 12b

---

## Referee Comment (RC2) · Glen Rowe (Referee) · 14 Feb 2020

This paper aims to provide a method for making the most from very short period (25 hours) sea level observations by utilising data from a nearby permanent tide gauge site in Antarctica. Ideally the method used would employ simultaneously observed data at both sites, but this is not been possible here so it has been necessary to use predicted data at the permanent gauge. Nevertheless, a satisfactory result has been achieved which, in itself, is of some interest. This then begs the question about what might be achieved with actual observations. At the time of the short observations were made at

[Figure]

JBARS, the gauge at Scott Base was operational and this data could be used to fully evaluate the CTSM+TCC method.

There is some repetition in the paper and the explanation of some of the figures/tables may be able to be simplified when the figure/table is placed in the body of the paper. A couple of times the paper appears to wander away from the topic and describes the nature of the tide elsewhere in Antarctica which doesn't add anything to the purpose of the paper.

Please see my comments below which I hope will be of benefit to this paper which demonstrates a method to draw the most benefit from the sparse sea level observations that the Antarctic environment allows to be made, not without considerable difficulty. This paper encourages us to make the most of the few opportunities available and on that basis I support it's publication in Ocean Science.

Line 9: The words 'as represented' are unnecessary and at the start of the next sentence change Though to However

Line 20: This sentence could end at regimes as the following words repeat what has already been stated.

Line 29: . . .based on as little as 25 h of sea level records when combined. . . Also, h, as used here and elsewhere in the paper, would be clearer if abbreviated to hr (or better still, written in full).

Line 35: I'm not aware of the US operating a gauge in McMurdo Sound and would be interested to know where/when. NZ has a gauge at Scott Base. Does Italy have a long-term gauge at MZS?

Line 36: Only the Italian base is in Terra Nova Bay – the others aren't anywhere near this bay.

Line 37: There is also the problem of securing against damage any cable connection from a subsurface device to datalogging/power equipment ashore.

Line 42: Of course, hydrographic surveys are ideally carried out when there is minimal sea ice; whether or not there is a permanent gauge site (line 40-41) is not the main factor when deciding when to conduct such surveys.

Line 72: . . . in the austral summertime . . .

Line 81: Residuals – observed compared to predicted?

Line 83: . . . the absence of a permanent tide station at JBARS, . . .

Line 94: Pairs in brackets unnecessary repetition from lines 92 and 93.

Lines 96 – 98: As Table 1 will be inserted here this sentence is redundant as it is just repeating what the table contains.

Line 100: . . . phase lags showed only slightly different values.

Line 101: for completeness, should the formula for F be stated?

Lines 103 – 111: Is this paragraph necessary? This study relates to a part of the Ross Sea – the tidal regimes around other parts of Antarctica are of no relevance to this investigation. Or maybe you are hinting that as the Ross Sea is different to the rest of the continent the results of this study may not be applicable elsewhere. If this paragraph is deleted then Figures A1 and A2 are no longer required.

Line 113: Delete the '-' in front of CTSM.

Lines 114 – 115: Are the italics necessary?

Line 116: Similar tidal characteristics at the reference and temporary site is given as one of the requirements of the CTSM+TCC method. However, it has been noted in lines 101 – 102 that ROBT is diurnal and JBARS is mixed, mainly diurnal. Are these regimes sufficiently alike to be considered 'similar' for the purposes of this method?

Lines 121 – 122: The records are not temporary – the records are from a temporary site.

Line 124: My record from ROBT does not have any gaps early February 2017.

Lines 127 – 129: This sentence reiterates the essence of the preceding sentence and, although it begins 'In short', is longer than the previous one. One of these two sentences could be deleted.

Lines 148 – 154: Is the first sentence in this block of lines necessary? The following two sentences describe the process and can stand on their own.

Line 154: Which are the 'initial tidal predictions'? It is not clear to me.

Line 163: Calculations, not experiments?

Line 164: 'in shorthand' seems unnecessary.

Lines 169 - 171: I had to read the first part of this sentence a few times to figure out what is going on. My take is that you obtained 17 datasets each one of which included 10-minute interval predictions spanning 17 days as derived from the harmonic analysis of each of the (17 in total) 25 hr slices of observed data. Is this correct? If not then I have clearly misunderstood, and if it is then that is good but, regardless, I'm not confident that I have it right.

Lines 177 – 187: This discussion about the correlation of tidal range and RMSEs and $R^2$ values is more difficult to follow than it could be. I feel the two sentences about the February 2 tide 'sandwiched' between the discussions about the results at greater tidal ranges has made the explanation somewhat convoluted. Dealing with the circumstances of the good statistics before moving on to the poorer results will enable this discussion to be expressed in a more succinct manner (and easier to follow).

Lines 188 – 192: Are these two sentences saying the same thing in different ways?

Lines 208, 209, 211, 212 and 213: I find the use of the adjectives 'maximum' and 'minimum' in association with declination to be confusing. Minimum could be taken to be on the celestial equator ($\delta = 0°$) and maximum could be greatest declination either

north or south. Better to use phrases like 'greatest southern declination' and 'greatest northern declination' to be more specific.

Line 227: Delete 'and'.

Lines 245 – 246: It would be helpful to give the dates for the two periods (ETT and TET). Is the 'minus' in front of tropic on line 246 a typo or does it mean the southern-most declination?

Line 247: . . . CTSM+TCC considering only 2 major tidal species . . .

Lines 249 – 256: Could this be shortened to just summarise the conclusion arrived at by the other authors. Is there a need to describe what they did – people interested can refer to the references.

Lines 267 - 268: Srun excluded . . . Run1 excluded . . . Run2 incorporated . . . (I think)

Lines 269 and 270: Should both instances if 'exclusion' be 'inclusion'?

Lines 270 – 271: Is there any reason why this suggested line of investigation has not been pursued in this paper?

Line 273: Section 5.2 does not seem to contribute to the main aim of the paper, i.e. to predict tides from 25 hr observations. 5.2 looks at the contrasting tidal environments of two areas and tries to explain why they differ. I think 5.2 could be removed.

Figures A1 and A2: If the paragraph at lines 103 -111 is deleted then these figures are no longer required.

Figure 2: Readers might find this more informative if the map covered the Ross Sea only.

Figure 3: The x-axis label should be 'Time (month/day)'. The description starts 'Seventeen day time series. . .". Isn't it seventeen sets of daily (25 hr) data slices as stated in line 130?

Figure 4: The x-axis for all four plots needs a label (Time (month/day)". The description refers to daily slices of the 17 day ROBT tidal predictions in the first sentence, but the second sentence refers to results of the 369 day 2013 ROBT analysis. Is this correct?

Figure 5: Both plots need a label for the x-axis. The description refers to dashed lines in the plots but these are not shown.

Figure 6: X-axis label (for both plots) should read Time (month/day). In the key, should the Moon's maximum declination be qualified as being either north or south? Line 429: the symbol (open circle) does not match the plot.

Figure 7: X-axis label could be consistent with the other figures. Line 435 has the word 'plus' – this makes me confused about the description. My take is that the plot compares predictions for day x of 2019 (as derived from analysis of data from day y of 2017) with observations made on day x of 2019. Have I got this correct?

Figure 8: X-axis labels again. As with Figure 7, I am confused by the statement that follows '(dashed box in (a)),'.

Figure 9: X-axis labelling differs from all other figures – could be altered for consistency. Line 444: should 'estimated' be 'calculated'?

Figure 10: X-axis labels again.

Figure 11: If Section 5.2 is deleted then these figures are no longer required. If retained then the word 'Horizontal' in the description is redundant. Is the area in the Weddell Sea coloured magenta?

Table 2: I would delete the Period and Angular speed columns. Not only are the amplitudes of most constituents in this table small, but by my analysis they also have small signal-to-noise ratios so are weakly determined. This caution about the reliability of these values should be noted.

Table 3: My records from ROBT for 2011 commence 21 November so the values given

for that year can't come from yearlong observations (as is the case for the others in the table).

---

## Editor Comment (EC1) · Philip Woodworth (Editor) · 14 Feb 2020

**14 February 2020**

Comments on 'Predicting tidal heights for extreme environments: From 25 h observations to accurate predictions at Jang Bogo Antarctic Research Station, Ross Sea, Antarctica' by Byun and Hart (OSD)

Here are some comments from me as editor additional to those of the two reviewers. My comments seem to be closer to those of Reviewer 2 than Reviewer 1. It will be best

if all three sets of comments are taken together for any new version (see below).

In the following I give a list of comments on the writing (there are several sentences without verbs, for example). But the main thing is that I thought there were 3 sections that either need considerable improvement or should be dropped.

(1) Section 3. I understand that the method is some kind of response method, and I have read the authors' 2015 paper. However, I defy anyone to understand this section as it stands. It is made worse by not defining many variables (e.g. line 128, what are r, eta and tau. I believe s is species; line 136, what are k,m etc.). And I am sure there must be errors in equation 2 although I am not sure what e.g. it has a parameter j which is a subscript of a constituent like 'i', but which is not summed over but used only as a lower limit i=(j,m), but the left side of the equation is not a function of j. That cannot be right. Then also, what is a 'representative harmonic constituent'?

I think a simpler thing to have done would have not included the little bits of maths here which just confuse everyone but just referred the reader to the 2015 paper for the method. I have many detailed comments on this section also which I list below.

(2) Section 5.1. You have records of the order of a fortnight so I daresay it is inevitable that there will be mismatches on that timescale between your method and the data. However, do you need a page to say that? I suggest that this aspect should be summarised in 5-6 lines in the Discussion section where it can be a pointer to improvements in the method. Also I wondered if you considered the missing fortnightly tide was consistent with that in FES2014.

(3) Section 5.2. Having shown that the Ross and Weddell Seas have different dominant tides (and form factors), end of story to me, you embark on generating predictions over 18.6 years which lo and behold have the ranges (you don't explain range = 2\*amplitude) which have exactly the equilibrium amounts that T-Tide must be coded with. So what have you learned? Nothing. The finding is presented as some kind of new result. I suggest, having indicated the map of form factors, you just say that because diurnal
tides have larger 'f' and 'u' variations than semidiurnal tides (reference a text book) then they will have larger ranges of tide over 18.6 years. In 5-6 lines again.

Also, on line 67, you say that section 5 will discuss double tidal peaks. I can't see anything about that in the section or the paper.

Because of the problems with these 3 sections, which make up most of the paper, I expect that it will not be acceptable for OS without considerable improvements. Anyway, I am unclear what has been learned new here which you didn't learn from NZ and Korea data in the 2015 paper - I realise this is a different tidal regime but a step change might have been to write a larger paper with as many regimes as possible if you wanted to demonstrate that your method works well.

Detailed comments:

9-10 - sentence 'Though obtaining'. This sentence has no verb.

13 - by a different tidal regime

18 - I have never seen this 'tropic-spring' description before (there are other examples below). Could you not just replace this simply with 'at high lunar declination', or whatever, which means something physical rather than poetic.

28 - the Rignot reference is rather old. There has been a lot of work using GPS for tides under ice sheets, and there new data sets (IceSAT etc.). I am sure you can find a couple of better references.

- 46 though -> although
- 50 transfers -> transfer
- 59 GPS is usually called GNSS these days
- 67 see above
- 75 what does high-frequency mean?
89 - year-long

92 - this needs rewording. the 2 main diurnal and semidiurnal tides are K1 and O1 and M2 and S2 of course - what you mean here are the 2 main relationships taken from Cape Roberts

97 - they have similar amplitudes. not 'characterised by'

97 - between -> at

98 - for S2 respectively.

99 - close -> short

But I don't consider 269 km a short distance. I am sure the tide around Korea or NZ, for example, changes enormously in that distance. And what does 'in tidal terms' mean?

- 100 phase lag usually has no hyphen
- 101 what does tidal patterns mean? You mean tidal characteristics?
- 104 database -> model

105 - drop horizontal

105-111 - there are amplitudes and phase lags, and there are co-amplitude (or sometimes co-range) and co-phase charts, sometimes combined as co-tidal charts. But there is no such thing as an 'increasing co-amplitude'. Please rewrite this paragraph. See below for the figures also.

113 - why a minus before CTSM?

- 114-115 why the italics?
- 125 remove simply
- 126 remove accurate. You have no way of showing how accurate they are.
- 127 remove sentence 'In short'. This is obvious.
- 128 see above. Also mention that phase lags are Greenwich lags.
- 130 sentence 'Note that'. Again I think that assumes you understand the method
- 134 peculiarities -> properties?

169 - again, I guess the reader will have to read the 2015 paper to understand why you produce 17 data sets? This has to be clearer.

- 175 where -> when
- 227 versus -> and

day -> days

add respectively at end of sentence

229-230 - sentence 'Hence the' has no verb

246 - remove the minus sign. Replace the tropic jargon business.

249-256 - I think I would replace this woffle with simply saying that good knowledge of tides is important for understanding ice shelf dynamics and give one reference as an example.

- 265 .. periods, rather than seasonal. (I think)
- 270 with additional exclusion (I think)

272 - well, you don't do that do you?! You have spent a page showing that the method could be improved with a digression into the ice shelves. There is very little in this section (see above also).

273 - Decadal timescale ..

274 - drop daily
276 - I don't understand this. The small magenta blob on the west coast of the Weddell Sea indicates a large (diurnal) form factor, right? Not semidiurnal. (You might also mention its latitude rather than 'half-way'). Most of the Wedddell Sea is blue (semidiurnal).

278 - you could point to Fig 1 for mention of Prydz Bay

279 - drop 'the increase in'

286 - drop 'feature ...tidal' which is just repetition. influences -> influence

292-298 - see above. This is just an inevitable consequence of the way T-Tide is coded with the equilibrium nodal dependencies.

- 302 Drop 'Of note', unless you want to refer to a tidal text book
- 328 drop database
- Fig A1 caption drop horizontal.
- co-amplitudes -> amplitudes. co-tides -> phase lags (Greenwich)

In the caption of the 4 figures, remove the dot after deg as there is no dot after cm. remove all the co- things. And co-tide should be Greenwich phase lag.

Figure A2 ditto the above. In (b) and (d) there is a mess of annotation of phase lags at a couple of amphidromic places. Please remove that mess.

Table 1. Please move the information in the Note column to be extra lines under ROBT etc. You give only one set of ADI and AT for JBARS but there must be two different sets of values in 2017 and 2019.

day -> days. No hyphen in phase lag.

Table 2. .. harmonic analysis of year-long .. No hyphen in phase lag

Figure 1 caption. Please say year and month this photo was taken
Figure 3 - y-axis phase lag should be (deg) and not (cm)

Figure 4 caption should say what (a), (b) etc. are and not just have text. Anyway I think the last two sentences contradict each other

Figure 5 - under (b) you should have Time (month/day) as for Figure 6

I think the last line should say JBARS and ROBT

Figure 6 - Time (day) should be Time (month/day). (a) and (b) are missing from the plots.

Line 429 - (thick line with o) should have a filled and not open o to correspond to the plot

Figure 7 - why the == on the y-axis? There is no break in the numeration. Time (day) should be Time (month/day)

Figure 8 - Time(month/day).

A difference like this is usually defined as an Obs minus Pred but I guess it doesn't matter too much.

The caption says 15 February, but the x-axis in (a) only goes up to 14 Feb. The caption should say what RMSE and R-squared are.

Figure 9 - the caption and the x-axis in (a) say 15 Feb, but the header says 16 Feb In (b), the caption and x-axis say 30 Jan but the header says Jan 18. I thought at first you were referring to the dates of the dashed boxes but it seems not.

line 1 of caption - estimated -> shown

Figure 10 - Time (month/day)

450 - Msf and Mf tides ('Exp2'). At least I think that is what is meant.

Figure 11 - please have an arrow on the colour scale to indicate values over 3. The

OSD
longitudes on the map are fuzzy.

caption - drop horizontal.

Figure 12. What you are showing here are the 'f' and 'u' nodal factors. They are both nodal factors, not just 'f'. They are not 'estimated', they are hard coded into T-Tide and can be found in any tides text book.

So you can tell I found many small problems with the paper, in addition to the problems with the three sections mentioned above. I hope you can produce a considerable better (and probably shorter) version.

---

## Author Comment (AC2) · 19 May 2020

Reply to interactive comment of 14 Feb 2020 on "Predicting tidal heights for extreme environments: From 25 h observations to accurate predictions at Jang Bogo Antarctic Research Station, Ross Sea, Antarctica" by Glen Rowe

Do-Seong Byun1, Deirdre E. Hart2 1Ocean Research Division, Korea Hydrographic and Oceanographic Agency, Busan 49111, Republic of Korea 2School of Earth and Environment, University of Canterbury, Christchurch 8140, Aotearoa New Zealand Correspondence to: Deirdre E. Hart (deirdre.hart@canterbury.ac.nz)

Format This review has been useful in helping to improve our paper. Below we have copied each individual reviewer comment, and written below it a response, and then pasted the text of the paper as it now reads.

Individual reviewer comments and our responses

Line 9: The words 'as represented' are unnecessary and at the start of the next sentence change Though to However Response: Both of these wording changes have been made exactly as suggested. The paper now reads: "Accurate tidal height data for the seas around Antarctica are much needed, given the crucial role of tidal processes in regional and global climate, ocean and marine cryosphere models. However obtaining long term sea level records for traditional tidal predictions is extremely difficult around ice affected coasts".

Line 20: This sentence could end at regimes as the following words repeat what has already been stated. Response: Upon reflection we decided that this sentence was not needed, since the previous sentence detailed the level of success of the method, so this former line 20 sentence has been deleted. The preceding sentence now reads: "Results reveal the CTSM+TCC method can produce accurate (to within ∼5 cm Root Mean Square Errors) tidal predictions for JBARS when using short-term (25 hr) tidal data from periods with higher than average tidal ranges (i.e. those at high lunar declinations and/or spring periods)".

Line 29: : : :based on as little as 25 h of sea level records when combined: : : Also, h, as used here and elsewhere in the paper, would be clearer if abbreviated to hr (or better still, written in full). Response: Regarding the second point above, the unit for hour, 'h' has been changed to 'hr' throughout the manuscript. Regarding the first point, the text has been altered as suggested (see below). The paper now reads: "However, Byun and Hart (2015) developed a new approach to successfully predict tidal heights based on as little as 25 hr of sea level records when combined

neighbouring reference site records, using their Complete Tidal Species Modulation with Tidal Constant Corrections (CTSM+TCC) method, on the coasts of Korea and New Zealand".

Line 35: I'm not aware of the US operating a gauge in McMurdo Sound and would be interested to know where/when. NZ has a gauge at Scott Base. Does Italy have a long-term gauge at MZS? Response: Padman et al. (2003) mentions a 1 year record from McMurdo Station. Also a tide gauge was set up at Mario Zucchelli Station (formerly named Terra Nova Station) from 1996 (see https://www.geoscience.scar.org/geodesy/perm_ob/tide/terranova.htm). We are currently attempting to track down these and any other available Ross Sea records for a further paper on the tides of this very interesting area. We have added these references to our paper so that our readers can clearly see the data sources behind our comment. The paper now reads (and includes the references below): "Long-term, quality sea level records in the Ross Sea are few and far between, and include observations from gauges operated by New Zealand at Cape Roberts (ROBT); by the United States in McMurdo Sound (see reference to this data in Padman et al., 2003); and by Italy at Mario Zucchelli Station (see: Gandolfi, 1996), all in the eastern Ross Sea". The reference list now includes: Gandolfi, S.: Terra Nova Bay Permanent Tide Gauge Observatory Site, https://www.geoscience.scar.org/geodesy/perm_ob/tide/terranova.htm, last access 4 Feb. 2020, 1996. Padman, L., Erofeeva, S. and Joughin, I.: Tides of the Ross Sea and Ross Ice Shelf cavity. Antarctic Science 15(1), 31-40, 2003

Line 36: Only the Italian base is in Terra Nova Bay – the others aren't anywhere near this bay. Response: Thank you – this error has now been corrected to 'eastern Ross Sea' (see full revised sentence above in response to comment on line 35).

Line 37: There is also the problem of securing against damage any cable connection from a subsurface device to datalogging/power equipment ashore. Response: Yes, though this is a challenge for any cabled shoreline instrument deployed for a long time in any coastal environment, we can imagine that it is particularly difficult in the harsh

environment of Antarctica. We have added this issue to the text. The paper now reads: "There is also the challenge of securing, and preventing damage to, the cables that join the subsurface instruments to their onshore data loggers and power supplies, across the seasonally dynamic and harsh coastal and subaerial environments of Antarctic shorelines".

Line 42: Of course, hydrographic surveys are ideally carried out when there is minimal sea ice; whether or not there is a permanent gauge site (line 40-41) is not the main factor when deciding when to conduct such surveys. Response: Yes, agreed – in order to better separate out these two pieces of information we have split one sentence into two. The paper now reads: "In the absence of a suitable permanent gauge site, hydrographic surveys have been conducted at the Korean Jang Bogo Antarctic Research Station (JBARS). Such surveys are best conducted during the summertime predominantly sea ice free window around mid-January to mid-February".

Line 72: : : : in the austral summertime : : : Response: Yes, the word austral has been added here as well as in another place in the paper. The paper now reads: "The Korea Hydrographic and Oceanographic Agency (KHOA) survey team went to JBARS in Northern Victoria Land's Terra Nova Bay, Ross Sea, Antarctica, in the austral summertime of 2017 (Fig. 2) for a preliminary fieldtrip to conduct hydrographic surveys and produce a nautical chart".

Line 81: Residuals – observed compared to predicted? Response: Yes, that's correct. We added the text in brackets below. The paper now reads: "Of these, the 20.54 day record produced between 29 December 2018 and 18 January 2019 comprised relatively high quality data with small residuals (i.e. observed minus predicted)".

Line 83: : : : the absence of a permanent tide station at JBARS, : : : Response: The text has been altered as suggested. The text now reads: "Due to the short duration of the KHOA survey team's 2017 and 2018 to 2019 forays into the Ross Sea, and in the absence of a permanent tide station at JBARS, it was not possible to collect

the year-long sea level records that are commonly employed to obtain reliable tidal constituents".

Line 94: Pairs in brackets unnecessary repetition from lines 92 and 93. Response: Your comment alerted up to the wordy nature of these two sentences, so instead of just deleting the pairs in brackets we rewrote both sentences as one replacement sentence, shortening our explanation of this step while retaining the key details. The text now reads: "The inference method was used to separate out neighbouring diurnal (K1 and P1) and semidiurnal (S2 and K2) tide constituents, with their amplitude ratios and phase lag differences obtained from harmonic analysis of the long-term ROBT reference station records".

Lines 96 – 98: As Table 1 will be inserted here this sentence is redundant as it is just repeating what the table contains. Line 100: : : : phase lags showed only slightly different values. Response: With regard to your line 96-98 comment, we deleted the text that unnecessarily highlighted the numbers displayed in Table 1, and in just kept the interpretive text found it best to merge two sentences together for tighter expression of the results. According to your line 100 comment we removed the hyphen from 'phase lag' throughout the paper – the below sentence provides an example. The text now reads: "Analysis revealed that the two main diurnal (O1 and K1) and semidiurnal (M2 and S2) tides had similar amplitudes at the two stations (Table 1), with the diurnal amplitudes being slightly larger at ROBT than at JBARS, the semidiurnal amplitudes being slightly smaller at ROBT than at JBARS, and the phase lags of all four tides having only slightly different values".

Line 101: for completeness, should the formula for F be stated? Response: Yes, agreed – we have now added explanation of this parameter to Table 1 caption, where it is now mentioned first in the paper. The Table 1 caption note now reads: "F is the ratio of the K1 and O1 diurnal tide amplitudes to the M2 and S2 semidiurnal tide amplitudes".

Lines 103 – 111: Is this paragraph necessary? This study relates to a part of the

Ross Sea – the tidal regimes around other parts of Antarctica are of no relevance to this investigation. Or maybe you are hinting that as the Ross Sea is different to the rest of the continent the results of this study may not be applicable elsewhere. If this paragraph is deleted then Figures A1 and A2 are no longer required. Response: According to your suggestion and comments by the Editor regards these lines, the whole paragraph has been deleted, as have the former Appendix 1 figures.

Line 113: Delete the '-' in front of CTSM. Response: This typo has been deleted, and the sentence has been altered significantly as a result of the Editor's suggestion that section 3 should be rewritten to describe the methodology more simply, removing much of the math. The text now reads: "Having analysed the tidal harmonic constants at the two stations based on their concurrent short-term records, we then employed the CTSM+TCC method (Byun and Hart, 2015) to generate tidal height predictions for JBARS, our 'temporary' tidal observation station (subscript o), using ROBT as the 'reference' station (subscript r).".

Lines 114 – 115: Are the italics necessary? Response: No they were unnecessary so have been removed in accordance with your comment. This sentence has been modified as a result of the section 3 rewrite. The text now reads: "This prediction approach (see Appendix 1 for the detailed calculations, and Byun and Hart (2015) for explanation of procedure development) is based on: using long-term ($\geq$183 days) reference station records (LHr) and CTSM calculations to make an initial anytime ($\tau$) tidal prediction ($\eta\_r$ ($\tau$)), which involves summing tidal species' heights for the reference station (Fig.3); and comparing the tidal harmonic constants (amplitude ratios and phase lag differences) of representative tidal constituents (e.g., M2 and K1) for each tidal species between the temporary and reference stations, calculated using T\_TIDE and concurrent short-term records ($\geq$25 hr duration, starting at midnight) from the temporary (SHo) and reference (SHr) stations; and using the step (ii) comparative data and the TCC calculations for each tidal species to adjust the $\eta\_r$ ($\tau$) tidal species' heights in order to generate accurate, anytime tidal height predictions for the temporary tidal station ($\eta\_o$ ($\tau$))".

Interactive
comment

Line 116: Similar tidal characteristics at the reference and temporary site is given as one of the requirements of the CTSM+TCC method. However, it has been noted in lines 101 – 102 that ROBT is diurnal and JBARS is mixed, mainly diurnal. Are these regimes sufficiently alike to be considered 'similar' for the purposes of this method? Response: Thank you, due to your question we have improved the text to really hone in on the similarity required. The text now reads: "Importantly, this method assumes that the reference and temporary tidal stations are situated in neighbouring regimes with similar dominant tidal constituent and tidal species characteristics, and that the tidal properties between the two stations remain similar through time. As explained above, both JBARS and ROBT have tidal regimes that are primarily dominated by diurnal tides. LHr must comprise high quality (e.g. few missing data) tidal height observations from anytime".

Lines 121 – 122: The records are not temporary – the records are from a temporary site. Response: Yes, thank you. We have made sure that this word placement mistake does not now occur in our paper. This particular sentence has also been deleted as part of the Section 3 re-write, recommended by the Editor.

Line 124: My record from ROBT does not have any gaps early February 2017. Response: ROBT data were downloaded from LINZ website. There are still no data files until 12 February 2017 as you can see at http://apps.linz.govt.nz/ftp/sea_level_data/ROBT/2017/00/ (last access: 29 February, 2020). We have, however, now received a file containing the full 2017 records, after finding out that they existed when consulting you with regards to the ROBT set up by telephone – thank you very much for supplying these excellent data. Please note that these data are not available on the Permanent Service for Mean Sea Level (PSMSL) website, where ROBT records are recorded as existing up until 2009. We have found finding the existence of, and then obtaining, good observational tidal data for the Ross Sea and elsewhere in Antarctic quite a challenging exercise. Since your LINZ records represent one of the best in existence, it might benefit Antarctic tide research to update the PSMSL website: https://www.psmsl.org/data/obtaining/stations/1763.php, including the comments made there on the low data quality of recent ROBT records. Currently this website says: "Documentation added 2011-11-17. There is no data available for 2010. Although the site is still working the data is of low quality and therefore unreliable. Plans are in place to repair the tide gauge when possible". We have re-written this sentence as a part of our Section 3 re-write. The text now reads: "This slight adjustment in approach arose since for the 2017 JBARS observation time period, the concurrent 2017 ROBT records available online (LINZ, 2019) had multiple missing data".

Lines 127 – 129: This sentence reiterates the essence of the preceding sentence and, although it begins 'In short', is longer than the previous one. One of these two sentences could be deleted. Response: In our re-write of Section 3 we deleted the last of these two sentences as suggested here. The remaining sentence reads: "We solved this issue by producing a year-long synthetic 2017 record for ROBT using T_TIDE (Pawlowicz et al., 2002) and the 2013 (i.e. LHr) observational record as input data".

Lines 148 – 154: Is the first sentence in this block of lines necessary? The following two sentences describe the process and can stand on their own. Response: This section of text has now been cut and pasted into an appendix detailing the math behind the CTSM+TCC approach (in response to a suggestion by the Editor to rewrite Section 3 more clearly and simply). In its new Appendix 1 location, the first sentence in this block has been modified to convey different/ extra information according to a comment by Reviewer 1, and terms that were repeated in the next two sentences have been deleted, eliminating overlap that you drew our attention to. The text now reads: "As the second step, under the 'credo of smoothness' assumption that the admittance or 'ratio of output to input' does not change significantly between constituents of the same species (Munk and Cartwright, 1966; Pugh and Woodworth, 2014), the amplitude ratio and phase lag difference of each representative tidal constituent for each tidal species between the temporary and reference stations were calculated from the results of tidal harmonic

analyses of concurrent 25 hr data slices (starting at 00.00) from the temporary tidal observation and reference tidal stations (i.e. from SHo and SHr). The process of selecting the optimal 25 hr window for these concurrent data slices from amongst the 17.04 days of available records is explained in Section 3 of this paper".

Line 154: Which are the 'initial tidal predictions'? It is not clear to me. Response: Thank you – this was not as clear as it could be: we meant 'tidal predictions at the reference station' calculated from the CTSM, and have improved the text accordingly. The text (cut and pasted into in Appendix 1) now reads: "Once the optimal 2017 short-term data window was selected, the third step involved adjusting the tidal predictions at the reference station calculated from Eq. (A1), to represent those for the temporary station ($\eta\_o$ $(\tau)$), by substituting the daily (i.e. SHo and SHr) amplitude ratios (($a\_o\hat{}((s)))/(a\_r\hat{}((s))$)) and phase lag differences ($G\_o\hat{}((s))$-$G\_r\hat{}((s))$ ) for the tidal constituents (K1 and M2) representing the diurnal and semidiurnal tidal species between the temporary and reference stations into Eq. (A1) as follows. . .".

Line 163: Calculations, not experiments? Line 164: 'in shorthand' seems unnecessary. Response: Yes to both – 'experiments' has been removed in the re-write of this text and we now describe these as 'prediction data sets', as opposed to experiments, at the end of the revised section 3. We also removed the 'in shorthand' text. The text now reads: "Each paired data set was then used with LHr to generate tidal height predictions for JBARS covering both the 2017 and 2019 KHOA temporary observation campaign time periods. Comparisons were made between the JBARS observations and the 17 prediction data sets generated for each observation campaign to identify which 25 hr short-term data window produces optimal $\eta\_o$ $(\tau)$ results".

Lines 169 - 171: I had to read the first part of this sentence a few times to figure out what is going on. My take is that you obtained 17 datasets each one of which included 10-minute interval predictions spanning 17 days as derived from the harmonic analysis of each of the (17 in total) 25 hr slices of observed data. Is this correct? If not then I have clearly misunderstood, and if it is then that is good but, regardless, I'm not

confident that I have it right. Response: Yes, that is correct and thank you for pointing out the difficulty of this sentence. This sentence has been re-written. Moreover the previous Section 3 description of the method applied has been improved significantly such that we anticipate readers will be much clearer by the time they reach section 4 about what we mean here. The text now reads: CTSM+TCC was used to produce 17 different JBARS tidal prediction datasets for the period 29 January to 14 February 2017, based on harmonic analysis results of their 'daily' (25 hr) K1 and M2 amplitudes and phase lags between our two tidal observation stations (Fig. 4)".

Lines 177 – 187: This discussion about the correlation of tidal range and RMSEs and R2 values is more difficult to follow than it could be. I feel the two sentences about the February 2 tide 'sandwiched' between the discussions about the results at greater tidal ranges has made the explanation somewhat convoluted. Dealing with the circumstances of the good statistics before moving on to the poorer results will enable this discussion to be expressed in a more succinct manner (and easier to follow). Response: Yes, agreed. We have reordered the text according to your helpful comment here. The text now reads: "RMSEs between observations and predictions ranged from 4.26 cm to 20.56 cm while R2 varied from 0 to 0.94 across the 17 'daily' experiments. Eleven of the experiments produced accurate results (i.e. excluding those based on 25 hr input data derived from 31 January; and 1 to 4 and 14 February records). Daily datasets from periods with relatively high tidal ranges (>83.5 cm) produced predictions with RMSEs <5 cm and R2 values >0.92. The maximum spring tidal range occurred on 9 February: the short-term data from this date produced predictions with a low (but not the lowest) RMSE (4.81 cm). The predictions with the lowest RMSE (4.259 cm) and highest R2 value (0.941) were produced using inputs derived from 25 hr data recorded one day earlier, on 8 February 2017. In contrast to the majority of successful experiments, the experiment based on data derived from the '2 February' 25 hr data slices produced predictions with very high RMSE (20.56 cm) and very low R2 (0.00) values. Notably, the 2 February tides were characterised by the smallest tidal range (11.95 cm) of the JBARS record, during a period of low lunar declination".
Lines 188 – 192: Are these two sentences saying the same thing in different ways? Response: They concerns the same idea, but the second sentence details the idea for a specific example case (Fig. 7) amongst the total 17 cases (Fig. 6). We have added "For example" to indicate this. The text now reads: "As with the 2017 predictions, RMSEs between the 2019 summertime predictions and observations were lower when generated using input data derived from 25 hr data slices from the 2017 spring tide periods at high lunar declination (as opposed to during neap tides and/or periods at low lunar declination) (Fig.6). For example, as in the earlier experiments, the 2019 summertime predictions made using input data derived from the 8 February 2017 (25 hr) data slices produced the lowest RMSE (5.3 cm) and highest R2 (0.913) values of the 2019 summertime experiments (Fig. 7)".

Lines 208, 209, 211, 212 and 213: I find the use of the adjectives 'maximum' and 'minimum' in association with declination to be confusing. Minimum could be taken to be on the celestial equator ($\delta = 0°$) and maximum could be greatest declination either north or south. Better to use phrases like 'greatest southern declination' and 'greatest northern declination' to be more specific. Response: Thank you for your useful suggestion – we have applied this change as recommended. The text now reads: "That is, maximum range tide days can be estimated for JBARS based on the day of the Moon's greatest southern (GS) and northern (GN) declinations. The time between the Moon's semi-monthly GN and GS declinations and their effects on tidal range, called the age of diurnal inequality (ADI), is commonly 1 to 2 days. As shown in Fig. 8, the GN and GS lunar declinations during our temporary station summertime observation periods occurred on 8 February 2017 (GN) and on 6 January 2019 (GS) respectively, with the maximum diurnal tides at JBARS expected approximately 1 day after each lunar declination peak".

Line 227: Delete 'and'. Response: Yes, this typo has been removed. The text now reads: "The ADI values were 0.57 and 0.23 or 0.30 days, while the AT values were -2.30 and -1.44 or -2.87 days, for ROBT and JBARS respectively (Table 1)".

Lines 245 – 246: It would be helpful to give the dates for the two periods (ETT and TET). Is the 'minus' in front of tropic on line 246 a typo or does it mean the southern-most declination? Line 247: : : : CTSM+TCC considering only 2 major tidal species : : : Response: In response to the suggestion from the Editor that we significantly shorten section 5.1 (he suggested 5-6 lines instead of 39 lines) we have deleted much of this detail (including mention of ETT and TET, and the sentence with the minus sign typo you mentioned). We also made the suggested '2 tidal species' change suggested above. The text now reads: "However, as shown in Fig. 7, our results contain a changing fortnightly timescale bias in estimates. This error pattern likely resulted from our application of CTSM+TCC considering only 2 major tidal species (diurnal and semidiurnal) whilst ignoring several long period tides".

Lines 249 – 256: Could this be shortened to just summarise the conclusion arrived at by the other authors. Is there a need to describe what they did – people interested can refer to the references. Response: Yes, we have removed most of the text explaining details and just left their findings that focus on what other constituents might be important. The remaining text has also been shifted slightly within the section. The text now reads: "Similarly, Rosier and Gudmundsson (2018) found that ice flows are modulated at various tidal frequencies, including that of the MSf tide. . . Nevertheless, studies indicate that incorporating major and minor tidal constituents, including long period tides, into tidal predictions may be advantageous for their use in ice flow and ice-ocean front modelling (e.g. Rignot et al., 2000; Rosier and Gudmundsson, 2018)".

Lines 267 - 268: Srun excluded : : : Run1 excluded : : : Run2 incorporated : : : (I think) Response: We have clarified this text as suggested. The text now reads: "Three 2019 tidal prediction experiments were conducted: Srun excluded all long-period tides (see list of exclusions in Table 2); Run1 was based on Srun but incorporated the Mf; and Run2 was based on Srun but incorporated the Mf and MSf".

Lines 269 and 270: Should both instances if 'exclusion' be 'inclusion'? Response: No, 'exclusion' is correct. We have reworded this part to avoid this confusion. The text now

reads: "Comparisons between Run1 and Srun predictions show that exclusion of the Mf tide (2.7 cm amplitude) can produce prediction biases during periods of lunar declination change (Fig. 9a), with comparisons between Run2 and Run1 results showing that the additional exclusion of the MSf tide (1.2 cm amplitude) intensifies the biases (Fig. 9b)".

Lines 270 – 271: Is there any reason why this suggested line of investigation has not been pursued in this paper? Response: Yes, basically, this is because the tidal constants for the long-period tides cannot be derived from short-term (25 hr) records, so it is beyond the scope of the present study, which was an initial assessment if the CTSM+TCC method could be used to generate predictions for JBARS, a temporary tidal station in an extreme environment with imperfect data record conditions. Now that we have demonstrated the usefulness of the method for making reasonable predictions here, we feel that further work could be done to hone the prediction approach for ice affected coasts if the data is to be used in ice flow modelling. Generating data for ice flow modeling was not the primary focus of our paper though, as this was an initial paper to see if predictions could be generated using a reference station, and in this diurnal tide dominated environment (whereas Byun and Hart 2015 has more complete data conditions and semidiurnal dominated tidal regimes). Further work beyond our paper, examining the long-period tidal constituents, could help inform the objectives of future Antarctic tidal measurement fieldwork campaigns.

Line 273: Section 5.2 does not seem to contribute to the main aim of the paper, i.e. to predict tides from 25 hr observations. 5.2 looks at the contrasting tidal environments of two areas and tries to explain why they differ. I think 5.2 could be removed. Response: In response to this comment, and additional detailed comments on this section from the Editor, we have substantively tightened section 5.2, removing much text exploring nodal modulation correction factors (including Fig. 13). We have also better explained the role of this section in our paper, being to show how the Ross Sea tides compare to the other diverse and out of phase regimes around Antarctica. The text now reads:

"5.2 Understanding the contrasting tidal environments around Antarctica Figure 11 illustrates the form factors of tidal regimes in the seas surrounding Antarctica, according to FES2014 model data. There are large areas characterised by 'diurnal' (F>3); 'mixed, mainly diurnal' (1.5>F<3); and 'mixed, mainly semidiurnal' (0.25>F<1.5) forms. Only in a small area half-way along the Weddell Sea coast of the Antarctic Peninsula (at 72°S) do tides exhibit a 'semidiurnal' form (F<0.25). Strong 'diurnal' tides predominate in the Ross Sea area of West Antarctica, around to the Amundsen Sea. In addition, a small area near Prydz Bay (Fig. 2) in East Antarctica exhibits diurnal and mixed mainly diurnal tides. The rest of the seas surrounding Antarctica, including the Weddell Sea, are predominantly characterised by 'mixed, mainly semidiurnal' tides. Since diurnal tides have larger nodal factor and nodal angle variations than semidiurnal tides (Pugh and Woodworth, 2014), areas like the Ross Sea will have larger variations in tidal height across the 18.61 yr lunar nodal cycle compared to areas like the Weddell Sea (see details for ROBT in Byun and Hart, 2019). As the nodal angle variations of the diurnal and semidiurnal tides are out of phase, this leads to differing tidal responses around Antarctica over 18.61 years, particularly between the Ross and Weddell Seas. Given that CTSM+TCC is based on modulated tidal constant corrections for each diurnal and semidiurnal species, it is applicable in studying a continent with such a diversity of tidal regime types. Accurate (cm scale) quantification of the contrasting tidal behaviours and environments around Antarctica's margins are not only of use for polar station maritime operations, they are essential for estimating ice flows to the sea. This paper has shown how the CTSM+TCC approach may be used to complement existing efforts to quantify variations in tidal processes around Antarctica, in particular for places with sparse in situ tidal monitoring, such as the Ross Sea".

Figures A1 and A2: If the paragraph at lines 103 -111 is deleted then these figures are no longer required. Response: Yes - these two pages of figures have now been deleted, as has the above mentioned paragraph.

Figure 2: Readers might find this more informative if the map covered the Ross Sea

only. Response: A map focusing on the Ross Sea area only has been added to Fig. 2 (see (b)). We retained the Antarctica map (a) as well since it is of use when interpreting Fig. 11. This figure has been uploaded with this reply and its full caption now reads: "Figure 2. Maps showing (a) the locations of the two tidal observation stations employed in this study within a wider Antarctic context: Jang Bogo Antarctic Research Station (JBARS, âŰš) and Cape Roberts (ROBT, âŰŔ); and (b) the case study station locations relative to two other (previous) temporary tidal observations stations, McMurdo Station (âŰă), and Mario Zucchelli Station (âŰŔ), in the Ross Sea".

Figure 3: The x-axis label should be 'Time (month/day)'. The description starts 'Seventeen day time series: : :". Isn't it seventeen sets of daily (25 hr) data slices as stated in line 130? Response: Yes, this x-axis label has been fixed. Also, the figure caption has been improved as suggested, and colour added to the lines and key. This figure has been uploaded with this reply and its full caption now reads: "Figure 3. Modulated tidal (a) species amplitudes and (b) phase lags for the diurnal and semidiurnal tidal species, calculated from Cape Roberts (ROBT) tidal prediction data (29 January to 14 February 2017), using Appendix 1 Eqs. (A1) and (A3)".

Figure 4: The x-axis for all four plots needs a label (Time (month/day)". The description refers to daily slices of the 17 day ROBT tidal predictions in the first sentence, but the second sentence refers to results of the 369 day 2013 ROBT analysis. Is this correct? Response: The x-axis labels of all four Fig. 4 plots have been fixed, and colour added to the lines and key. Note that this figure has been re-drawn to include the two plots (a and f) that were formerly Figure 5. The words 'harmonic' and "In addition" have been added to make this caption clearer. This figure (combining the previous Figures 4 and 5 together) has been uploaded with this reply and its full caption now reads: "Figure 4. Daily amplitudes (a, c); phase lags (b, d); amplitude ratios (e); and phase lag differences (f) of the K1 and M2 tides (representative diurnal and semidiurnal tide species) at ROBT (a, b) and JBARS (c, d), and between JBARS and ROBT (e, f), calculated from 25 hr 'daily' slices of the 29 January to 14 February 2017 ROBT tidal

[Figure]

predictions and JBARS sea level observations. In addition, thick blue (K1) and thin pink (M2) horizontal lines in the panels indicate the amplitudes and phase lags derived from harmonic analyses of the 369 day 2013 ROBT sea level records (a, b) and of the 17 day 2017 JBARS sea level records (c, d), along with their amplitude ratios and phase lag differences (e, f)".

Figure 5: Both plots need a label for the x-axis. The description refers to dashed lines in the plots but these are not shown. Response: Thank you for spotting this typo. Plot x-axis labels have been added (now Figure 4 e and f). Also the caption description has been amended (see figure and caption in the response to the comment immediately above).

Figure 6: X-axis label (for both plots) should read Time (month/day). In the key, should the Moon's maximum declination be qualified as being either north or south? Line 429: the symbol (open circle) does not match the plot. Response: The label in x-axis of this figure (now Fig. 5) has been fixed. The qualifier 'northern' has been added, and the description has been changed to the Moon's greatest northern declination in the figure key. In the caption the symbol âŮŃ has been swapped to âŹę. This figure (formerly 6, now 5) has been uploaded with this reply and its full caption now reads: "Figure 5. (a) Time series (29 January to 14 February 2017) of Root Mean Square Errors (RMSE, thick blue line with âŮŔ) and coefficients of determination (R2, thin black line with âŮŃ) between JBARS 10 min interval sea level observations and the CTSM+TCC prediction datasets, generated for this site using harmonic analysis results from the JBARS daily (25 hr) sea level data slices and concurrent daily (25 hr) 2017 tidal prediction data slices and harmonic analysis results from ROBT station's year-long (2017) tidal predictions. (b) Time series of predicted 2017 tidal heights (thin blue line) and daily tidal ranges (thick black line with âŹę) for ROBT, based on harmonic analysis of this station's 2013, 5 min interval sea level records, plus an indication of the moon's phase and declination".

Figure 7: X-axis label could be consistent with the other figures. Line 435 has the

word 'plus' – this makes me confused about the description. My take is that the plot compares predictions for day x of 2019 (as derived from analysis of data from day y of 2017) with observations made on day x of 2019. Have I got this correct? Response: The x-axis label has been amended to match that of the other figures, and colour has been added. Yes, your understanding is correct and we have taken on board the comment regards caption readability. To make this caption easier to read we replaced the word 'plus' with 'along with', and added brackets around details of the CTSM+TCC data inputs. This figure (formerly 7) has been uploaded and its full caption now reads: Figure 6. Time series of Root Mean Square Errors (RMSE, thick blue line with åŮŔ) and coefficients of determination (R2, thin black line with åŮŃ) between JBARS 10 min interval sea level observations (29 December 2018 to 18 January 2019) and the CTSM+TCC prediction data sets generated for this site (using harmonic analysis results from daily (25 hr) summertime 2017 sea level data slices from JBARS along with concurrent daily (25 hr) tidal prediction slices and harmonic analysis results from ROBT station's year-long (2017) tidal predictions).

Figure 8: X-axis labels again. As with Figure 7, I am confused by the statement that follows '(dashed box in (a)),'. Response: Thank you - the plot x-axis labels have been fixed. We have amended the (now second to last) confusing sentence of this caption to make it precise and easier to interpret, and added redrawn the plot. This figure (formerly 8, now 7) has been uploaded and its caption now reads: "Figure 7. Time series of JBARS sea level observations, predicted tidal heights, and sea level residuals (i.e. observations minus predictions) from (a) 29 January to 14 February 2017; and (b) 29 December 2018 to 18 January 2019. The JBARS predictions were generated via the CSTM+TCC method (using a daily (25 hr) slice of local sea level observations from 8 February 2017 (dashed box in (a)), along with concurrent (to time periods a and b) ROBT predictions; and year-long (2017) 5 min interval ROBT tidal predictions). RMSE and R2 denote the comparison Root Mean Square Errors and coefficients of determination, respectively".

Figure 9: X-axis labelling differs from all other figures – could be altered for consistency. Line 444: should 'estimated' be 'calculated'? Response: The axis has been amended for consistency. 'Estimated' has been changed to 'calculated'. This figure (formerly numbered 9, now numbered 8) has been uploaded and its full caption now reads: "Figure 8. Time series of the Moon's declination, calculated at daily intervals for two observation periods: (a) 1 January to 15 February 2017; and (b) 16 December 2018 to 30 January 2019. Dashed boxes indicate the sea level observation windows examined in this study".

Figure 10: X-axis labels again. Response: The plot x-axis labels have been fixed (note this figure is now numbered 9). This figure's plot x-axes all now read: "Time (month/day)". This figure (formerly 10, now 9) has been uploaded and its caption now reads: "Figure 9. Time series of ROBT tidal predictions (a) made without long-period constituents ('SRun', i.e. excluding the constituents listed in Table 2) versus with the Mf tide ('Exp1'); and (b) time series of ROBT tidal predictions made ('SRun') without the long-period constituents versus ('Exp2') with the MSf and Mf tides. All predictions were generated based on tidal harmonic analysis results from the year-long (2013) ROBT sea level records".

Figure 11: If Section 5.2 is deleted then these figures are no longer required. If retained then the word 'Horizontal' in the description is redundant. Is the area in the Weddell Sea coloured magenta? Response: Fig. 12 has been deleted in shortening section 5.2. In Fig. 11 the word 'Horizontal' has been deleted. Yes, there is a magenta area shown in the Weddell Sea, indicating an area with a semi-diurnal tidal regime (F<0.25). We have added '(72°S) to make locating this spot easier. The Fig. 11 caption now reads: "Figure 11. Distribution of tidal form factor (F) values around Antarctica. Note the magenta area (72°S) on the Antarctic Peninsula's Weddell Sea coast denotes the only area with a properly semidiurnal tide regime (F<0.25) in the Antarctic region".

Table 2: I would delete the Period and Angular speed columns. Not only are the amplitudes of most constituents in this table small, but by my analysis they also have small

signal-to-noise ratios so are weakly determined. This caution about the reliability of these values should be noted. Response: Yes, the period and angular speed columns have been deleted while columns indicating amplitude standard errors and signal-to-noise ratios have been added, and a caution has been added into the text as follows. This sentence has been added to the last paragraph of section 5.1: "However, because these tides' amplitudes have small signal-to-noise ratios (SNR) (<1) with large standard errors (Table 2), caution should be exercised when elucidating fortnightly tide effects using these constituents". Table 2 will be included in the merged 'reply to all 3 reviews' file, and its caption now reads: "Table 2. Harmonic constants for 6 long-period tidal constituents, derived from harmonic analyses of year-long observations (2013) measured at the Cape Roberts sea level gauge (ROBT), using T_Tide (Pawlowicz et al., 2002). Phase lags are referenced to 0°, Greenwich and SNR denotes the signal-to-noise ratios.

Table 3: My records from ROBT for 2011 commence 21 November so the values given for that year can't come from yearlong observations (as is the case for the others in the table). Response: We have deleted this data and Table 3, in response to a comment in the Editor's review, so this point is no longer included in the paper. However we have re-checked our data records and found that we were correct in our original description of the full year of 2011 data, starting 1 January 2011. These data are available via: http://apps.linz.govt.nz/ftp/sea_level_data/ROBT/2011/00/. Please check this page as it may need to be altered: http://apps.linz.govt.nz/ftp/sea_level_data/ROBT/ROBT_readme.txt
* * *
[Figure]

**Fig. 1.** Figure 2. Maps showing (a) the locations of the two tidal observation stations employed in this study within a wider Antarctic context: Jang Bogo Antarctic Research Station (JBARS, âŰš) and Cape Roberts

[Figure]

**Fig. 2.** Figure 3. Modulated tidal (a) species amplitudes and (b) phase lags for the diurnal and semidiurnal tidal species, calculated from Cape Roberts (ROBT) tidal prediction data (29 January to 14 February

**Fig. 3.** Figure 4. Daily amplitudes (a, c); phase lags (b, d); amplitude ratios (e); and phase lag differences (f) of the K1 and M2 tides (representative diurnal and semidiurnal tide species) at ROBT (a, b) an

[Figure]

**Fig. 4.** Figure 5. (a) Time series (29 January to 14 February 2017) of Root Mean Square Errors (RMSE, thick blue line with âŮŔ) and coefficients of determination (R2, thin black line with âŮŃ) between JBARS 10 min

**RMSE & R² (JBARS, 2019)**

Figure showing RMSE (cm) on left axis (0–25), Coefficient of determination (R²) on right axis (0–1), Time (month/day) on x-axis from 1/29 2017 to 2/13.

**Fig. 5.** Figure 6. Time series of Root Mean Square Errors (RMSE, thick blue line with âŮŔ) and coefficients of determination (R2, thin black line with âŮŃ) between JBARS 10 min interval sea level observations (29

[Figure]

**Fig. 6.** Figure 7. Time series of JBARS sea level observations, predicted tidal heights, and sea level residuals (i.e. observations minus predictions) from (a) 29 January to 14 February 2017; and (b) 29 Decemb

**OSD**

Interactive
comment

**Fig. 7.** Figure 8. Time series of the Moon's declination, calculated at daily intervals for two observation periods: (a) 1 January to 15 February 2017; and (b) 16 December 2018 to 30 January 2019. Dashed boxes

[Figure]

[Figure]

**Fig. 8.** Figure 9. Time series of ROBT tidal predictions (a) made without long-period constituents ('SRun', i.e. excluding the constituents listed in Table 2) versus with the Mf tide ('Exp1'); and (b) time ser

---

## Author Comment (AC3) · 19 May 2020

Reply to interactive comment of 14 Feb 2020 on "Predicting tidal heights for extreme environments: From 25 h observations to accurate predictions at Jang Bogo Antarctic Research Station, Ross Sea, Antarctica" by Philip Woodworth (Editor)

Do-Seong Byun1, Deirdre E. Hart2 1Ocean Research Division, Korea Hydrographic and Oceanographic Agency, Busan 49111, Republic of Korea 2School of Earth and Environment, University of Canterbury, Christchurch 8140, Aotearoa New Zealand Cor-

respondence to: Deirdre E. Hart (deirdre.hart@canterbury.ac.nz)

Format We are very grateful for this editorial comment as it has been useful in helping to improve our paper. In this reply we copied each individual reviewer point, wrote below it a response, then pasted the modified text of the paper to show the changes made.

Individual review comments and our responses.

My comments seem to be closer to those of Reviewer 2 than Reviewer 1. It will be best if all three sets of comments are taken together for any new version (see below). In the following I give a list of comments on the writing (there are several sentences without verbs, for example). But the main thing is that I thought there were 3 sections that either need considerable improvement or should be dropped. Response: According to this useful suggestion, we have completed replies and paper adjustments in response to review 2 (by Rowe) and review 3 (the present review) together, occasionally cross-referencing the two replies.

(1) Section 3. I understand that the method is some kind of response method, and I have read the authors' 2015 paper. However, I defy anyone to understand this section as it stands. It is made worse by not defining many variables (e.g. line 128, what are r, eta and tau. I believe s is species; line 136, what are k,m etc.). And I am sure there must be errors in equation 2 although I am not sure what e.g. it has a parameter j which is a subscript of a constituent like 'i', but which is not summed over but used only as a lower limit i=(j,m), but the left side of the equation is not a function of j. That cannot be right. Then also, what is a 'representative harmonic constituent'? I think a simpler thing to have done would have not included the little bits of maths here which just confuse everyone but just referred the reader to the 2015 paper for the method. I have many detailed comments on this section also which I list below. Response: From this comment we appreciated the need to improve our methodological communication based on your comments above. As a result we wrote a simpler explanation of the
approach for Section 3, and cut and pasted all of the original math parts into the new Appendix 1. In this new appendix we defined all undefined terms and fixed the issues you identify below. The newly focused Section 3 does a better job of highlighting differences in application of the Byun and Hart (2015) approach applied in this paper (i.e. the use of prediction data for SHr; and the procedure to select an optimal 25 hr data window in a diurnal tide dominated setting), differences that arose due to the extreme and particular (diurnal dominated) environment of the Ross Sea. The text of section 3 now reads: "Having analysed the tidal harmonic constants at the two stations based on their concurrent short-term records, we then employed the CTSM+TCC method (Byun and Hart, 2015) to generate tidal height predictions for JBARS, our 'temporary' tidal observation station (subscript o), using ROBT as the 'reference' station (subscript r). This prediction approach (see Appendix 1 for the detailed calculations, and Byun and Hart (2015) for explanation of procedure development) is based on: using long-term ($\geq$183 days) reference station records (LHr) and CTSM calculations to make an initial anytime ($\tau$) tidal prediction ($\eta\_r\,(\tau)$), which involves summing tidal species' heights for the reference station (Fig.3); comparing the tidal harmonic constants (amplitude ratios and phase lag differences) of representative tidal constituents (e.g., M2 and K1) for each tidal species between the temporary and reference stations, calculated using T_TIDE and concurrent short-term records ($\geq$25 hr duration, starting at midnight) from the temporary (SHo) and reference (SHr) stations; and using the step (ii) comparative data and the TCC calculations for each tidal species to adjust the $\eta\_r\,(\tau)$ tidal species' heights in order to generate accurate, anytime tidal height predictions for the temporary tidal station ($\eta\_o\,(\tau)$). In this Ross Sea case study we used the 2017 austral summertime JBARS tidal observation records (i.e. 17.04 days from 00:00 29 January to 01:00 15 February) as a source of SHo, keeping the second 2019 JBARS observation record for evaluation purposes. Importantly, this method assumes that the reference and temporary tidal stations are situated in neighbouring regimes with similar dominant tidal constituent and tidal species characteristics, and that the tidal properties between the two stations remain similar through time. As explained above, both JBARS and ROBT

have tidal regimes that are primarily dominated by diurnal tides. LHr must comprise high quality (e.g. few missing data) tidal height observations from anytime. For SHo and SHr, Byun and Hart (2015) employed observational data for both records, but as will be demonstrated in this paper, the method can also be applied using tidal predictions as a source of SHr. This slight adjustment in approach arose since for the 2017 JBARS observation time period, the concurrent 2017 ROBT records available online (LINZ, 2019) had multiple missing data. We solved this issue by producing a year-long synthetic 2017 record for ROBT using T_TIDE (Pawlowicz et al., 2002) and the 2013 (i.e. LHr) observational record as input data. The 17.04 days of predicted tides that were concurrent with the 2017 JBARS observation record were then used as our SHr source. While this CTSM+TCC adjustment was procedurally small, it represents an important adaptation in the context of generating tidal predictions for stations situated in extreme environments, since concurrent temporary and reference station observations might be challenging to obtain in such contexts. When using CTSM+TCC, if the available temporary tidal station observation record covers multiple days, it is best practice to experiment by generating multiple $\eta\_o$ ($\tau$), each using different concurrent pairs of SHo and SHr daily data slices in step (ii) above to produce daily amplitude ratios and phase lag differences between the two stations for the diurnal K1 and semidiurnal M2 tidal constituents. Comparisons are then made between the different $\eta\_o$ ($\tau$) data sets produced and the original temporary station observations, to determine the optimal 25 hr window to use in subsequent calculations of anytime $\eta\_o$ ($\tau$). Once the optimal window is selected, tidal height predictions can be generated for the temporary observation station for any time period. Thus, 17 individual 25 hr duration data slices were clipped from the 2017 summertime JBARS observation records and from the concurrent ROBT predictions, forming 17 pairs of SHo and SHr 'daily' slices. Each paired data set was then used with LHr to generate tidal height predictions for JBARS covering both the 2017 and 2019 KHOA temporary observation campaign time periods. Comparisons were made between the JBARS observations and the 17 prediction data sets generated for each observation campaign to identify which 25 hr short-term data window produces optimal $\eta\_o\ (\tau)$ results".

And the new Appendix 1 (complimenting the new section 3) now reads: Appendix 1 This appendix describes the calculations involved in using the CTSM+TCC approach as employed in this Ross Sea, Antarctica, case study. For a fuller description of the development of this approach and its application in 'semidiurnal' and 'mixed mainly semidiurnal' tidal regime settings, see Byun and Hart (2015). As explained in the main body of this paper, we used 25 hr slices of the 2017 short-term observations from JBARS (SHo), our temporary tidal observation station (subscript o), and 2013 year-long observations (LHr) and 2017 short-term tidal predictions (SHr, concurrent with SHo) from ROBT, our reference tidal station (subscript r), as the basis of JBARS tidal prediction calculations. We then employed the full 17.04 day 2017 JBARS tidal observation data set, and an additional 21.54 day 2019 JBARS tidal observation dataset, to evaluate the success of the CTSM+TCC tidal prediction calculations for this site. The CTSM+TCC, expressed as the summation of each tidal species cosine function, includes three key steps: calculating each tidal species' modulation at the reference tidal station; comparing the tidal harmonic constants between the temporary observation and reference stations (e.g., the tidal amplitude ratios and phase lag differences of each representative tidal constituent for each tidal species calculated from concurrent observation records between two stations); and adjusting the tidal species modulations calculated in the first step using the correction factors calculated in the second step to produce predictions for the temporary tidal station. As a first step, tidal height predictions for the temporary station ($\eta\_o\ (\tau)$) were initially derived from reference station predictions ($\eta\_r\ (\tau)$) on the assumption that the tidal properties between the two stations remain similar through time. Using the modulated amplitude ($A\_r\hat{}((s))$) and the modulated phase lag ($\varphi\_r\hat{}((s))$) for each tidal species, this step is expressed as: $\eta\_r\ (\tau)=\sum\_(s\ =\ 1)^k A\_\Theta(s))(\tau)$ cosâĄą$(\omega\_R\hat{}((s))$ t-$\varphi\_r\hat{}((s))\ (\tau))$ ãĂŮ (A1) with $A\_r\hat{}((s))\ (\tau)=\sqrt{}(\sum\_(i\ =\ 1)^n [f(\tau)ãĂŮ\_i\hat{}((s))\ a\_i\hat{}((s))\ ]\hat{}2+2\sum\_(i\ =\ 1)^(m-2)\sum\_(j=i+1)^n [f(\tau)ãĂŮ\_i\hat{}((s))\ a\_i\hat{}((s))\ ][ãĂŮf(\tau)ãĂŮ\_j\hat{}((s))\ a\_j\hat{}((s))\ ]$ cosâĄą$\{(\omega\_i\hat{}((s))-\omega\_j\hat{}((s))\ )t+[ãĂŮV(t\_0\ )ãĂŮ\_i\hat{}((s))+ãĂŮu(\tau)ãĂŮ\_i\hat{}((s))-G\_i\hat{}((s))\ ]-$

[ãĂŰV(t_0 )ãĂŮ_jˆ((s))+ãĂŰu($\tau$)ãĂŮ_jˆ((s))-G_jˆ((s)) ]} ãĂŮãĂŮ) (A2) and $\varphi$_rˆ((s)) ($\tau$)=ãĂŰtanãĂŮˆ(-1) (($\sum$_(i = 1)ma_$\Theta$(s))sin[($\omega$_iˆ((s))-$\omega$_Rˆ((s)))t+ ãĂŰV(t_0 )ãĂŮ_iˆ((s))+ãĂŰu($\tau$)ãĂŮ_iˆ((s))-G_iˆ((s)) ãĂŮ])/($\sum$_(i = 1)ma_$\Theta$(s))cos[($\omega$_iˆ((s))-$\omega$_Rˆ((s)))t+ ãĂŰV(t_0 )ãĂŮ_iˆ((s))+ãĂŰu($\tau$)ãĂŮ_iˆ((s))-G_iˆ((s)) ãĂŮ])) (A3) where superscript s denotes the type of tidal species (e.g., 1 for diurnal species and 2 for semidiurnal species); m is the number of tidal constituents; t_0 is the reference time; t is the time elapsed since t_0; and ãĂŰ$\tau$= tãĂŮ_0+t; $\omega$_iˆ((s)) are the angular frequencies of each tidal constituent (subscripts i and j), $\omega$_Rˆ((s)) are the angular frequencies of each tidal constituent representing a tidal species (subscript R), with the dominant tidal constituent of each tidal species used as the representative for that species (e.g., K1 and M2 are used as representative of the diurnal and semidiurnal species, respectively). For each tidal constituent, a_iˆ((s)) and G_iˆ((s)) are the tidal harmonic amplitudes and phase lags (referenced to Greenwich), ãĂŰf($\tau$)ãĂŮ_iˆ((s)) is the nodal amplitude factor of each tidal constituent, ãĂŰu($\tau$)ãĂŮ_iˆ((s)) is the nodal angle and ãĂŰV(t_0 )ãĂŮ_iˆ((s)) is the astronomical argument. T_TIDE was used for tidal harmonic analysis as well as for calculation of the nodal amplitude factors, nodal angles and astronomical arguments, for the representative tidal constituents. As the second step, under the 'credo of smoothness' assumption that the admittance or 'ratio of output to input' does not change significantly between constituents of the same species (Munk and Cartwright, 1966; Pugh and Woodworth, 2014), the amplitude ratio and phase lag difference of each representative tidal constituent for each tidal species between the temporary and reference stations were calculated from the results of tidal harmonic analyses of concurrent 25 hr data slices (starting at 00.00) from the temporary tidal observation and reference tidal stations (i.e. from SHo and SHr). The process of selecting the optimal 25 hr window for these concurrent data slices from amongst the 17.04 days of available records is explained in Section 3 of this paper. Once the optimal 2017 short-term data window was selected, the third step involved adjusting the tidal predictions at the reference station calculated from Eq. (A1), to represent those for the temporary station ($\eta$_o ($\tau$)), by substituting the daily (i.e. SHo and SHr) amplitude ratios $((a\_o^{((s))})/(a\_r^{((s))}))$ and phase lag differences $(G\_o^{((s))}-G\_r^{((s))})$ for the tidal constituents (K1 and M2) representing the diurnal and semidiurnal tidal species between the temporary and reference stations into Eq. (A1) as follows (Byun and Hart, 2015): $\eta\_o(\tau)=\sum\_(s=1)^{\text{H}}A\_@(s))(\tau)$ cosâĄą$(\omega\_R^{((s))}$ t-$\varphi\_o^{((s))}(\tau))$ ãĂŮ (A4) with $A\_o^{((s))}(\tau)=A\_r^{((s))}(\tau)((a\_o^{((s))})/(a\_r^{((s))}))$, and (A5) $\varphi\_o^{((s))}(\tau)=\varphi\_r^{((s))}(\tau)+G\_o^{((s))}-G\_r^{((s))}$ (A6) Substituting Eqs. (A5) and (A6) into Eq. (A4), $\eta\_o(\tau)$ can be expressed as: $\eta\_o(\tau)=\sum\_(s=1)^{\text{H}}A\_@(s))(\tau)((a\_o^{((s))})/(a\_r^{((s))}))$ cosâĄą$[\omega\_R^{((s))}$ t-$(\varphi\_r^{((s))}(\tau)+G\_o^{((s))}-G\_r^{((s))})]$ ãĂŮ (A7)

The T_TIDE based CTSM code is available from https://au.mathworks.com/matlabcentral/fileexchange/73764-ctsm_t_tide.

(2) Section 5.1. You have records of the order of a fortnight so I dare say it is inevitable that there will be mismatches on that timescale between your method and the data. However, do you need a page to say that? I suggest that this aspect should be summarised in 5-6 lines in the Discussion section where it can be a pointer to improvements in the method. Also I wondered if you considered the missing fortnightly tide was consistent with that in FES2014. Response: Yes, we agree so have reduced section 5.1 significantly, deleted Table 3 altogether, and re-ordered some sentences, so that this section makes a clear point for future research improving the prediction work specifically to make it useful for ice flow studies. Interannual harmonic analysis results at ROBT show that the fortnightly tide has large variations with large standard errors and small signal to noise ratios. We do not think that they are easily comparable with those in FES2014. The first 2/3 of section 5.1 (excluding the double tide peaks explanation – see further below) text now reads: "5.1 Explaining fortnightly tide effects and double tide peaks in Ross Sea tidal predictions We have demonstrated that the CTSM+TCC approach can produce reasonably accurate tidal predictions (RMSE <5 cm, R2 >0.92) for a new site in the Ross Sea, Antarctica, based on 25 hr temporary observation records from periods with higher than average tidal ranges, plus neighbouring reference station records. Our results compare favourably with those of Han et al. (2013), who reviewed the tidal height prediction accuracy of 4 models for Terra Nova Bay, Ross Sea: these models generated similar quality results to our CTSM+TCC results, with R2 values between 0.876 and 0.907, and RMSEs ranging from 3.6 to 4.1 cm. However, as shown in Fig. 7, our results contain a changing fortnightly timescale bias in estimates. This error pattern likely resulted from our application of CTSM+TCC considering only 2 major tidal species (diurnal and semidiurnal) whilst ignoring several long period tides. Table 2 summarises the characteristics of 6 long-period tides (Sa, Ssa, MSm, Mm, Mf, MSf) at the ROBT station, derived from tidal harmonic analysis of year-long (2013) in situ observation records. To verify the main cause of the apparent fortnightly prediction biases in results, in particular that in the 2019 predictions (Fig. 7b), we examined the effects of two fortnightly tidal constituents (Mf, and MSf) at ROBT. Three 2019 tidal prediction experiments were conducted: Srun excluded all long-period tides (see list of exclusions in Table 2); Run1 was based on Srun but also incorporated the Mf; and Run2 was based on Srun but also incorporated the Mf and MSf. Comparisons between Run1 and Srun predictions show that exclusion of the Mf tide (2.7 cm amplitude) can produce prediction biases during periods of lunar declination change (Fig. 9a), with comparisons between Run2 and Run1 results showing that the additional exclusion of the MSf tide (1.2 cm amplitude) intensifies the biases (Fig. 9b). Rosier and Gudmundsson (2018) found that ice flows are modulated at various tidal frequencies, including that of the MSf tide. However, because these tides' amplitudes have small signal-to-noise ratios (SNR) (<1) with large standard errors (Table 2), caution should be exercised when elucidating fortnightly tide effects using these constituents. Nevertheless, studies indicate that incorporating major and minor tidal constituents, including long period tides, into tidal predictions may be advantageous for their use in ice flow and ice-ocean front modelling specifically (e.g. Rignot et al., 2000; Rosier and Gudmundsson, 2018). Consideration of additional, long period tides in predictions is one recommendation we have for future work on improving tidal predictions for Ross Sea coasts".

(3) Section 5.2. Having shown that the Ross and Weddell Seas have different dominant tides (and form factors), end of story to me, you embark on generating predictions over 18.6 years which lo and behold have the ranges (you don't explain range = 2*amplitude) which have exactly the equilibrium amounts that T-Tide must be coded with. So what have you learned? Nothing. The finding is presented as some kind of new result. I suggest, having indicated the map of form factors, you just say that because diurnal tides have larger 'f' and 'u' variations than semidiurnal tides (reference a text book) then they will have larger ranges of tide over 18.6 years. Response: Yes, agreed. We have significantly shorted this section to better situate the task of understanding Ross Sea tides within the different regimes that occur around Antarctica. Also, we added explanation of the term 'tidal range' near its first use, after the abstract, in section 4.1. The section 4.1 text now includes: "As illustrated in Fig.5, the RMSE and R2 results varied in relation to the JBARS tidal range (range being twice amplitude), with greater accuracy evident in predictions made using data derived from 25 hr periods when the tidal range was higher than average". Section 5.2 now reads: "Figure 11 illustrates the form factors of tidal regimes in the seas surrounding Antarctica, according to FES2014 model data. There are large areas characterised by 'diurnal' (F>3); 'mixed, mainly diurnal' (1.5<F<3); and 'mixed, mainly semidiurnal' (0.25<F<1.5) forms. Only in a small area half-way along the Weddell Sea coast of the Antarctic Peninsula (at 72°S) do tides exhibit a 'semidiurnal' form (F<0.25). Strong 'diurnal' tides predominate in the Ross Sea area of West Antarctica, around to the Amundsen Sea. In addition, a small area near Prydz Bay (Fig. 2) in East Antarctica exhibits diurnal and mixed mainly diurnal tides. The rest of the seas surrounding Antarctica, including the Weddell Sea, are predominantly characterised by 'mixed, mainly semidiurnal' tides. Since diurnal tides have larger nodal factor and nodal angle variations than semidiurnal tides (Pugh and Woodworth, 2014), areas like the Ross Sea will have larger variations in tidal height across the 18.61 yr lunar nodal cycle compared to areas like the Weddell Sea (see details for ROBT in Byun and Hart, 2019). As the nodal angle variations of the diurnal and semidiurnal tides are out of phase, this leads to differing tidal responses around Antarctica over 18.61 years, particularly between the Ross and Weddell Seas. Given

that CTSM+TCC is based on modulated tidal constant corrections for each diurnal and semidiurnal species, it is applicable in studying a continent with such a diversity of tidal regime types. Accurate (cm scale) quantification of the contrasting tidal behaviours and environments around Antarctica's margins are not only of use for polar station maritime operations, they are essential for estimating ice flows to the sea. This paper has shown how the CTSM+TCC approach may be used to complement existing efforts to quantify variations in tidal processes around Antarctica, in particular for places with sparse in situ tidal monitoring, such as the Ross Sea".

In 5-6 lines again. Also, on line 67, you say that section 5 will discuss double tidal peaks. I can't see anything about that in the section or the paper. Because of the problems with these 3 sections, which make up most of the paper, I expect that it will not be acceptable for OS without considerable improvements. Anyway, I am unclear what has been learned new here which you didn't learn from NZ and Korea data in the 2015 paper - I realise this is a different tidal regime but a step change might have been to write a larger paper with as many regimes as possible if you wanted to demonstrate that your method works well. Response: We have added explanation of the occurrence of double tidal peaks at the end of Section 5.1, and also included the new Fig. 10 to explain their occurrence. Thank you for the suggestions regards a step change paper – we agree that this is a very good idea for showing the method works well in a range of different tidal environments. We still feel that this, now shorter and significantly improved, paper is of use for showing the method can fulfil a need specifically for tidal predictions in extreme environments where data may be scarce, with the added bonus that this case study is of diurnal dominated tides, contrasting the Byun and Hart (2015) paper, which focused on semidiurnal dominated environments. We hope that you feel we have given proper consideration and response to your comments throughout. The last part of section 5.1 now reads: "Another characteristic of our results needing explanation is the double tidal peaks evident in both the tidal observations and predictions at JBARS. These peaks occur, for example, in Fig. 7b between January 11th and 17th, 2019. To explore why these double peaks occur, we generated JBARS tidal height predictions using Eq. (A1) and the 2019 tidal constants listed in Table 1 for the two major diurnal (K1, O1) and semidiurnal tides (M2, S2). Fig. 10a shows separately the resulting diurnal (with their period of 13.66 days) and semi-diurnal (with their period of 14.77 days) species' tide predictions. The combination of these out-of-phase tidal species generates double peaks (or double troughs) around low tide (Fig. 10b) for periods when the diurnal tide amplitudes are low, and the amplitude ratio of the semi-diurnal to diurnal tide species is >0.5 (Fig. 10c). Double peaks also occur around high tide during periods of low lunar declination (Fig. 8b), when the semidiurnal to diurnal species amplitude ratio is >1, and the phase lag difference between the diurnal and semidiurnal species is between -78° and 46° (Fig. 10). Since the semi-diurnal tides are slightly stronger, and the diurnal tides are slightly weaker, at JBARS compared to at ROBT (Table 1), these double tide peaks occur more commonly at JBARS (e.g., compare Fig. 5b and Fig. 7)". The new Fig. 10 has been uploaded and its full caption now reads: "Figure 10. Time series (29 December 2018 to 18 January 2019) of (a) predictions of the diurnal (K1+O1) tides (blue line) and the semidiurnal (M2+S2) tides (magenta line) for JBARS; (b) their combined JBARS predictions (red line) and observations (black dashed line); (c) the ROBT diurnal (blue line) and semidiurnal (magenta line) species amplitudes and their ratio (green line); and (d) the ROBT diurnal (blue line) and semidiurnal (magenta line) species phase lags and their difference (diurnal – semidiurnal) (green line)".

Detailed comments: 9-10 - sentence 'Though obtaining'. This sentence has no verb. Response: The verb in this sentence is "is" (line 10 of submitted PDF abstract). In response to reviewer 2, we have altered the qualifier at the beginning by swapping "Though" for "However". The text now reads: "However obtaining long term sea level records for traditional tidal predictions is extremely difficult around ice affected coasts".

- by a different tidal regime Response: Thank you, we have added the word "tidal". The text now ends: "...to accurately predict tides for a temporary tidal station in the Ross Sea, Antarctica using records from a neighbouring reference station characterised by a similar tidal regime".

- I have never seen this 'tropic-spring' description before (there are other examples below). Could you not just replace this simply with 'at high lunar declination', or whatever, which means something physical rather than poetic. Response: Yes, "tropic" has been removed throughout the paper and replaced with "at high lunar declinations". For example, line 18 test now reads: "Results reveal the CTSM+TCC method can produce accurate (to within ~5 cm Root Mean Square Errors) tidal predictions for JBARS when using short-term (25 hr) tidal data from periods with higher than average tidal ranges (i.e. those at high lunar declinations and/or spring periods)".

- the Rignot reference is rather old. There has been a lot of work using GPS for tides under ice sheets, and there new data sets (IceSAT etc.). I am sure you can find a couple of better references. Response: We have removed the Rignot et al. (2000) reference from this first Introduction paragraph and instead inserted some text referring to more recent work improving tide models for shallow, ice affected seas including that using IceSAT. The text now reads: "Obtaining long term records for such tidal analyses is extremely difficult for sea ice affected coasts, like that surrounding Antarctica. As compliment to in situ tidal records, recent work has significantly advanced our understanding of tide models for the shallow seas around Antarctica and Greenland via the assimilation of laser altimeter data and use of Differential Interferometric Synthetic Aperture Radar (DInSAR) imagery, amongst other methods (Padman et al., 2008; 2018; King et al., 2011; Wild et al., 2019)".

- though –> although Response: This change has been made as suggested. The text now reads: "Accurate tidal records from the Ross Sea and other areas around Antarctica are thus scarce compared to those available from other regions, although these data are much needed given the crucial role of tidal processes around this continent..."

- transfers –> transfer Response: This has been changed as suggested. The text now reads: "...and control heat transfer and ocean mixing in cavities beneath the marine cryosphere...".

- GPS is usually called GNSS these days Response: This has been changed as suggested. The text now reads: "Ice thickness is typically measured via the subtraction of tidal height oscillations from highly accurate, but relatively low frequency, satellite imagery based observations of ice surface elevation and/or from in situ Global Navigation Satellite System (GNSS) instrument observations (Padman et al., 2008)".

- see above Response: Yes, as indicated above we have now added an explanation of the double tidal peaks to section 5.1.

- what does high-frequency mean? Response: This text has been improved to properly indicate the frequency, and of what. The text now reads: "High-frequency sea level oscillations (<3 hr) were removed from the observation record using a fifth-order low-pass Butterworth filter".

- year-long Response: The term 'yearlong' has been replaced with 'year-long' throughout the paper. For example, this line now reads: "...it was not possible to collect the year-long sea level records that are commonly employed to obtain reliable tidal constituents".

- this needs rewording. the 2 main diurnal and semidiurnal tides are K1 and O1 and M2 and S2 of course - what you mean here are the 2 main relationships taken from Cape Roberts 97 - they have similar amplitudes. not 'characterised by' 97 - between –> at 98 - for S2 respectively. 99 - close –> short But I don't consider 269 km a short distance. I am sure the tide around Korea or NZ, for example, changes enormously in that distance. And what does 'in tidal terms' mean? Response: Thank you for each of these comments. We are replying to them together since we have re-worded several stanzas in this paragraph to make all of the suggested changes and modifications. Specifically, we modified the 'line 92' sentence to correct it according to the issues you identified in our description of the inference method and with our previous adjective choices. The two 'line 97' changes have also been made as indicated, while the line 98 text was deleted in response to a comment in the second (Rowe) review. Regards 'line 99', both 'close distance' and 'in tidal terms' have been deleted. This paragraph now reads: "Using the T_TIDE toolbox (Pawlowicz et al., 2002), we obtained the tidal harmonic constants of the 8 and 6 major tidal constituents for ROBT and JBARS, respectively. The inference method was used to separate out neighbouring diurnal (K1 and P1) and semidiurnal (S2 and K2) tide constituents, with their amplitude ratios and phase lag differences obtained from harmonic analysis of the long-term ROBT reference station records. Analysis revealed that the two main diurnal (O1 and K1) and semidiurnal (M2 and S2) tides had similar amplitudes at the two stations (Table 1), with the diurnal amplitudes being slightly larger at ROBT than at JBARS, the semidiurnal amplitudes being slightly smaller at ROBT than at JBARS, and the phase lags of all four tides having only slightly different values".

- phase lag usually has no hyphen Response: Phase-lag has been amended to phase lag throughout the text of the entire paper (deleting the hyphen).

- what does tidal patterns mean? You mean tidal characteristics? Response: Yes, agreed – this word has been changed (and F has also been explained in Table 1, due to a comment in review 2). The text now reads: "The amplitude differences result in slightly different tidal characteristics as indicated by the two sites' tidal form factors (e.g., F in Table 1)".

- database –> model 105 - drop horizontal 105-111 - there are amplitudes and phase lags, and there are co-amplitude (or sometimes co-range) and co-phase charts, sometimes combined as co-tidal charts. But there is no such thing as an 'increasing co-amplitude'. Please rewrite this paragraph. See below for the figures also. Response: This entire paragraph has been deleted in response to a comment from Reviewer 2 (Rowe).

- why a minus before CTSM? 114-115 - why the italics? Response: Thank you for picking up this minus sign typo: it was fixed, as indicated in the response to the second (Rowe) review, before this section was rewritten, cutting and pasting the math into a new Appendix 1. Regarding the italics, we intended to highlight these terms but removed them in response to a similar comment in the Rowe review (see details of new text there).

- remove simply 126 - remove accurate. You have no way of showing how accurate they are. 127 - remove sentence 'In short'. This is obvious. 128 - see above. Also mention that phase lags are Greenwich lags. Response: All of these changes have been implemented in the re-write of section 3 (see comment explaining this towards the beginning of this reply). This particular part of the new section 3 text now reads: "We solved this issue by producing a year-long synthetic 2017 record for ROBT using T_TIDE (Pawlowicz et al., 2002) and the 2013 (i.e. LHr) observational record as input data. The 17.04 days of predicted tides that were concurrent with the 2017 JBARS observation record were then used as our SHr source. While this CTSM+TCC adjustment was procedurally small, it represents an important adaptation in the context of generating tidal predictions for stations situated in extreme environments, since concurrent temporary and reference station observations might be challenging to obtain in such contexts".

- sentence 'Note that'. Again I think that assumes you understand the method Response: We have explained this much more clearly in the new Appendix 1, which includes improved text from the old section 3. Please see Appendix 1 towards the start of this reply.

- peculiarities –> properties? Response: This amendment has been made. The text, now in Appendix 1, now reads: "…on the assumption that the tidal properties between the two stations remain similar through time".

- again, I guess the reader will have to read the 2015 paper to understand why you produce 17 data sets? This has to be clearer. Response: The 17 data sets were produced since we had 17.04 days of quality input data from the temporary tidal station observation records of 2017. In fact the 2017 JBARS record spanned 19 days, but the first and last days' data were incomplete, so not useful for creating daily (25 hr) datasets. We have now made the origin of the 17 data sets clearer in section 2.1 of the paper, in response to your query. We also hope that our clearer section 3 eliminates the confusion created here. The text of section 2.1 now reads: "The Korea Hydrographic and Oceanographic Agency (KHOA) survey team went to JBARS in Northern Victoria Land's Terra Nova Bay, Ross Sea, Antarctica, in the austral summertime of 2017 (Fig. 2) for a preliminary fieldtrip to conduct hydrographic surveys and produce a nautical chart. This mission collected the first 19 day sea level records for JBARS: 10 min interval observation data, recorded between 28 January and 16 February 2017 using a bottom-mounted pressure sensor (WTG-256S AAT, Korea). High-frequency sea level oscillations (<3 hr) were removed from the observation record using a fifth-order low-pass Butterworth filter. Note that the first and last days of this campaign comprised partial day records, so we excluded these end days from our tidal prediction experiments, since our method requires continuous 25 hr input data, starting from midnight, for each prediction experiment. That left us with 17 days and 1 hour of useable tidal observation data as the basis of the temporary tidal station's primary observation record in this study".

- where –> when Response: This change has been made. The text now reads: "As illustrated in Fig.5, the RMSE and R2 results varied in relation to the JBARS tidal range (range being twice amplitude), with greater accuracy evident in predictions made using data derived from 25 hr periods when the tidal range was higher than average".

- versus –> and day –> days add respectively at end of sentence Response: These changes have been made. The text now reads: "Results revealed that the ADI are very similar, and there is <1 day AT difference, between the two stations. The ADI values were 0.57 and 0.23 or 0.30 days, while the AT values were -2.30 and -1.44 or -2.87 days, for ROBT and JBARS respectively (Table 1)".

229-230 - sentence 'Hence the' has no verb Response: This sentence has been amended. The text now reads: "This similarity explains why we found the CTSM+TCC method successful in generating JBARS tidal predictions, using concurrent 25 hr records from both stations and long-term reference records from ROBT".

- remove the minus sign. Replace the tropic jargon business. Response: The minus sign typo, and all mention of tropic to equatorial tides (TET) and of equatorial to tropic tides (ETT) has been deleted in the shortening of section 5.1 while brief mention has been made of lunar declination changes. The text of this section, for example, now reads: "However, as shown in Fig. 7, our results contain a changing fortnightly timescale bias in estimates... Comparisons between Run1 and Srun predictions show that exclusion of the Mf tide (2.7 cm amplitude) can produce prediction biases during periods of lunar declination change (Fig. 9a), with comparisons between Run2 and Run1 results showing that the additional exclusion of the MSf tide (1.2 cm amplitude) intensifies the biases (Fig. 9b)".

249-256 - I think I would replace this woffle with simply saying that good knowledge of tides is important for understanding ice shelf dynamics and give one reference as an example. Response: We have deleted most of this text, including the details of the methods used by different authors, in response to this comment and one in the Rowe review, leaving points that are of use for discussion later in the paper. The text now reads: "Rosier and Gudmundsson (2018) found that ice flows are modulated at various tidal frequencies, including that of the MSf tide... studies indicate that incorporating major and minor tidal constituents, including long period tides, into tidal predictions may be advantageous for their use in ice flow and ice-ocean front modelling specifically (e.g. Rignot et al., 2000; Rosier and Gudmundsson, 2018)".

- .. periods, rather than seasonal. (I think) Response: We were not referring to seasonal effects with our 'summertime' adjective but rather to the monitoring period. We have clarified the text by removing the adjective and, thus, any ambiguity created by its inclusion. The text now reads: "To verify the main cause of the apparent fortnightly prediction biases in results, in particular that in the 2019 predictions (Fig. 7b), we examined the effects of two fortnightly tidal constituents (Mf, and MSf) at ROBT".

- with additional exclusion (I think) Response: Yes, this has been modified as suggested. The text now reads: "Comparisons between Run1 and Srun predictions show that exclusion of the Mf tide (2.7 cm amplitude) can produce prediction biases during periods of lunar declination change (Fig. 9a), with comparisons between Run2 and Run1 results showing that the additional exclusion of the MSf tide (1.2 cm amplitude) intensifies the biases (Fig. 9b)".

- well, you don't do that do you?! You have spent a page showing that the method could be improved with a digression into the ice shelves. There is very little in this section (see above also). Response: This section has been significantly reduced in response to your comments and those of Rowe, including deletion of almost all the digression into ice shelves. Please see the new text early in this reply, and the response to Rowe regarding section 5.1.

- Decadal timescale . Response: The section 5.2 title has been changed to reflect the significantly truncated text and more focussed purpose of this section. The section title now reads: "Understanding the contrasting tidal environments around Antarctica". 274 - drop daily Response: This modification has been made. The text now reads: "Figure 11 illustrates the form factors of tidal regimes in the seas surrounding Antarctica, according to FES2014 model data".

- I don't understand this. The small magenta blob on the west coast of the Weddell Sea indicates a large (diurnal) form factor, right? Not semidiurnal. (You might also mention its latitude rather than 'half-way'). Most of the Weddell Sea is blue (semidiurnal). Response: This magenta patch represents an area where tides are characterised by semidiurnal form factors (<0.25). The rest of the Weddell Sea is characterised by 'mixed, mainly semidiurnal' tides (F between 0.25 and 1.5). We have amended our key (colour bar) to end at 0.25 to remove confusion regards classification of the majority area of the Weddell Sea. We have also added the latitude note, as suggested. See further below for the new figure caption. The paper text now reads: "Only in a small area half-way along the Weddell Sea coast of the Antarctic Peninsula (at 72°S) do tides exhibit a 'semidiurnal' form (F<0.25)".

- drop 'the increase in' 286 - drop 'feature ..tidal' which is just repetition. influences –> influence 292- 298 - see above. This is just an inevitable consequence of the way T-Tide is coded with the equilibrium nodal dependencies. Response: These sentences have been deleted in the shortening of section 5.2.

- Drop 'Of note', unless you want to refer to a tidal text book Response: These lines (formerly 292-304) have been deleted in response to your comments and to the Rowe review.

- drop database Response: This change has been made. The text now reads: "Details of the FES2014 tide model are found in Carrère et al. (2016) and via https://www.aviso.altimetry.fr/en/data/products/auxiliary-products/global-tide-fes.html".

Fig A1 caption - drop horizontal. co-amplitudes –> amplitudes. co-tides –> phase lags (Greenwich) In the caption of the 4 figures, remove the dot after deg as there is no dot after cm. remove all the co- things. And co-tide should be Greenwich phase lag. Figure A2 ditto the above. In (b) and (d) there is a mess of annotation of phase lags at a couple of amphidromic places. Please remove that mess. Response: Both figures (i.e. the entire original Appendix 1) have been deleted in response to the Rowe review comments.

Table 1. Please move the information in the Note column to be extra lines under ROBT etc. You give only one set of ADI and AT for JBARS but there must be two different sets of values in 2017 and 2019. day –> days. No hyphen in phase lag. Response: All of these changes have been made in Table 1. The new Table 1 will be included in the merged 'reply to all reviews' file, and its caption now reads: "Table 1. Major tidal harmonic results for diurnal and semidiurnal constituents from harmonic analyses of sea level observations: year-long (2013) records from Cape Roberts (ROBT), and 17.04 day records (29 January to 15 February 2017) and 20.54 day records (29 December 2018 to 18 January 2019) from Jang Bogo Antarctic Research Station (JBARS) in the Ross Sea (see source details in Sect. 2). For the JBARS tidal harmonic analyses, the inference method was applied to separate out the K1 (S2) and P1 (K2) tidal constituents, using inference parameters estimated from the ROBT 2013 harmonic analysis. Note that Amp. denotes amplitude; Pha. denotes phase lag, referenced to 0°, Greenwich; F is the ratio of the K1 and O1 diurnal tide amplitudes to the M2 and S2 semidiurnal tide amplitudes; and ADI and AT denote the age of diurnal inequality and the age of the tide".

Table 2. .. harmonic analysis of year-long .. No hyphen in phase lag Response: All of these changes have been made in Table 2, as well as throughout the paper. The new Table 2 will be included in the merged 'reply to all reviews' file, and its caption now reads: "Table 2. Harmonic constants for 6 long-period tidal constituents, derived from harmonic analyses of year-long observations (2013) measured at the Cape Roberts sea level gauge (ROBT), using T_Tide (Pawlowicz et al., 2002) Phase lags are referenced to 0°, Greenwich and SNR denotes the signal-to-noise ratios".

Figure 1 caption. Please say year and month this photo was taken Response: This has been added. The Fig. 1 caption now reads: "Figure 1. Drifting ice, including icebergs and mobile sea ice, around the Jang Bogo Antarctic Research Station (JBARS), photographed on 29 January 2017".

Figure 3 - y-axis phase lag should be (deg) and not (cm) Response: This has been fixed. Please see details of Figure 3 changes in the response to the Rowe review.

Figure 4 caption should say what (a), (b) etc. are and not just have text. Anyway I think the last two sentences contradict each other Response: These improvements have been made as indicated. Please see details of Figure 4 caption changes in the response to the Rowe review.

Figure 5 - under (b) you should have Time (month/day) as for Figure 6 I think the last line should say JBARS and ROBT Response: Both of these corrections have been made as indicated, and also this figure has been combined with the former Fig. 4 as its new panels e and f. Please see details of Figure 4 (formerly 5) changes in the response to the Rowe review.

Figure 6 - Time (day) should be Time (month/day). (a) and (b) are missing from the plots. Line 429 - (thick line with o) should have a filled and not open o to correspond to the plot Response: These errors have all been fixed. Please see details of Figure 5 (formerly 6) changes in the response to the Rowe review.

Figure 7 - why the == on the y-axis? There is no break in the numeration. Time (day) should be Time (month/day) Response: These errors have been fixed. Please see details of Figure 6 (formerly 7) changes in the response to the Rowe review.

Figure 8 - Time(month/day). A difference like this is usually defined as an Obs minus Pred but I guess it doesn't matter too much. The caption says 15 February, but the x-axis in (a) only goes up to 14 Feb. The caption should say what RMSE and R-squared are. Response: The x-axis label (month/day), and the caption (regarding 14 Feb) have been amended. RMSE and R2 definitions have been added to the caption. We have also altered this figure to show observations minus predictions, as suggested. Please see details of Figure 7 (formerly 8) changes in the response to the Rowe review.

Figure 9 - the caption and the x-axis in (a) say 15 Feb, but the header says 16 Feb In (b), the caption and x-axis say 30 Jan but the header says Jan 18. I thought at first you were referring to the dates of the dashed boxes but it seems not. line 1 of caption - estimated –> shown Response: The two header errors have been corrected. 'Estimated' has been replaced by 'calculated' in the caption. Please see details of Figure 8 (formerly 9) changes in the response to the Rowe review.

Figure 10 - Time (month/day) 450 - Msf and Mf tides ('Exp2'). At least I think that is what is meant. Response: The axis label and caption have been fixed as indicated.

Please see details of Figure 9 (formerly 10) changes in the response to the Rowe review.

Figure 11 - please have an arrow on the colour scale to indicate values over 3. The longitudes on the map are fuzzy. caption - drop horizontal. Response: Improvements in relation to all three points have been made. This figure has been uploaded and its full caption now reads: "Figure 11. Distribution of tidal form factor (F) values around Antarctica. Note the magenta area (72°S) on the Antarctic Peninsula's Weddell Sea coast denotes the only area of fully semidiurnal tides (F<0.25) in the Antarctic region".

Figure 12. What you are showing here are the 'f' and 'u' nodal factors. They are both nodal factors, not just 'f'. They are not 'estimated', they are hard coded into T-Tide and can be found in any tides text book. Response: This figure has been deleted due to the shortening/ tightening of focus of section 5.2.

So you can tell I found many small problems with the paper, in addition to the problems with the three sections mentioned above. I hope you can produce a considerable better (and probably shorter) version. Response: Thank you for your detailed review. We have made the changes suggested to deal with all of the small problems. We have also have taken on board your more major criticisms, and these have helped us to significantly improve the 3 sections you identified as problematic, with the result being a more focused and shorter paper.
* * *
(a) **Run1 (M$_f$)** vs. **SRun**

(b) **Run2 (M$_f$+MS$_f$)** vs. **SRun**

Run  Srun  Difference (Srun-Run)

**Fig. 1.** Figure 9. Time series of ROBT tidal predictions (a) made without long-period constituents ('SRun', i.e. excluding the constituents listed in Table 2) versus with the Mf tide ('Exp1'); and (b) time ser

[Figure]

**Fig. 2.** Figure 10. Time series (29 December 2018 to 18 January 2019) of (a) predictions of the diurnal (K1+O1) tides (blue line) and the semidiurnal (M2+S2) tides (magenta line) for JBARS; (b) their combined

![Figure showing distribution of tidal form factor values around Antarctica in polar projection]

**Fig. 3.** Figure 11. Distribution of tidal form factor (F) values around Antarctica. Note the magenta area (72°S) on the Antarctic Peninsula's Weddell Sea coast denotes the only area with a properly semidiurnal

---

## Author Comment (AC4) · 19 May 2020

Apologies - we mistakenly uploaded Figure 10 instead of Figure 9, in our reply to this Rowe review. The correct Figure 9 is uploaded here (and has also been uploaded in the correct order in our reply to the Editor's comments).

[Figure]

**(a) Run1 (M$_f$) vs. SRun**

**(b) Run2 (M$_f$+MS$_f$) vs. SRun**

Run        Srun        Difference (Srun-Run)

**Fig. 1.** Figure 9. Time series of ROBT tidal predictions (a) made without long-period constituents ('SRun', i.e. excluding the constituents listed in Table 2) versus with the Mf tide ('Exp1'); and (b) time ser

---

## Author Response (AR1)

Merged reply to three reviews of "Predicting tidal heights for extreme environments: From 25 h 1 observations to accurate predictions at Jang Bogo Antarctic Research Station, Ross Sea, Antarctica" 2 3 4 Do-Seong Byun1, Deirdre E. Hart2 1Ocean Research Division, Korea Hydrographic and Oceanographic Agency, Busan 49111, Republic of Korea 6 2School of Earth and Environment, University of Canterbury, Christchurch 8140, Aotearoa New Zealand 7 Correspondence to: Deirdre Hart (deirdre.hart@canterbury.ac.nz) 8 Format We are very grateful for the two reviewers' and Editor's reviews of our paper received. Collectively, these reviews have been useful in improving this paper. Below we reply to the reviews in chronological order, with each individual reviewer 9 10 comment copied in blue, a response written below it, and then the final modified text, table or figure copied below the response. 11 12 1. Reply to Reviewer 1's interactive comments of 17 Jan. 2020 13 14 p1, line35: Could you add these neighbouring sites to the map? And it would be good to find out what data is publicly available, and use them for further validation if possible. 15 16 Response: According to reviewer's comment, these sites have been added. Thank you for the suggestion regards 17 validation and other publically available records. Unfortunately it is relatively difficult to find recent online 18 available records, but we found mention of a 1 year record from McMurdo Station in a Padman et al. (2003) paper 19 and of a tide gauge being set up at Mario Zucchelli Station (formerly named Terra Nova Station) from 1996 (see 20 21 22 https://www.geoscience.scar.org/geodesy/perm\_ob/tide/terranova.htm). We will indeed attempt to track down these and any other available Ross Sea records for a further paper on the tides of this very interesting area. We have added these references to our paper so that out authors can see the data sources behind our comment.  $\begin{array}{c} 23\\ 24\\ 25\\ 26\\ 27\\ 28\\ 29\\ 30\\ 31\\ 32\\ 33\\ 34\\ 35\\ 36\\ 37\\ 38\\ 39\\ 40\\ 41\\ 42\\ 43\\ 44\\ 45\\ 46\\ 47\\ 48 \end{array}$ The paper text and reference list now read: "Long-term, quality sea level records in the Ross Sea are few and far between, and include observations from gauges operated by New Zealand at Cape Roberts (ROBT); by the United States in McMurdo Sound (see reference to data in Padman et al., 2003); and by Italy at Mario Zucchelli Station (Gandolfi, 1996), all in the eastern Ross Sea". Gandolfi, S.: Terra Nova Bay Permanent Tide Gauge Observatory Site, https://www.geoscience.scar.org/geodesy/perm\_ob/tide/terranova.htm, last access 4 Feb. 2020, 1996. Padman, L., Erofeeva, S. and Joughin, L.: Tides of the Ross Sea and Ross Ice Shelf cavity. Antarctic Science 15(1), 31-40, 2003. p4, line22: thanks for mentioning atmospheric conditions, too often ignored. Response: Yes, agreed. We have ensured that this point remains in the re-drafted methods section. This text still reads: "Byun and Hart (2015) recommended the use of short-term records gathered during periods of calm weather, to minimise errors due to atmospheric influences". p4, line148: you could mention somewhere here that bundling all the constituents in a species together is valid due to the "credo of smoothness" assumption. Response: Yes, according to your comment, this has been added. Please note that this and other calculation explaining method details have been shifted into a new Appendix 1. The paper now reads: "As the second step, under the 'credo of smoothness' assumption that the admittance or 'ratio of output to input' does not change significantly between constituents of the same species (Munk and Cartwright, 1966; Pugh and Woodworth, 2014), the amplitude ratio and phase lag difference of each representative tidal constituent for each tidal species between the temporary and reference stations were calculated from the results of tidal harmonic analyses of concurrent 25 hr data slices (starting at 00.00) from the temporary observation and reference tidal stations (i.e. from SHo and SHr)". p6, line206: In figure 6, it looks like the ADI is negative as the peak is before the max declination? 49 Response: Thank you for this query – upon checking, we found that location of symbols for Moon's maximum ( 50 51 and zero declination (**■**) was not correct. The Moon's maximum declination is 19:00 7/2/2017 (18.867°) and the zero declination is around 09:30 1/2/2017. We have now fixed these in the figure. 52 53 Figure 5 (formerly Fig. 6) now looks like:

66

67

87

55 Figure 5. (a) Time series (29 Jan. to 14 Feb. 2017) of Root Mean Square Errors (RMSE, thick blue line with •) and coefficients of 56 determination (R2, thin black line with  $\circ$ ) between JBARS 10 min interval sea level observations and the CTSM+TCC prediction 57 58 datasets, generated for this site using harmonic analysis results from the JBARS daily (25 hr) sea level data slices and concurrent daily (25 hr) 2017 tidal prediction data slices and harmonic analysis results from ROBT station's year-long (2017) tidal predictions. 59 (b) Time series of predicted 2017 tidal heights (thin blue line) and daily tidal ranges (thick black line with �) for ROBT, based on harmonic analysis of this station's 2013, 5 min interval sea level records, plus an indication of the moon's phase and declination. 60

**61 62 p7, line 251: (And elsewhere, please check all), Msf should be MSf [Moon-Sun-fortnight]. Similarly Msm should be MSm 63 [Moon-Sun-month]. 64**

- Response: Yes, these have both been fixed throughout.
- For example, the Sect. 5.1 text now reads: "Table 2 summarises the characteristics of 6 long-period tides (Sa, Ssa, MSm, Mm, Mf, MSf) at the ROBT station, derived from tidal harmonic analysis of year-long (2013) in situ observation records".

p7, line 270: Given MSf is important, I wonder if it might be worth including MS4? It might mop up the high frequency residual in figure 8. Worth checking the amplitude in the long record.

Response: Thank you for this suggestion - we checked the MS4 amplitude from the one year (2013) harmonic analysis results of ROBT. The amplitude was 0.69 cm, indicating that the MS4 tide is not a major constituent here.

p8, line 302: So the tides in the Ross Sea will be almost 1.5 times larger in 2025 than in 2016? I wonder how aware the ice modelling community are of this?

- 68 69 70 71 72 73 74 75 76 77 78 79 80 81 82 83 84 Response: Yes, it is interesting to consider. However, due to comments from the second reviewer and Editor, who pointed out that Sect. 5.2 contained a bit of a digression from the aim and topic focus of this paper, we have cut a lot of this detail from Sect. 5.2 including this the nodal factor discussion. But we have started drafting a new paper focused on exploring such features in the tides of the Ross and Weddell Seas, so the point is not lost but deferred to another piece of work.
- The shortened Section 5.2 text relating to this point now reads: "Since diurnal tides have larger nodal amplitude factor and nodal angle variations than semidiurnal tides (Pugh and Woodworth, 2014), areas like the Ross Sea will have larger variations in tidal height across the 18.61 year lunar nodal cycle compared to areas like the Weddell Sea". 85

86 fig 6: Is the split y axis really necessary here?

Response: We originally thought to employ a split y-axis scale in order to show as clearly as possible (magnify) the ٠

difference in RMSE results between Fig. 6(a) and Fig. 7. However, the effect of the split was a minor one, so we have changed these axes in line with your comment as you are right that it was not fully necessary. Please see above for the new Fig. 5 (formerly Fig.6), and below for the new Fig. 6 (formerly 7).

**• The new Fig. 6 (formerly 7) now looks like:**

Figure 6. Time series of Root Mean Square Errors (RMSE, thick blue line with ●) and coefficients of determination (R2, thin black
line with ○) between JBARS 10 min interval sea level observations (29 Dec. 2018 to 18 Jan. 2019) and the CTSM+TCC prediction
data sets generated for this site (using harmonic analysis results from daily (25 hr) summertime 2017 sea level data slices from
JBARS along with concurrent daily (25 hr) tidal prediction slices and harmonic analysis results from ROBT station's year-long
(2017) tidal predictions).

**100 Language:**

99

- I am particularly impressed by how clearly written this paper is I thank the authors for making the reviewing task easy. I
   wish I wrote as well!
- **Response:** Thank you we really appreciated this comment and hope that you find the revised paper clear to read.

**105 p1,line9: "Though" should be "However"**

- Response: This has been changed as suggested.
- The revised text now reads: "However obtaining long term sea level records for traditional tidal predictions is extremely difficult around ice affected coasts".

**110 p7 line 246: -tropic ?**

- Response: Thank you for spotting this -the hyphen had been misplaced. However, in response to comments from the Editor, we have removed this mention of tropic tides (including the hyphen) and replaced it with discussions of lunar declination.
- The relevant replacement text reads: However, as shown in Fig. 7, our results contain a changing fortnightly timescale bias in estimates... Comparisons between *Run1 and Srun* predictions show that exclusion of the Mf tide (2.7 cm amplitude) can produce prediction biases during periods of lunar declination change (Fig. 9a), with comparisons between *Run2 and Run1* results showing that the additional exclusion of the MSf tide (1.2 cm amplitude) intensifies the biases (Fig. 9b).

120 *p8 line 275: The abreviations DD etc aren't used again, delete.*

• **Response:** Yes, these have been deleted.

**123 References:**

119

- P&W 2014: Pugh, D.T. and Woodworth, P.L. 2014. Sea-level science : understanding tides, surges tsunamis and mean sea-level changes. Cambridge University Press https://doi.org/10.1080/00107514.2015.1005682 M&C 1966: Tidal spectroscopy and prediction, Walter Heinrich Munk and David Edgar Cartwright
- 127 https://doi.org/10.1098/rsta.1966.0024
- 128 *Oh, and you need to add doi to some of your other references!*
- **Response:** We have added these 2 references, and added reference doi numbers where missing elsewhere.
- 130 The reference list now contains:
- Munk, W. H. and Cartwright, D. E.: Tidal spectroscopy and prediction, Math. Phys. Sci., 259, 533-581,
   doi.org/10.1098/rsta.1966.0024, 1966.
- Pugh, D. T. and Woodworth, P. L.: Sea-level science: Understanding tides, surges, tsunamis and mean sea-level
   changes, Cambridge University Press, United Kingdom, doi.org/10.1080/00107514.2015.1005682, 2014.

**2. Reply to Glen Rowe Review 2 interactive comments of 14 Feb. 2020 135 136 137 Line 9: The words 'as represented' are unnecessary and at the start of the next sentence change Though to However 138 Response: Both of these wording changes have been made exactly as suggested. 139 The paper now reads: "Accurate tidal height data for the seas around Antarctica are much needed, given the crucial 140 role of these tides in regional and global ocean, marine cryosphere, and climate processes. However obtaining long 141 term sea level records for traditional tidal predictions is extremely difficult around ice affected coasts". 142 143 Line 20: This sentence could end at regimes as the following words repeat what has already been stated. Response: Upon reflection we decided that this sentence was not needed, since the previous sentence detailed the 144 145 level of success of the method, so this former line 20 sentence has been deleted. 146 The preceding sentence now reads: "Results reveal the CTSM+TCC method can produce accurate (to within ~5 cm Root Mean Square Errors) tidal predictions for JBARS when using short-term (25 hr) tidal data from periods with 147 148 higher than average tidal ranges (i.e. those at high lunar declination)". 149 150 Line 29: : : : based on as little as 25 h of sea level records when combined: : : Also, h, as used here and elsewhere in the 151 paper, would be clearer if abbreviated to hr (or better still, written in full). Response: Regarding the second point above, the unit for hour, 'h' has been changed to 'hr' throughout the 152 153 manuscript. Regarding the first point, the text has been altered as suggested (see below). 154 The paper now reads: "However, Byun and Hart (2015) developed a new approach to successfully predict tidal 155 heights based on as little as 25 hr of sea level records when combined with neighbouring reference site records, using 156 their Complete Tidal Species Modulation with Tidal Constant Corrections (CTSM+TCC) method, on the coasts of 157 Korea and New Zealand". 158 159 Line 35: I'm not aware of the US operating a gauge in McMurdo Sound and would be interested to know where/when. NZ 160 has a gauge at Scott Base. Does Italy have a long-term gauge at MZS? 161 Response: Padman et al. (2003) mentions a 1 year record from McMurdo Station. Also a tide gauge was set up at Mario Zucchelli Station (formerly named Terra Nova Station) from 1996 (see 162 163 https://www.geoscience.scar.org/geodesy/perm\_ob/tide/terranova.htm). We are currently attempting to track down 164 these and any other available Ross Sea records for a further paper on the tides of this very interesting area. We have added these references to our paper so that our readers can clearly see the data sources behind our comment. 165 166 The paper now reads (and includes the references below): "Long-term, quality sea level records in the Ross Sea 167 are few and far between, and include observations from gauges operated by New Zealand at Cape Roberts (ROBT); 168 by the United States in McMurdo Sound (see reference to data in Padman et al., 2003); and by Italy at Mario Zucchelli 169 Station (Gandolfi, 1996), all in the eastern Ross Sea". 170 171 Line 36: Only the Italian base is in Terra Nova Bay – the others aren't anywhere near this bay. 172 Response: Thank you - this error has now been corrected to 'eastern Ross Sea' (see full revised sentence above in 173 response to comment on line 35). 174 175 Line 37: There is also the problem of securing against damage any cable connection from a subsurface device to 176 datalogging/power equipment ashore. 177 **Response:** Yes, though this is a challenge for any cabled shoreline instrument deployed for a long time in any coastal environment, we can imagine that it is particularly difficult in the harsh environment of Antarctica. We have added 178 179 this issue to the text. 180 The paper now reads: "There is also the challenge of securing and preventing damage to the cables that join the 181 subsurface instruments to their onshore data loggers and power supplies, across the seasonally dynamic and harsh 182 coastal and subaerial environments of Antarctic shorelines". 183 184 Line 42: Of course, hydrographic surveys are ideally carried out when there is minimal sea ice; whether or not there is a permanent gauge site (line 40-41) is not the main factor when deciding when to conduct such surveys 185 186 Response: Yes- in order to better separate out these two pieces of information we have split the sentence into two. 187 The paper now reads: "In the absence of a suitable permanent gauge site, hydrographic surveys have been 188 conducted at the Korean Jang Bogo Antarctic Research Station (JBARS). Such surveys are best conducted during 189 the summertime predominantly sea ice free window around mid-January to mid-February" 190 191 Line 72: : : : in the austral summertime : : : 192 Response: Yes, the word austral has been added here, as well as in another place in the paper. 193 The paper now reads: "The Korea Hydrographic and Oceanographic Agency (KHOA) survey team went to JBARS 194 in Northern Victoria Land's Terra Nova Bay, Ross Sea, Antarctica, in the austral summertime of 2017 (Fig. 2) for a 195 preliminary fieldtrip to conduct hydrographic surveys and produce a nautical chart". 196 197 Line 81: Residuals – observed compared to predicted?**

**Response: Yes, that's correct. We added the text in brackets below. The paper now reads: "Of these, the 20.54 day record produced be**

• **The paper now reads:** "Of these, the 20.54 day record produced between 29 Dec. 2018 and 18 Jan. 2019 comprised relatively high quality data with small residuals (i.e. observations minus predictions)".

**202 Line 83: : : : the absence of a permanent tide station at JBARS, : : :**

203 • Response: The text has been altered as suggested.
204 • The text now reads: "Due to the short duration of

200

201

205

206

207

209

210

211

228

229

230

231

236

237

238

241

242 243

244

245

248

The text now reads: "Due to the short duration of the KHOA survey team's forays into the Ross Sea, and in the
absence of a permanent tide station at JBARS, it was not possible to collect the year-long sea level records that are
commonly employed to obtain reliable tidal harmonic constants for tidal prediction".

208 Line 94: Pairs in brackets unnecessary repetition from lines 92 and 93.

- Response: Your comment alerted up to the wordy nature of these two sentences, so instead of just deleting the pairs
  in brackets we rewrote both sentences as one replacement sentence, shortening our explanation of this step while
  retaining the key details and only once stating the pairs in brackets.
- The text now reads: "Also the inference method was used to separate out neighbouring diurnal (K1 and P1) and semidiurnal (S2 and K2) tide constituents, with their amplitude ratios and phase lag differences obtained from harmonic analysis of the long-term ROBT reference station records".

**Lines 96 – 98: As Table 1 will be inserted here this sentence is redundant as it is just repeating what the table contains. Line 100: : : : phase lags showed only slightly different values.**

- Response: With regard to your line 96-98 comment, we deleted the text that unnecessarily highlighted the numbers displayed in Table 1, and in just kept the interpretive text found it best to merge two sentences together for tighter expression of the results. According to your line 100 comment we removed the hyphen from 'phase lag' throughout the entire paper the below sentence provides an example.
- The text now reads: "Analyses revealed that the two main diurnal (O1 and K1) and semidiurnal (M2 and S2) tides had similar amplitudes at the two stations, with the diurnal (semidiurnal) amplitudes being slightly larger (smaller) at ROBT than at JBARS, and the phase lags of all four tides having only slightly different values. The amplitude differences result in slightly different tidal form factors at the two sites (e.g., *F* in Table 1)".

**226 227 Line 101: for completeness, should the formula for F be stated?**

- **Response:** Yes, agreed we have now added explanation of this parameter to Table 1 caption, where it is now mentioned first in the paper.
- The Table 1 note now includes: "F is the amplitude ratio of the  $(K_1 + O_1)/(M_2 + S_2)$  tides".

Lines 103 – 111: Is this paragraph necessary? This study relates to a part of the Ross Sea – the tidal regimes around other parts of Antarctica are of no relevance to this investigation. Or maybe you are hinting that as the Ross Sea is different to the rest of the continent the results of this study may not be applicable elsewhere. If this paragraph is deleted then Figures A1 and A2 are no longer required.

• **Response:** According to your suggestion (and comments by the Editor) regards these lines, the whole paragraph has been deleted, as have the former Appendix 1 figures.

**239 Line 113: Delete the '-' in front of CTSM. 240 • Response: This typo has been of**

- Response: This typo has been deleted, and the sentence has been altered significantly as a result of the Editor's
  suggestion that section 3 should be rewritten to describe the methodology more simply, removing much of the math.
- The text now reads: "Having analysed the tidal harmonic constants at the two stations, we then employed the CTSM+TCC method (Byun and Hart, 2015) to generate tidal height predictions for JBARS, our 'temporary' tidal observation station (subscript *o*), using ROBT as the 'reference' station (subscript *r*)".

**246 Lines 114 – 115: Are the italics necessary? 247 • Response: No they were unnecess**

Response: No they were unnecessary so have been removed in accordance with your comment. This sentence has
also been modified as a result of the section 3 rewrite.

**• The text now reads:**

- 250 "This prediction approach (see Appendix 1 for the detailed calculations, and Byun and Hart (2015) for explanation of procedure
  251 development) is based on:
  252 (i) using long-term (1 year, in our case) reference station records (LHr) and CTSM calculations to make an initial anytime
- (i) using long-term (1 year, in our case) reference station records (LHr) and CTSM calculations to make an initial anytime ( $\tau$ ) tidal prediction ( $\eta_r(\tau)$ ), which involves summing tidal species' heights for the reference station (Fig.3);
- (ii) comparing the tidal harmonic constants (amplitude ratios and phase lag differences) of representative tidal constituents (e.g.,  $M_2$  and  $K_1$ ) for each tidal species between the temporary and reference stations, calculated using T\_TIDE and concurrent short-term records ( $\geq$ 25 hr duration, starting at midnight) from the temporary (SHo) and reference (SHr) stations; and
- 258 (iii) using the step (ii) comparative data and the TCC calculations for each tidal species to adjust the  $\eta_r(\tau)$  tidal species' 259 heights in order to generate accurate, anytime tidal height predictions for the temporary tidal station ( $\eta_o(\tau)$ )".
- 260

**261 Line 116: Similar tidal characteristics at the reference and temporary site is given as one of the requirements of the CTSM+TCC method. However, it has been noted in lines 101 - 102 that ROBT is diurnal and JBARS is mixed, mainly 262**

263 diurnal. Are these regimes sufficiently alike to be considered 'similar' for the purposes of this method?

264 Response: Thank you, due to your question we have improved the text to really hone in on the similarity required. 265 • The text now reads: "Importantly, this method assumes that the reference and temporary tidal stations are situated 266 in neighbouring regimes with similar dominant tidal constituent and tidal species characteristics, and that the tidal 267 properties between the two stations remain similar through time. As explained above, both JBARS and ROBT have 268 tidal regimes that are primarily dominated by diurnal tides. LHr must comprise high quality (e.g. few missing data) 269 tidal height observations from anytime" 270

**271 Lines 121 – 122: The records are not temporary – the records are from a temporary site.**

272 Response: Yes, thank you. We have made sure that this word placement mistake does not now occur in our paper. 273 This particular sentence has also been deleted as part of the Sect. 3 rewrite, recommended by the Editor. 274

**275 Line 124: My record from ROBT does not have any gaps early February 2017.**

- 276 Response: ROBT data were downloaded from LINZ website. There are still no data files until 12 February 2017 as 277 you can see at http://apps.linz.govt.nz/ftp/sea\_level\_data/ROBT/2017/00/ (last access: 29 February, 2020). We have, 278 however, now received a file containing the full 2017 records, after finding out that they existed when consulting you 279 with regards to the ROBT set up by telephone - thank you very much for supplying these excellent data.
- 280 Please note that these data are not available on the Permanent Service for Mean Sea Level (PSMSL) website, where 281 ROBT records are recorded as existing up until 2009. We have found discovering the existence of, and then obtaining, 282 good observational tidal data for the Ross Sea and elsewhere in Antarctic quite a challenging exercise. Since your 283 LINZ records represent one of the best in existence, it might benefit Antarctic tide research to update the PSMSL 284 website: https://www.psmsl.org/data/obtaining/stations/1763.php, including the comments made there on the low 285 data quality of recent ROBT records. Currently this website says: "Documentation added 2011-11-17. There is no data available for 2010. Although the site is still working the data is of low quality and therefore unreliable. Plans 286 287 are in place to repair the tide gauge when possible" 288 We have re-written this sentence as a part of our Section 3 re-write.
- 289 The text now reads: "This adjustment in approach arose since for the 2017 JBARS observation time period, the concurrent 2017 ROBT records available online (LINZ, 2019) had multiple missing data". 290

**291 292 Lines 127 – 129: This sentence reiterates the essence of the preceding sentence and, although it begins 'In short', is longer 293 than the previous one. One of these two sentences could be deleted.**

- 294 Response: In our re-write of Section 3 we deleted the last of these two sentences as suggested here. 295
  - The remaining sentence reads: "We solved this issue by producing a year-long synthetic 2017 record for ROBT
- 296 using T\_TIDE (Pawlowicz et al., 2002) and the 2013 (i.e. LHr) observational record as input data". 297

**298 Lines 148 – 154: Is the first sentence in this block of lines necessary? The following two sentences describe the process and 299 can stand on their own.**

- 300 Response: This section of text has now been cut and pasted into an appendix detailing the maths behind the 301 CTSM+TCC approach (in response to a suggestion by the Editor to rewrite Section 3 more clearly and simply). In its 302 new Appendix 1, the first sentence in this block has been modified to convey different/ extra information according 303 to a comment by Reviewer 1, and terms that were repeated in the next two sentences have been deleted, eliminating 304 the overlap that you drew our attention to with this comment.
- 305 The text now reads: "As the second step, under the 'credo of smoothness' assumption that the admittance or 'ratio 306 of output to input' does not change significantly between constituents of the same species (Munk and Cartwright, 307 1966; Pugh and Woodworth, 2014), the amplitude ratio and phase lag difference of each representative tidal 308 constituent for each tidal species between the temporary and reference stations were calculated from the results of 309 tidal harmonic analyses of concurrent 25 hr data slices (starting at 00.00) from the temporary observation and 310 reference tidal stations (i.e. from SHo and SHr). The process of selecting the optimal 25 hr window for the concurrent 311 data slices from amongst the 17.04 days of available records is explained in Sect. 3".

313 Line 154: Which are the 'initial tidal predictions'? It is not clear to me.

- 314 Response: Yes, this was not as clear as it could've been - we meant 'tidal predictions at the reference station' 315 calculated from the CTSM, and have improved the text accordingly.
- 316 The text (cut and pasted into in Appendix 1) now reads: "Once this 2017 window was selected, the third step 317 involved adjusting the tidal predictions at the reference station calculated from Eq. (A1), to represent those for the

temporary station  $(\eta_o(\tau))$ , by substituting the daily (i.e. SHo and SHr) amplitude ratios  $\left(\frac{a_o^{(5)}}{a_o^{(5)}}\right)$  and phase lag differences 318

 $(G_0^{(s)} - G_r^{(s)})$  for the tidal constituents (K1 and M2) representing the diurnal and semidiurnal tidal species between 319 320 the temporary and reference stations into Eq. (A1) as follows ...". 321

322 Line 163: Calculations, not experiments?

**323 Line 164: 'in shorthand' seems unnecessary.**

| 525        | Line 104. In shormana seems unnecessary.                                                                                                                                                                                                                                                                                                                                                                                                                                                                                                                                                                                                                                                                                                                                                                                                                                                                                                                                                                                                                                                                                                                                                                                                                                                                                                                                                                                                                                                                                                                                                                                                                                                                                                                                                                                                                                                                                                                                                                                                                                                                                         |
|------------|----------------------------------------------------------------------------------------------------------------------------------------------------------------------------------------------------------------------------------------------------------------------------------------------------------------------------------------------------------------------------------------------------------------------------------------------------------------------------------------------------------------------------------------------------------------------------------------------------------------------------------------------------------------------------------------------------------------------------------------------------------------------------------------------------------------------------------------------------------------------------------------------------------------------------------------------------------------------------------------------------------------------------------------------------------------------------------------------------------------------------------------------------------------------------------------------------------------------------------------------------------------------------------------------------------------------------------------------------------------------------------------------------------------------------------------------------------------------------------------------------------------------------------------------------------------------------------------------------------------------------------------------------------------------------------------------------------------------------------------------------------------------------------------------------------------------------------------------------------------------------------------------------------------------------------------------------------------------------------------------------------------------------------------------------------------------------------------------------------------------------------|
| 324        | • Response: Yes to both of these suggestions - 'experiments' has been removed in the re-write of this text and we now                                                                                                                                                                                                                                                                                                                                                                                                                                                                                                                                                                                                                                                                                                                                                                                                                                                                                                                                                                                                                                                                                                                                                                                                                                                                                                                                                                                                                                                                                                                                                                                                                                                                                                                                                                                                                                                                                                                                                                                                            |
| 325        | describe these as 'prediction data sets', as opposed to experiments, at the end of the revised Sect. 3. We also removed                                                                                                                                                                                                                                                                                                                                                                                                                                                                                                                                                                                                                                                                                                                                                                                                                                                                                                                                                                                                                                                                                                                                                                                                                                                                                                                                                                                                                                                                                                                                                                                                                                                                                                                                                                                                                                                                                                                                                                                                          |
| 326        | the 'in shorthand' text.                                                                                                                                                                                                                                                                                                                                                                                                                                                                                                                                                                                                                                                                                                                                                                                                                                                                                                                                                                                                                                                                                                                                                                                                                                                                                                                                                                                                                                                                                                                                                                                                                                                                                                                                                                                                                                                                                                                                                                                                                                                                                                         |
| 327        | • The text now reads: "Each paired data set was then used with LH r to generate tidal height predictions for JBARS                                                                                                                                                                                                                                                                                                                                                                                                                                                                                                                                                                                                                                                                                                                                                                                                                                                                                                                                                                                                                                                                                                                                                                                                                                                                                                                                                                                                                                                                                                                                                                                                                                                                                                                                                                                                                                                                                                                                                                                                    |
| 328        | covering both the 2017 and 2019 KHOA observation campaign time periods. Comparisons were made between the                                                                                                                                                                                                                                                                                                                                                                                                                                                                                                                                                                                                                                                                                                                                                                                                                                                                                                                                                                                                                                                                                                                                                                                                                                                                                                                                                                                                                                                                                                                                                                                                                                                                                                                                                                                                                                                                                                                                                                                                                        |
| 329        | JBARS observations and the 17 prediction data sets generated for each campaign to identify which 25 hr short-term                                                                                                                                                                                                                                                                                                                                                                                                                                                                                                                                                                                                                                                                                                                                                                                                                                                                                                                                                                                                                                                                                                                                                                                                                                                                                                                                                                                                                                                                                                                                                                                                                                                                                                                                                                                                                                                                                                                                                                                                                |
| 330        | data window produced optimal $\eta_o(\tau)$ results".                                                                                                                                                                                                                                                                                                                                                                                                                                                                                                                                                                                                                                                                                                                                                                                                                                                                                                                                                                                                                                                                                                                                                                                                                                                                                                                                                                                                                                                                                                                                                                                                                                                                                                                                                                                                                                                                                                                                                                                                                                                                            |
| 331        |                                                                                                                                                                                                                                                                                                                                                                                                                                                                                                                                                                                                                                                                                                                                                                                                                                                                                                                                                                                                                                                                                                                                                                                                                                                                                                                                                                                                                                                                                                                                                                                                                                                                                                                                                                                                                                                                                                                                                                                                                                                                                                                                  |
| 332        | Lines 169 - 171: I had to read the first part of this sentence a few times to figure out what is going on. My take is that you                                                                                                                                                                                                                                                                                                                                                                                                                                                                                                                                                                                                                                                                                                                                                                                                                                                                                                                                                                                                                                                                                                                                                                                                                                                                                                                                                                                                                                                                                                                                                                                                                                                                                                                                                                                                                                                                                                                                                                                                   |
| 333        | obtained 17 datasets each one of which included 10-minute interval predictions spanning 17 days as derived from the                                                                                                                                                                                                                                                                                                                                                                                                                                                                                                                                                                                                                                                                                                                                                                                                                                                                                                                                                                                                                                                                                                                                                                                                                                                                                                                                                                                                                                                                                                                                                                                                                                                                                                                                                                                                                                                                                                                                                                                                              |
| 334        | harmonic analysis of each of the (17 in total) 25 hr slices of observed data. Is this correct? If not then I have clearly                                                                                                                                                                                                                                                                                                                                                                                                                                                                                                                                                                                                                                                                                                                                                                                                                                                                                                                                                                                                                                                                                                                                                                                                                                                                                                                                                                                                                                                                                                                                                                                                                                                                                                                                                                                                                                                                                                                                                                                                        |
| 335        | misunderstood, and if it is then that is good but, regardless, I'm not confident that I have it right.                                                                                                                                                                                                                                                                                                                                                                                                                                                                                                                                                                                                                                                                                                                                                                                                                                                                                                                                                                                                                                                                                                                                                                                                                                                                                                                                                                                                                                                                                                                                                                                                                                                                                                                                                                                                                                                                                                                                                                                                                           |
| 336        | Response: Yes, that is correct and thank you for pointing out the difficulty of this sentence. This sentence has been                                                                                                                                                                                                                                                                                                                                                                                                                                                                                                                                                                                                                                                                                                                                                                                                                                                                                                                                                                                                                                                                                                                                                                                                                                                                                                                                                                                                                                                                                                                                                                                                                                                                                                                                                                                                                                                                                                                                                                                                            |
| 337        | re-written. Moreover the previous Section 3 description of the method applied has been improved significantly such                                                                                                                                                                                                                                                                                                                                                                                                                                                                                                                                                                                                                                                                                                                                                                                                                                                                                                                                                                                                                                                                                                                                                                                                                                                                                                                                                                                                                                                                                                                                                                                                                                                                                                                                                                                                                                                                                                                                                                                                               |
| 338        | that we anticipate readers will be much clearer by the time they reach Sect. 4 about what we mean here.                                                                                                                                                                                                                                                                                                                                                                                                                                                                                                                                                                                                                                                                                                                                                                                                                                                                                                                                                                                                                                                                                                                                                                                                                                                                                                                                                                                                                                                                                                                                                                                                                                                                                                                                                                                                                                                                                                                                                                                                                          |
| 339        | The text now reads: CTSM+TCC was used to produce 17 different JBARS tidal prediction datasets for the period                                                                                                                                                                                                                                                                                                                                                                                                                                                                                                                                                                                                                                                                                                                                                                                                                                                                                                                                                                                                                                                                                                                                                                                                                                                                                                                                                                                                                                                                                                                                                                                                                                                                                                                                                                                                                                                                                                                                                                                                                     |
| 340        | 29 Jan. to 14 Feb. 2017, based on harmonic analysis results of the 'daily' (25 hr) $K_1$ and $M_2$ amplitudes and phase                                                                                                                                                                                                                                                                                                                                                                                                                                                                                                                                                                                                                                                                                                                                                                                                                                                                                                                                                                                                                                                                                                                                                                                                                                                                                                                                                                                                                                                                                                                                                                                                                                                                                                                                                                                                                                                                                                                                                                                                          |
| 341        | lags at our two tidal observation stations (Fig. 4)".                                                                                                                                                                                                                                                                                                                                                                                                                                                                                                                                                                                                                                                                                                                                                                                                                                                                                                                                                                                                                                                                                                                                                                                                                                                                                                                                                                                                                                                                                                                                                                                                                                                                                                                                                                                                                                                                                                                                                                                                                                                                            |
| 342        |                                                                                                                                                                                                                                                                                                                                                                                                                                                                                                                                                                                                                                                                                                                                                                                                                                                                                                                                                                                                                                                                                                                                                                                                                                                                                                                                                                                                                                                                                                                                                                                                                                                                                                                                                                                                                                                                                                                                                                                                                                                                                                                                  |
| 343        | Lines $177 - 187$ : This discussion about the correlation of tidal range and RMSEs and $R^2$ values is more difficult to follow                                                                                                                                                                                                                                                                                                                                                                                                                                                                                                                                                                                                                                                                                                                                                                                                                                                                                                                                                                                                                                                                                                                                                                                                                                                                                                                                                                                                                                                                                                                                                                                                                                                                                                                                                                                                                                                                                                                                                                                                  |
| 344        | than it could be. I feel the two sentences about the February 2 tide 'sandwiched' between the discussions about the results at                                                                                                                                                                                                                                                                                                                                                                                                                                                                                                                                                                                                                                                                                                                                                                                                                                                                                                                                                                                                                                                                                                                                                                                                                                                                                                                                                                                                                                                                                                                                                                                                                                                                                                                                                                                                                                                                                                                                                                                                   |
| 345        | greater tidal ranges has made the explanation somewhat convoluted. Dealing with the circumstances of the good statistics                                                                                                                                                                                                                                                                                                                                                                                                                                                                                                                                                                                                                                                                                                                                                                                                                                                                                                                                                                                                                                                                                                                                                                                                                                                                                                                                                                                                                                                                                                                                                                                                                                                                                                                                                                                                                                                                                                                                                                                                         |
| 346        | before moving on to the poorer results will enable this discussion to be expressed in a more succinct manner (and easier to                                                                                                                                                                                                                                                                                                                                                                                                                                                                                                                                                                                                                                                                                                                                                                                                                                                                                                                                                                                                                                                                                                                                                                                                                                                                                                                                                                                                                                                                                                                                                                                                                                                                                                                                                                                                                                                                                                                                                                                                      |
| 347        | follow).                                                                                                                                                                                                                                                                                                                                                                                                                                                                                                                                                                                                                                                                                                                                                                                                                                                                                                                                                                                                                                                                                                                                                                                                                                                                                                                                                                                                                                                                                                                                                                                                                                                                                                                                                                                                                                                                                                                                                                                                                                                                                                                         |
| 348        | • Response: Yes, agreed. We have reordered the text according to your nelprul comment here.                                                                                                                                                                                                                                                                                                                                                                                                                                                                                                                                                                                                                                                                                                                                                                                                                                                                                                                                                                                                                                                                                                                                                                                                                                                                                                                                                                                                                                                                                                                                                                                                                                                                                                                                                                                                                                                                                                                                                                                                                               |
| 349        | • The text now reads: "RMSEs between observations and predictions ranged from 4.26 cm to 20.56 cm, while R 2                                                                                                                                                                                                                                                                                                                                                                                                                                                                                                                                                                                                                                                                                                                                                                                                                                                                                                                                                                                                                                                                                                                                                                                                                                                                                                                                                                                                                                                                                                                                                                                                                                                                                                                                                                                                                                                                                                                                                                                                          |
| 350        | varied from 0 to 0.94, across the 1/ daily experiments. Eleven of the experiments produced accurate results (i.e.                                                                                                                                                                                                                                                                                                                                                                                                                                                                                                                                                                                                                                                                                                                                                                                                                                                                                                                                                                                                                                                                                                                                                                                                                                                                                                                                                                                                                                                                                                                                                                                                                                                                                                                                                                                                                                                                                                                                                                                                                |
| 351        | excluding mose derived from 51 Jan; and 1 to 4 and 14 rep. data slices). Daily datasets from periods with relatively
high side data served in the server of |
| 252        | mgn udar ranges (>>>.3 cm) produced predictions with RMSEs $< 5$ cm and K 2 values >0.92. The maximum spring                                                                                                                                                                                                                                                                                                                                                                                                                                                                                                                                                                                                                                                                                                                                                                                                                                                                                                                                                                                                                                                                                                                                                                                                                                                                                                                                                                                                                                                                                                                                                                                                                                                                                                                                                                                                                                                                                                                                                                                                          |
| 333
254 | use range occurred on 9 red., we data snees nom this date produced predictions with a low (but not the lowest)
$PMSE$ (4.81 cm). The predictions with the lowest $PMSE$ (4.250 cm) and bickets $P^2$ value (0.041) were preduced                                                                                                                                                                                                                                                                                                                                                                                                                                                                                                                                                                                                                                                                                                                                                                                                                                                                                                                                                                                                                                                                                                                                                                                                                                                                                                                                                                                                                                                                                                                                                                                                                                                                                                                                                                                                                                                                                              |
| 255        | KNDD (4.6) cm). The predictions with the lowest KNDD (4.25) cm) and ingnest K Value (0.341) were produced                                                                                                                                                                                                                                                                                                                                                                                                                                                                                                                                                                                                                                                                                                                                                                                                                                                                                                                                                                                                                                                                                                                                                                                                                                                                                                                                                                                                                                                                                                                                                                                                                                                                                                                                                                                                                                                                                                                                                                                                                        |
| 356        | using data sinces from one day earlier, $\sigma = 0.2017$ . In contrast to the majority of successful experiments, the avaparimetrized transition of the 2 Eq. (20.56)                                                                                                                                                                                                                                                                                                                                                                                                                                                                                                                                                                                                                                                                                                                                                                                                                                                                                                                                                                                                                                                                                                                                                                                                                                                                                                                                                                                                                                                                                                                                                                                                                                                                                                                                                                                                                                                                                                                                                           |
| 357        | $c_{1}$ and $v_{2}$ vary law $P_{2}^{2}(0,0)$ values The 2 Feb 2017 ides were observed ratio when very law $P_{2}^{2}(0,0)$ values The 2 Feb 2017 ides were observed ratio when the smallest ideal range (11.95 cm)                                                                                                                                                                                                                                                                                                                                                                                                                                                                                                                                                                                                                                                                                                                                                                                                                                                                                                                                                                                                                                                                                                                                                                                                                                                                                                                                                                                                                                                                                                                                                                                                                                                                                                                                                                                                                                                                                                              |
| 358        | of the IRARS record during a period of low lunar declination"                                                                                                                                                                                                                                                                                                                                                                                                                                                                                                                                                                                                                                                                                                                                                                                                                                                                                                                                                                                                                                                                                                                                                                                                                                                                                                                                                                                                                                                                                                                                                                                                                                                                                                                                                                                                                                                                                                                                                                                                                                                                    |
| 220        | of the particle reported during a portion of to the future dovining to the second se                                                                                                                                                                                                                                                                                                                                                                                                                                                                                                                                                                                                                                                                                                                                                                                                                                                                                                                                                                                                                                                                                                                                  |

360 Lines 188 – 192: Are these two sentences saying the same thing in different ways?

- **Response:** They concern the same idea, but the second sentence details the idea for a specific example (Fig. 7) amongst the total 17 cases (Fig. 6). We have adjusted the text and added "For example" to distinguish these sentences.
  - amongst the total 17 cases (Fig. 6). We have adjusted the text and added "For example" to distinguish these sentences.
    The text now reads: "As with the 2017 predictions, RMSEs between the 2019 predictions and observations were lower when generated using data slices from 2017 periods at high lunar declination (Fig.6). For example, 2019 predictions made using input data derived from the 8 Feb. 2017 data slices produced the lowest RMSE (5.3 cm) and highest R2 (0.913) values of the 2019 experiments (Fig. 7)".

368 Lines 208, 209, 211, 212 and 213: I find the use of the adjectives 'maximum' and 'minimum' in association with declination 369 to be confusing. Minimum could be taken to be on the celestial equator ( $\delta = 0^{\circ}$ ) and maximum could be greatest declination 370 either north or south. Better to use phrases like 'greatest southern declination' and 'greatest northern declination' to be 371 more specific.

- **Response:** Thank you for your useful suggestion we have applied this change as recommended.
- The text now reads: "That is, maximum tidal range days can be estimated for JBARS based on the day of the Moon's greatest northern (GN) and southern (GS) declinations. The time between the Moon's semi-monthly GN and GS declinations and their effects on tidal range, called the age of diurnal inequality (*ADI*), is commonly 1 to 2 days. As shown in Fig. 8, the GN and GS lunar declinations during our temporary station summertime observation periods occurred on 8 Feb. 2017 (GN) and on 6 Jan. 2019 (GS) respectively, with the maximum diurnal tides at JBARS expected around 1 day after each lunar declination peak".

**380 Line 227: Delete 'and'**

359

363

364

365

366 367

- Response: Yes, this typo has been removed. We also removed the numbers from this sentence and left only their interpretation, since the numbers will appear in Table 1, which is cited here. This latter adjustment we thought to do based on a your 'redundancy' point in the comment on lines 96-98 above.
- **The text now reads:** "Results revealed that the *ADI* are very similar, and there is <1 day *AT* difference, between ROBT and JBARS (Table 1), indicating that the tidal characteristics of the representative tidal constituents for each species between the two stations are very similar, in particular the dominant diurnal species".

| 387 |                                                                                                                                                                                                                                 |
|-----|---------------------------------------------------------------------------------------------------------------------------------------------------------------------------------------------------------------------------------|
| 388 | Lines 245 – 246: It would be helpful to give the dates for the two periods (ETT and TET).                                                                                                                                |
| 389 | Is the 'minus' in front of tropic on line 246 a typo or does it mean the southernmost declination?                                                                                                                              |
| 390 | Line 247: : : : CTSM+TCC considering only 2 major tidal species : : :                                                                                                                                                           |
| 391 | • Response: In response to the suggestion from the Editor that we significantly shorten section 5.1 (he suggested 5-6                                                                                                    |
| 392 | lines instead of 39 lines) we have deleted much of this detail (including mention of ETT and TET and the sentence                                                                                                               |
| 393 | with the minus sign typo you mentioned) We also made the '2 tidal major species' change suggested above                                                                                                                         |
| 30/ | • The fact new reads: "Howaver as shown in Fig. 7 our results contain a changing forthold the messale bigs in                                                                                                                   |
| 205 | • The text now reads. However, as shown in Fig. 7, our results contain a changing formignity unlescale of as in estimates. This area particular birdly experied from our application of CTSM TCC considering only 2 mains tidal |
| 206 | species (diversal and somilarmal) visited from our appreador of CFSM+FCC considering only 2 major tuda                                                                                                                          |
| 207 | species (durinal and semidurinal) whist ignoring several long period and small amplitude short period dues .                                                                                                                    |
| 200 | Lines 240 256. Could this he showed to just summarize the conclusion arrived at hu the other subcars. In these s need to                                                                                                        |
| 200 | Lines 249 – 250: Collia Inis de snorienea lo jusi summarise ine conclusion arrivea al dy ine other autors, is inere a need to                                                                                                   |
| 399 | aescribe what mey all – people interested can refer to the references.                                                                                                                                                          |
| 400 | • Response: We have removed most of the text explaining details and just left their findings that focus on what other                                                                                                    |
| 401 | constituents might be important. The remaining text has also been shifted slightly within the section.                                                                                                                          |
| 402 | • The text now reads: "Rosier and Gudmundsson (2018) found that ice flows are modulated at various tidal                                                                                                                        |
| 403 | trequencies, including that of the MS f tide. However, because these tides' amplitudes have small signal-to-noise ratios                                                                                             |
| 404 | (<1) with large standard errors (Table 2), caution should be exercised when elucidating fortnightly tide effects using                                                                                                          |
| 405 | these constituents. Nevertheless, studies indicate that incorporating major and minor tidal constituents, including long                                                                                                        |
| 406 | period tides, into tidal predictions may be advantageous for their use in ice flow and ice-ocean front modelling                                                                                                                |
| 407 | specifically (e.g. Rignot et al., 2000; Rosier and Gudmundsson, 2018)".                                                                                                                                                         |
| 408 |                                                                                                                                                                                                                                 |
| 409 | Lines 267 - 268: Srun excluded : : : Run1 excluded : : : Run2 incorporated : : : (I think)                                                                                                                                      |
| 410 | • Response: We have clarified this text as suggested.                                                                                                                                                                    |
| 411 | • The text now reads: "Three 2019 tidal prediction experiments were conducted:                                                                                                                                                  |
| 412 | • Srun excluded all long-period tides (see list of exclusions in Table 2);                                                                                                                                                      |
| 413 | • Run1 was based on Srun but also incorporated the M f , and                                                                                                                                                         |
| 414 | • Run2 was based on Srun but also incorporated the Mr and MS?                                                                                                                                                                   |
| 415 |                                                                                                                                                                                                                                 |
| 416 | Lines 269 and 270: Should both instances if 'exclusion' be 'inclusion'?                                                                                                                                                         |
| 417 | Response: No 'exclusion' is correct. We have reworded this part to avoid this confusion                                                                                                                                         |
| 110 | • The toxy new meets "Comparisons between burned and and man predictions show that avaluation of the Metide (2.7 cm                                                                                                             |
| 410 | • The text how reads. Comparisons between Kun1 and stud predictions show in a exclusion of the Wir doe (2.7 cm                                                                                                           |
| 420 | hatman pund and pund results showing that the additional avalusion of the MS, tide (12, 2d), will comparison                                                                                                      |
| 420 | the bigges (Fig. 0b)"                                                                                                                                                                                                           |
| 421 |                                                                                                                                                                                                                                 |
| 422 | Lines 270 271. Is there any reason why this suggested line of investigation has not have numered in this surgest                                                                                                                |
| 423 | Lines 2/0 - 2/1. Is mere any reason why his suggestee time of investigation has not been pursuea in this paper?                                                                                                                 |
| 424 | • Response: res, pasically, this is because the tidal constants for the long-period tides cannot be derived from short-                                                                                                  |
| 425 | term (25 nr) records, so it is beyond the scope of the present study, which was an initial assessment if the CTSM+TCC                                                                                                           |
| 426 | memory tudal station in an extreme environment with                                                                                                                                                                             |
| 427 | imperfect data record conditions. Now that we have demonstrated the usefulness of the method for making reasonable                                                                                                              |
| 428 | predictions here, we feel that further work could be done to hone the prediction approach for ice affected coasts if the                                                                                                        |
| 429 | data is to be used in detailed ice flow modelling. Generating data for ice flow modelling was not the primary focus of                                                                                                          |
| 430 | our paper, as this was an initial paper to see if predictions could be generated using a reference station, and in this                                                                                                         |
| 431 | diurnal tide dominated environment (whereas Byun and Hart 2015 had more complete data conditions, and                                                                                                                           |
| 432 | semidiurnal tide dominated regimes). Further work beyond our paper, examining the long-period tidal constituents,                                                                                                               |
| 433 | could help inform the objectives of future Antarctic tidal measurement fieldwork campaigns.                                                                                                                                     |
| 434 |                                                                                                                                                                                                                                 |
| 435 | Line 273: S

---

## Author Response (AR2)

**Reply to Editor's comments of 5 July 2020 on the paper "Predicting tidal heights for extreme environments: From 25 h observations to accurate predictions at Jang Bogo Antarctic Research Station, Ross Sea, Antarctica"**

4

37

39

- 5 Do-Seong Byun1, Deirdre E. Hart2
- 6 1Ocean Research Division, Korea Hydrographic and Oceanographic Agency, Busan 49111, Republic of Korea
- 7 2School of Earth and Environment, University of Canterbury, Christchurch 8140, Aotearoa New Zealand
- 8 Correspondence to: Deirdre Hart (deirdre.hart@canterbury.ac.nz)

Format We are grateful for the Editor's final comments on our paper. The review process has been useful in improving this 10 paper. Below we reply to the Editor's review, with each individual comment copied in blue, a response written below it, and 11 then the final modified text or figure copied below the response.

Topic Editor Decision: Publish subject to minor revisions (review by editor) (05 Jul 2020) by Philip Woodworth Comments
 to the Author: 5 July 2020

Comments on revised version of "Predicting tidal heights for extreme environments: From 25 hr observations to accurate
 predictions at Jang Bogo Antarctic Research Station, Ross Sea, Antarctica" by Byun and Hart

I did not look again in detail at the responses to the reviewers as I had seen them already. I just read the new version afresh 20 and below are some remarks. This draft is certainly much improved on the first one but there are some remaining things that 21 need attending to, some trivial, including obvious problems with a couple of figures. I believe when these issues are fixed the 22 next version should be fine.

*I must admit that there are aspects of the method used here I don't understand (other than where I can relate to them as a response method). I guess one has to use it to understand it properly, although I am sure it will interest other people.*

**27 8 - in the regional**

- **Response:** This change has been made as suggested.
- **This text now reads:** "Accurate tidal height data for the seas around Antarctica are much needed, given the crucial role of these tides in the regional and global ocean, marine cryosphere, and climate processes".

**32 *12 - using a record from**

*13 - regime*34 • **Res**

- **Response:** These two changes have been made as suggested.
- **The text now reads:** "This study evaluates the ability of a relatively new, tidal species based approach, the Complete Tidal Species Modulation with Tidal Constant Corrections (CTSM+TCC) method, to accurately predict tides for a temporary observation station in the Ross Sea, Antarctica, using a record from a neighbouring reference station characterised by a similar tidal regime".

**40 53-67 - I must say, as I think I said before, that some of this attempted justification**

- 41 *is a bit over the top. But does no harm I guess.*
  - **Response:** This text has been shortened in response to this comment.
- The text now reads: "Floating ice shelves occupy around 75% of Antarctica's perimeter (Padman et al., 2018). Tidal oscillations at the ice-ocean interface influence the location and extent of grounding zones (Padman et al., 2002), and control heat transfer and ocean mixing in cavities beneath the marine cryosphere (Padman et al., 2018) and the calving and drift of icebergs (Rignot et al. 2000). Tides also affect variability in polynyas; seasonal sea ice patterns; and thus the functioning of marine ecosystems. And tides affect the dynamics of landfast sea ice, which provides aircraft landing zones (Han and Lee, 2018).
- Accurate Antarctic region tide data are needed for models examining changes in global climate and ocean circulation
   (Han and Lee, 2018) while coastal tide data are needed for ice mass balance and motion studies (Padman et al., 2008;
   Rignot et al. 2000; Rosier and Gudmundsson, 2018). Ice thickness is typically measured by subtracting tidal heights
   from highly accurate but relatively low resolution (temporally or spatially) satellite or in situ observations of ice
   surface elevation (Padman et al., 2008). Where ice shelves and glacier tongues occur, grounding zone and ice flexure
   mechanics make ice thickness and motion determination challenging, so that accurate tidal height inputs are crucial
   (Wild et al. 2019)".

**57 68 - the applicability**

- **Response:** This change has been made as suggested.
- This text now reads: "In this study, we tested the applicability of Byun and Hart's (2015) CTSM+TCC method".
- 60

80 and 96 - could you please make it clear what the JBARS and CR instruments are delivering i.e. real sea level or subsurface pressure? In the JBARS case it must be SSP as it is a bottom-mounted instrument, unless there is some processing to 63 remove air pressure that is not mentioned. The CR instrument is a bubbler gauge I believe, although your text does not say 64 so, so that would be delivering sea level I guess - you can check with Glen Rowe. This does not matter much for diurnal and 65 semidiurnal tides but it certainly does for the longer period ones. See below.

- Response: We agree that this was not fully clear, so the different data types have been clarified in this section (see text below) as well as attention drawn to these differences in Sect. 5.1, where they become significant in the long period tides discussion.
- Concerning the JBARS data, this text now reads: "This mission collected the first, 19 day sea level related record for JBARS: 10 min interval subsurface pressure observations were recorded between 28 Jan. and 16 Feb. 2017 using a bottom-mounted pressure sensor (WTG-256S AAT, Korea), and the data were converted to sea level heights using the hydrostatic equation. High-frequency sea level oscillations (<3 hr) were removed from the observation record using a fifth-order low-pass Butterworth filter".</li>
- Concerning the ROBT data, this text now reads: "Five minute interval seawater pressure data have been collected at ROBT since November 2011 using GEOKON 4500 series standard piezometers, vented to the atmosphere, with this data converted to sea level heights using the hydrostatic equation".

**78 102 - I think you mean you infer P1 from K1 and K2 from S2? It would be best to express it that way.**

- **Response:** This change has been made as suggested.
- This text now reads: "Also the inference method was used to infer the P1 constituent from the K1, and the K2 constituent from the S2, with their amplitude ratios and phase lag differences obtained from harmonic analysis of the long-term ROBT reference station records".

- it would be worth adding 'at the two stations' again at the end of the sentence. When I read it first time it looked like
you were saying all the amplitudes were the same at each station.

- **Response:** This change has been made as suggested.
- This text now reads: "Analyses revealed that the two main diurnal (O1 and K1) and semidiurnal (M2 and S2) tides
   had similar amplitudes at the two stations, with the diurnal (semidiurnal) amplitudes being slightly larger (smaller)
   at ROBT than at JBARS, and the phase lags of all four tides having only slightly different values at the two
   stations".

**92 125 - from anytime --> throughout**

82

- **Response:** We have rewritten this sentence, dividing it sentence into two clauses, and reversing the order of the two clauses to make the meaning clearer.
- **This text now reads:** "LHr can come from any time period, but must comprise high quality (e.g. few missing data) tidal height observations throughout".

**98 143 - between the complete JBARS**

- **Response:** This change has been made as suggested.
- This text now reads: "Comparisons were made between the complete JBARS observations and the 17 prediction data sets generated for each campaign to identify which 25 hr short-term data window produced optimal  $\eta_0(\tau)$  results".

157 - 'spring tide' --> 'diurnal tide'. 'Springs' and 'neaps' are usually reserved for semidiurnals, although some people like to
refer to diurnal springs (not recommended).

- **Response:** Agreed we removed mention of spring tide for the diurnal record here.
- **This text now reads:** "The maximum tidal range occurred on 9 Feb., with step (ii) data slices from this date producing predictions with a low (but not the lowest) RMSE (4.81 cm)".

163-166 - please can you make this much clearer? What I believe you are doing is first relating 2017 JBARS to the 2017 CR
predictions through the method and seeing when it works best (Fig 5). Then you use the constants from the best section of
2017 to produce predictions for 2019. Right? (Fig 6 in principle). There is no CR data used directly in the latter. Again
right? This could be worded better.

- **Response:** This text of Sect. 4.1 has been improved to clarify the point about the data used, and to point out that this comment refers to the effect of using certain data in step 2 of the method on the final prediction results.
- This text now reads: "Interestingly, RMSEs and R2 values between the 2019 CTSM+TCC tidal predictions and observations were almost identical to those of the 2017 comparisons, revealing that our approach performed consistently across different prediction years.
- As in the 2017 experiments, the 2019 prediction dataset made using the 8 Feb. 2017 data slices (i.e., in step (ii) of the method) produced the lowest RMSE (5.3 cm) and highest  $R^2$  (0.913) values of the 2019 experiments (Fig. 5b)".

Anyway there is a problem with figures. Fig 6 has 2019 in the header but the figure itself is the same as Fig 5a. And then the end of this para refers to Fig 7 but that is for lunar declination and would be better including in the next section.

**Response:** Fig. 6 illustrated the same sort of results as in Fig. 5a, with very slight differences in the RMSEs (that were too small to pick up visually) and the same  $R^2$  values – so we have deleted the original Fig. 6 as we now see it 125 was not necessary to explain our key points here. We also swapped around order of Fig. 5. and 7. So that the 126 127 example prediction versus observation results figure now comes first (now Fig 5., formerly Fig.7) and the RMSE 128 evaluation of the predictions versus observations comes second (Fig.6 now, formerly Fig. 5). Fig. 8, which 129 focusses on lunar declination, is introduced in the next section - Sect. 4.2. 130 131 Section 4.2 - again the figures do not seem to be assigned to the text as well as they might. 132 Response: This comment made us reflect carefully on the results figure placement and order. This was also required given the recommended (and accepted) deletion of several figures. As such, we have carefully improved 133 134 the placement of references to figures in Sect 4.1 and 4.2. 135 The placement in Sect. 4.2 of Fig. 7 is now as follows: "Similarly, in a diurnal tide regime or a mixed, mainly 136 diurnal tide regime, preferred temporary station observation days can be estimated based on the lunar declination 137 (Fig. 7), which varies at a period of 13.66 days. That is, maximum tidal range days can be estimated for JBARS 138 based on the day of the Moon's greatest northern (GN) and southern (GS) declinations. The time between the 139 Moon's semi-monthly GN and GS declinations and their effects on tidal range, called the age of diurnal inequality 140 (ADI), is commonly 1 to 2 days. The GN and GS lunar declinations during our temporary station summertime 141 observation periods occurred on 8 Feb. 2017 (GN) and on 6 Jan. 2019 (GS) respectively (Fig. 7), with the 142 maximum diurnal tides at JBARS expected around 1 day after each lunar declination peak". 143 144 145 Also it occurred to me at this point - why is 25 hr so important? It is obviously the minimum for looking at one tidal cycle, 146 although you don't actually say that anywhere. But how well would say short records of 50 or 100 hours do? 147 **Response:** Yes – you are right that we omitted to explain this point explicitly. Please see below new text inserted 148 into Sect. 2, where we discuss the data sets used. This text now reads: "Note that short-term records >25 hr may be used in CTSM+TCC but, as demonstrated in 149 150 Byun and Hart (2015), large tidal range (range being twice amplitude) and high data quality have a much greater 151 positive impact on prediction results than any increase in the length of the concurrent short-term records employed". 152 153 198 - it would be worth saying somewhere that this negative AT is very unusual. You do say somewhere that AT is 1 or 2 154 days in most parts of the world so this aspect of this area is worth noting. Response: A small additional note has been added as suggested. 155 This text in Sect. 4.3 now reads: "Note that the negative AT values in Table 1 are an unusual feature of the Ross 156 157 Sea tides, given that elsewhere spring tides commonly occur a day or two after the full and new moon". 158 159 215 - better to say: Srun excluded all long-period tides (i.e. the 6 listed in Table 2) 160 216 drop 'the' 217 ditto 161 and I think you mean 'And the CSTM+TCC repeated in each case'. 162 **Response:** These changes have been made as suggested. 163 This text now reads: 164 "To investigate the main cause of the apparent fortnightly prediction biases in our results, we examined the effects of 165 two fortnightly tidal constituents (Mf, and MSf) at ROBT using T\_TIDE. Three 2019 tidal prediction experiments were 166 conducted: 167 Srun excluded all long-period tides (see list of exclusions in Table 2); • 168 Run1 was based on Srun but also incorporated Mf; and • Run2 was based on Srun but also incorporated Mf and MSf; 169 with T\_TIDE predictions made for each case. Comparisons between Run1 and Srun predictions revealed that exclusion 170 171 of the  $M_{\rm f}$  tide (2.7 cm amplitude) can produce prediction biases during periods of lunar declination change, with 172 comparisons between Run2 and Run1 results revealing that the additional exclusion of the MSf tide (1.2 cm amplitude) 173 intensifies the biases. While these results elucidate an issue with predicting Ross Sea tides based on the diurnal and semidiurnal species alone, the aforementioned differences in gauge and record types in themselves can also result in 174 175 different harmonic analysis results and, in turn, different prediction results.". 176 177 But please also see above my question at line 80 about SSP and sea level. If you have sea level at CR and SSP at JBARS then 178 you cannot assume the long period tides (especially Sa and Ssa) are the same. 179 **Response:** Yes – thank you for raising this valid point. There is a difference in the JBARS and ROBT 180 measurements and therefore their optimal usage. To fully address this issue we have now: added a note to acknowledge the different measurement types in the text describing the observations 181 0 182 collection (please see earlier in this file); 183 added the note regards the Srun/ Run1/ Run2 experiment findings (please see response just above); and 0 in addition, we have now added a note about the ROBT record used and its limitations in terms of 184 0 185 atmospheric 'noise' in the sea level record. This text of Sect. 5.1 now reads: "Table 2 summarises the characteristics of 6 long-period tides (Sa, Ssa, MSm, Mm, 186 187 Mf, MSf) at the ROBT station, derived from tidal harmonic analysis of year-long (2013) in situ observation records. 188 Note that since the ROBT observation record was derived from seawater pressure measurement, and thus includes proportionately large non-tidal (atmospheric) sea level variations, caution should be exercised in comparing the 189 190 harmonic analysis results of the non-astronomical constituents, which are affected by seawater density and 191 atmospheric forcing (i.e.  $S_a$  and  $S_{sa}$ )".

- why does an ice sheet respond only to MSf? Surely it responds to them all? 193

- 194 **Response:** Ice shelves respond at several tidal frequencies, including the MSf, as highlighted by Rosier and Gudmundsson (2008, p. 1709) in their paper: "The non-linear rheology of ice means that, as an ice shelf bends to 195 accommodate vertical tidal motion, stresses generated in the grounding zone reduce the effective viscosity of ice. 196 197 This leads to modulation of ice shelf velocity at a number of frequencies, including the MSf frequency, which is 198 readily observed on many Antarctic ice shelves (King et al., 2011; Minchew et al., 2016; Gudmundsson et al., 2017; 199 Rosier et al., 2017a)".
- Our text reads: "Rosier and Gudmundsson (2018) found that ice flows are modulated at various tidal frequencies, 200 201 including that of the MSf tide".
- 202

228-238 - surely the main thing is the amplitudes of K1 and O1 are almost the same, so they will double or cancel over a 204 fortnight, and when they are cancelling then the small semidiurnals will manifest themselves. Obvious really. 205

Response: Yes agreed. We have modified the text to emphasis this main (simpler) explanation.

This text now reads: "The combination of these out-of-phase tidal species generates double peaks (or double troughs) around low and high tide (Fig. 8b) for periods when the diurnal tide amplitude is low, due to the similar amplitude  $K_1$  and  $O_1$  tides cancelling each other out across a fortnight, allowing the combined  $M_2$  and  $S_2$  amplitudes to temporarily approach or exceed that of the combined K1 and O1 tides (Fig. 8c)".

**211 243 - please can you put underflow and overflow arrows on the colour scale of Fig 11? As it stands there is no area allowed below 0.25 or above 3 in the plot. Also there is a red (?) blob half way down the east coast of the peninsula that draws the 212 eye and you don't comment on. 213**

- **Response:** These figure changes have been made as suggested, and the red blob you mention has now been explicitly mentioned and explained.
- This text now reads: "The Weddell Sea is dominated by mixed, mainly semidiurnal tides, excepting the 216 217 semidiurnal area mentioned and another small area exhibiting diurnal tides (F>3) at around 76.5°S, where 218 amphidromic points (i.e. zero amplitudes) occur for both the M2 and S2 tides".
- 219 Figure 9 (formerly Fig. 11) now looks like this:

Figure 9. Distribution of tidal form factor (F) values around Antarctica. Note the magenta area (72°S) on the Antarctic Peninsula's 222 Weddell Sea coast denotes the only area with a properly semidiurnal tide regime (F < 0.25) in the Antarctic region.

251-256 - these lines are just repeating yourself.

- **Response:** We agree that this was repetitive, considering what is stated earlier in the introduction and in the subsequent conclusion section, so have deleted the original lines 253 to 256.
- 227 The ending of Section 5.2 now reads: "As the nodal amplitude factor variations of the diurnal and semidiurnal 228 tides are out of phase, this leads to differing tidal responses around Antarctica over 18.61 years, particularly between the Ross and Weddell Seas (see details for ROBT in Byun and Hart, 2019). Given that CTSM+TCC is 229 230 based on modulated tidal amplitude and phase lag corrections for each diurnal and semidiurnal species, this

402 - records --> record (three times)

248

**404 - you mean P1 from K1 and K2 from S2. Your use of brackets is confusing.**

- **Response:** These changes and clarifications have been made as suggested.
- This caption text now reads: "Table 1. Major tidal harmonic results for diurnal and semidiurnal constituents from harmonic analyses of sea level observations: the year-long (2013) record from Cape Roberts (ROBT), and 17.04 day record (29 Jan. to 15 Feb. 2017) and 20.54 day record (29 Dec. 2018 to 18 Jan. 2019) from Jang Bogo Antarctic Research Station (JBARS) in the Ross Sea (see source details in Sect. 2). For the JBARS tidal harmonic analyses, the inference method was used to infer the P1 constituent from the K1, and the K2 constituent from the S2, with their amplitude ratios and phase lag differences obtained from harmonic analysis of the long-term ROBT 2013 reference station record".

approach is applicable in studying a continent with such a diversity of tidal regime types".

408 - from a harmonic analysis of one year-long

- **Response:** This change has been made as suggested.
- **This text now reads:** "Table 2. Harmonic constants for 6 long-period tidal constituents, derived from harmonic analysis of one year-long observations (2013) measured at the Cape Roberts sea level gauge (ROBT), using T\_Tide (Pawlowicz et al., 2002). Note that this gauge is a vented piezometer so caution should be exercised in interpreting the results (particularly those for Sa and Ssa) given the inclusion of proportionately large non-tidal (atmospheric) variations in this kind of sea level record".

**Please see my question at line 80. Sa and Ssa will be very different if SSP or sea level are being used, although I believe if CR is a bubbler gauge then it should be sea level ok - check with Glen.**

• **Response:** Thank you for raising this point – please see paper adjustments detailed earlier in this file.

**256 Fig 2b - the lons and lats are upside down**

- **Response:** Yes the orientation of these have been corrected.
- Figure 2 now looks like:

260

*Fig 4f - degrees is missing from y-title*

11g 4J acgrees is missing from y title 262 430 - of the entire 369 ... record ... entire 17 ... record

- **Response:** These figure and caption corrections have been made as suggested.
- Figure 4 and its caption now look like:
- 264 265

---

## Author Response (AR3)

*Reply to* **Topic Editor's decision and comments of 26 July 2020 on the paper "Predicting tidal**
**heights for extreme environments: From 25 h observations to accurate predictions at Jang Bogo**
**Antarctic Research Station, Ross Sea, Antarctica"**

Do-Seong Byun[1], Deirdre E. Hart[2]

[1]Ocean Research Division, Korea Hydrographic and Oceanographic Agency, Busan 49111, Republic of Korea
[2]School of Earth and Environment, University of Canterbury, Christchurch 8140, Aotearoa New Zealand

*Correspondence to*: Deirdre Hart (deirdre.hart@canterbury.ac.nz)

**Format** We are grateful for the Editor's final technical corrections on our paper. Below we reply to the individual comments
(copied in blue), with a response, and then the final modified text.
*Topic Editor Decision: Publish subject to technical corrections (26 Jul 2020) by Philip Woodworth*
*Comments to the Author: 26 July 2020*
*Remaining comments on this manuscript, mainly to do with describing the instruments again:*
*77 - thanks for clarifying the instruments. I think I would add 'absolute' and 'equivalent' to this to make it clear to the reader*
*that this instrument is not measuring sea level as such i.e. "using a bottom-mounted absolute pressure sensor (WTG-256S*
*AAT, Korea) with the data converted to equivalent sea level heights using the hydrostatic equation". That is, if I have*
*understood you correctly.*
• **Response:** This change has been made exactly as suggested.
• **This text now reads:** "using a bottom-mounted absolute pressure sensor (WTG-256S AAT, Korea) with the data
converted to equivalent sea level heights using the hydrostatic equation"
*218 - I would write this: Note that since the ROBT observation record was derived from a differential (vented) pressure sensor,*
*unlike the absolute sensor at JBARS, and thus it .... affected by atmospheric (air pressure) forcing (i.e. Sa and Ssa).*
• **Response:** This change has been made exactly as suggested.
• **This text now reads:** "Note that since the ROBT observation record was derived from a differential (vented)
pressure sensor, and thus it includes proportionately large non-tidal (atmospheric) sea level variations, caution
should be exercised in comparing the harmonic analysis results of the non-astronomical constituents, which are
affected by atmospheric (air pressure) forcing (i.e. $S_a$ and $S_{sa}$)".
*229-232 - you say now: "While these results elucidate an issue with predicting Ross Sea tides based on the diurnal and*
*semidiurnal species alone, the aforementioned differences in gauge and record types in themselves can also result in different*
*harmonic analysis results and, in turn, different prediction results". I guess you are now trying to accommodate the gauge*
*type issue at line 77. Perhaps reword to be clearer if you agree but of course use your own words: "These results indicate one*
*particular issue to do with long-period tides when predicting Ross Sea tides based on the diurnal and semidiurnal species*
*alone. We note that the aforementioned differences in gauge records (subsurface pressure or real sea level) introduce another,*
*in that, while diurnal and semidiurnal tides can be considered to be measured equivalently accurately, the longer-period*
*components might be expected to instrument-dependent and so have uncertainties for the above experiments".*
• **Response:** This change has been made as suggested.
• **This text now reads:** "These results elucidate one particular issue to do with long-period tides when predicting Ross
Sea tides based on the diurnal and semidiurnal species alone. We note that the aforementioned differences in gauge
records (subsurface pressure or real sea level) introduce another. That is, while the diurnal and semidiurnal tides might
be considered to be measured equivalently accurately, the longer-period components are instrument-dependent and
so have uncertainties for the above experiments".
*419 - a one year-long observation*
• **Response:** This change has been made exactly as suggested.
• **This text now reads:** "Table 2. Harmonic constants for 6 long-period tidal constituents, derived from harmonic
analysis of a one year-long observation (2013) measured at the Cape Roberts sea level gauge (ROBT)".
*Figure 6 - I think I would drop <Key> although up to you.*
**Response:** We decided to keep the word <Key> in this figure.

[revised manuscript text omitted]